# Polynomial Width is Sufficient for Set Representation with High-dimensional Features

**Peihao Wang[1], Shenghao Yang [3], Shu Li [4], Zhangyang Wang [1], Pan Li[2,4]**
[1]University of Texas at Austin, [2]Georgia Tech, [3]University of Waterloo, [4]Purdue University
{peihaowang,atlaswang}@utexas.edu, panli@gatech.edu,
shenghao.yang@uwaterloo.ca, shuli@purdue.edu,

## Abstract

Set representation has become ubiquitous in deep learning for modeling the inductive bias of neural networks that are insensitive to the input order. DeepSets is the most widely used neural network architecture for set representation. It involves embedding each set element into a latent space with dimension $L$, followed by a sum pooling to obtain a whole-set embedding, and finally mapping the whole-set embedding to the output. In this work, we investigate the impact of the dimension $L$ on the expressive power of DeepSets. Previous analyses either oversimplified high-dimensional features to be one-dimensional features or were limited to complex analytic activations, thereby diverging from practical use or resulting in $L$ that grows exponentially with the set size $N$ and feature dimension $D$. To investigate the minimal value of $L$ that achieves sufficient expressive power, we present two set-element embedding layers: (a) linear + power activation (LP) and (b) linear + exponential activations (LE). We demonstrate that $L$ being $\mathrm{poly}(N, D)$ is sufficient for set representation using both embedding layers. We also provide a lower bound of $L$ for the LP embedding layer. Furthermore, we extend our results to permutation-equivariant set functions and the complex field.

## 1 Introduction

Enforcing invariance into neural network architectures has become a widely-used principle to design deep learning models (LeCun et al., 1995; Cohen & Welling, 2016; Bronstein et al., 2017; Kondor & Trivedi, 2018; Maron et al., 2018; Bogatskiy et al., 2020; Wang et al., 2023). In particular, when a task is to learn a function with a set as the input, the architecture enforces permutation invariance that asks the output to be invariant to the permutation of the input set elements (Qi et al., 2017; Zaheer et al., 2017). Neural networks to learn a set function have found a variety of applications in particle physics (Mikuni & Canelli, 2021; Qu & Gouskos, 2020), computer vision (Zhao et al., 2021; Lee et al., 2019) and population statistics (Zhang et al., 2019; 2020; Grover et al., 2020), and have recently become a fundamental module (the aggregation operation of neighbors' features in a graph (Morris et al., 2019; Xu et al., 2019; Corso et al., 2020)) in graph neural networks (GNNs) (Scarselli et al., 2008; Hamilton et al., 2017) that show even broader applications.

Previous works have studied the expressive power of neural network architectures to represent set functions (Qi et al., 2017; Zaheer et al., 2017; Maron et al., 2019; Wagstaff et al., 2019; 2022; Segol & Lipman, 2020; Zweig & Bruna, 2022). Formally, a set with $N$ elements can be represented as $\mathcal{S} = \{\boldsymbol{x}^{(1)}, \cdots, \boldsymbol{x}^{(N)}\}$ where $\boldsymbol{x}^{(i)}$ is in a feature space $\mathcal{X}$, typically $\mathcal{X} = \mathbb{R}^D$. To represent a set function that takes $\mathcal{S}$ and outputs a real value, the most widely used architecture DeepSets (Zaheer et al., 2017) follows Eq. 1.

$$f(\mathcal{S}) = \rho\left(\sum_{i=1}^{N} \phi(\boldsymbol{x}^{(i)})\right), \text{where } \phi : \mathcal{X} \to \mathbb{R}^L \text{ and } \rho : \mathbb{R}^L \to \mathbb{R} \text{ are continuous functions.} \quad (1)$$

DeepSets encodes each set element individually via $\phi$, and then maps the encoded vectors after sum pooling to the output via $\rho$. The continuity of $\phi$ and $\rho$ ensure that they can be well approximated by fully-connected neural networks (Cybenko, 1989; Hornik et al., 1989), which has practical implications. DeepSets enforces permutation invariance because of the sum pooling, as shuffling

Table 1: A comprehensive comparison among all prior works on expressiveness analysis with $L$. Our results achieve the tightest bound on $L$ while being able to analyze high-dimensional set features and extend to the equivariance case.

| Prior Arts | $L$ | $D > 1$ | Exact Rep. | Equivariance |
|---|---|---|---|---|
| DeepSets (Zaheer et al., 2017) | $N + 1$ | ✗ | ✓ | ✓ |
| Wagstaff et al. (Wagstaff et al., 2019) | $N$ | ✗ | ✓ | ✓ |
| Segol et al. (Segol & Lipman, 2020) | $\binom{N+D}{N} - 1$ | ✓ | ✗ | ✓ |
| Zweig & Bruna (Zweig & Bruna, 2022) | $\exp(\min\{\sqrt{N}, D\})$ | ✓ | ✗ | ✗ |
| Our results | $\text{poly}(N, D)$ | ✓ | ✓ | ✓ |

the order of $\boldsymbol{x}^{(i)}$ does not change the output. However, the sum pooling compresses the whole set into an $L$-dimension vector, which places an information bottleneck in the middle of the architecture. Therefore, a core question on using DeepSets for set function representation is that given the input feature dimension $D$ and the set size $N$, what the minimal $L$ is needed so that the architecture Eq. 1 can represent/universally approximate any continuous set functions. The question has attracted attention in many previous works (Zaheer et al., 2017; Wagstaff et al., 2019; 2022; Segol & Lipman, 2020; Zweig & Bruna, 2022) and is the focus of the present work.

An extensive understanding has been achieved for the case with one-dimensional features ($D = 1$). Zaheer et al. (Zaheer et al., 2017) proved that this architecture with bottleneck dimension $L = N$ suffices to *accurately* represent any continuous set functions when $D = 1$. Later, Wagstaff et al. proved that accurate representations cannot be achieved when $L < N$ (Wagstaff et al., 2019) and further strengthened the statement to *a failure in approximation* to arbitrary precision in the infinity norm when $L < N$ (Wagstaff et al., 2022).

However, for the case with high-dimensional features ($D > 1$), the characterization of the minimal possible $L$ is still missing. Most of previous works (Zaheer et al., 2017; Segol & Lipman, 2020; Gui et al., 2021) proposed to generate multi-symmetric polynomials to approximate permutation invariant functions (Bourbaki, 2007). As the algebraic basis of multi-symmetric polynomials is of size $L^* = \binom{N+D}{N} - 1$ (Rydh, 2007) (exponential in $\min\{D, N\}$), these works by default claim that if $L \geq L^*$, $f$ in Eq. 1 can approximate any continuous set functions, while they do not check the possibility of using a smaller $L$. Zweig & Bruna (2022) constructed a set function that $f$ requires bottleneck dimension $L > N^{-2} \exp(O(\min\{D, \sqrt{N}\}))$ (still exponential in $\min\{D, \sqrt{N}\}$) to approximate while it relies on the condition that $\phi, \rho$ only adopt *complex analytic activation functions*. This condition is overly strict, as many practical neural networks work on real numbers [1] and even allow the use of non-analytic activations, such as ReLU. Zweig & Bruna thus left an open question *whether the exponential dependence on $N$ or $D$ of $L$ is still necessary if $\phi, \rho$ work in the real domain and allow using non-analytic activations.*

**Our Contribution.** The main contribution of this work is to confirm a negative response to the above question. Specifically, we present the first theoretical justification that $L$ being *polynomial* in $N$ and $D$ is sufficient for DeepSets (Eq. 1) like architecture to represent any real/complex *continuous* set functions with *high-dimensional* features ($D > 1$). To mitigate the gap to the practical use, we consider two architectures to implement $\phi$ (in Eq. 1) and specify the bounds on $L$ accordingly:

- $\phi$ adopts *a linear layer with power mapping*: The minimal $L$ holds a lower bound and an upper bound, which is $N(D + 1) \leq L < N^5 D^2$.

- $\phi$ adopts *a linear layer plus an exponential activation function*: The minimal $L$ holds a tighter upper bound $L \leq N^4 D^2$.

We start from the real domain and prove that if the function $\rho$ could be any continuous function, the above two architectures reproduce the precise construction of any set functions for high-dimensional features $D > 1$, akin to the result in Zaheer et al. (2017) for $D = 1$. This result contrasts with Segol & Lipman (2020); Zweig & Bruna (2022) which only present approximating representations. If $\rho$

---

[1]Note that for $x, y \in \mathbb{R}$, the function in the complex domain $f_1(x + y\sqrt{-1}) = x$ is not complex analytic, while the function with real variables $f_2([x, y]) = x$ can be simply and accurately implemented by practical neural networks. Moreover, complex analytic functions are not dense in the space of complex continuous functions, while polynomials (and thus real analytic functions) are dense in the space of real continuous functions. So, the assumption considered in (Zweig & Bruna, 2022) substantially limits the space of the functions that can be approximated.

adopts a fully-connected neural network that allows approximation of any real continuous functions on a bounded input space (Cybenko, 1989; Hornik et al., 1989), then the DeepSets architecture $f(\cdot)$ can approximate any set functions universally on that bounded input space. We extend our theory to permutation-equivariant functions and set functions in the complex field, where the minimal $L$ shares the same bounds up to some multiplicative constants.

Another comment on our contributions is that Zweig & Bruna (2022) leverage difference in the needed dimension $L$, albeit with the complex analytic assumption, to illustrate the gap between DeepSets (Zaheer et al., 2017) and Relational Network (Santoro et al., 2017) in their expressive powers, where the latter encodes set elements in a pairwise manner rather than in an element-wise separate manner. The gap well explains the empirical observation that Relational Network achieves better expressive power with smaller $L$ (Murphy et al., 2018; Wagstaff et al., 2019). Our theory does not violate such an observation while it shows that without the above strict assumption, the gap can be reduced from an exponential order in $N$ and $D$ to a polynomial order.

**Practical Implications.** Many real-world applications have computation constraints where only DeepSets instead of Relational Network can be used, e.g., the neighbor aggregation operation in GNN being applied to large networks (Hamilton et al., 2017), and hypergraph neural diffusion operations in hypergraph neural networks (Wang et al., 2023). Our theory points out that in this case, it is sufficient to use polynomial $L$ dimension to embed each element, while one needs to adopt a decoder network $\rho$ with non-analytic activations.

## 2 PRELIMINARIES

### 2.1 NOTATIONS AND PROBLEM SETUP

We are interested in the approximation and representation of functions defined over sets [2]. We start with the real field and then extend the result. In convention, an $N$-sized set $\mathcal{S} = \{\boldsymbol{x}^{(1)}, \cdots, \boldsymbol{x}^{(N)}\}$, where $\boldsymbol{x}^{(i)} \in \mathbb{R}^D, \forall i \in [N](\triangleq \{1, 2, ..., N\})$, can be denoted by a data matrix $\boldsymbol{X} = \begin{bmatrix} \boldsymbol{x}^{(1)} & \cdots & \boldsymbol{x}^{(N)} \end{bmatrix}^\top \in \mathbb{R}^{N \times D}$. Note that we use the superscript $(i)$ to denote the $i$-th set element and the subscript $i$ to denote the $i$-th column/feature channel of $\boldsymbol{X}$, i.e., $\boldsymbol{x}_i = \begin{bmatrix} x_i^{(1)} & \cdots & x_i^{(N)} \end{bmatrix}^\top$. Let $\Pi(N)$ denote the set of all $N$-by-$N$ permutation matrices. To characterize the unorderedness of a set, we define an equivalence class over $\mathbb{R}^{N \times D}$:

**Definition 2.1** (Equivalence Class). If matrices $\boldsymbol{X}, \boldsymbol{X}' \in \mathbb{R}^{N \times D}$ represent the same set $\mathcal{X}$, then they are called equivalent up a row permutation, denoted as $\boldsymbol{X} \sim \boldsymbol{X}'$. Or equivalently, $\boldsymbol{X} \sim \boldsymbol{X}'$ if and only if there exists a matrix $\boldsymbol{P} \in \Pi(N)$ such that $\boldsymbol{X} = \boldsymbol{P}\boldsymbol{X}'$.

Set functions can be in general considered as permutation-invariant or permutation-equivariant functions, which process the input matrices regardless of the order by which rows are organized. The formal definitions of permutation-invariant/equivariant functions are presented as below:

**Definition 2.2.** (Permutation Invariance) A function $f : \mathbb{R}^{N \times D} \to \mathbb{R}^{D'}$ is called permutation-invariant if $f(\boldsymbol{P}\boldsymbol{X}) = f(\boldsymbol{X})$ for any $\boldsymbol{P} \in \Pi(N)$.

**Definition 2.3.** (Permutation Equivariance) A function $f : \mathbb{R}^{N \times D} \to \mathbb{R}^{N \times D'}$ is called permutation-equivariant if $f(\boldsymbol{P}\boldsymbol{X}) = \boldsymbol{P}f(\boldsymbol{X})$ for any $\boldsymbol{P} \in \Pi(N)$.

In this paper, we investigate the approach to designing a neural network architecture with permutation invariance/equivariance. Below we will first focus on permutation-invariant functions $f : \mathbb{R}^{N \times D} \to \mathbb{R}$. Then, in Sec. 5, we show that we can easily extend the established results to permutation-equivariant functions through the results provided in Sannai et al. (2019); Wang et al. (2023) and to the complex field. The obtained results for $D' = 1$ can also be easily extended to $D' > 1$ as otherwise $f$ can be written as $\begin{bmatrix} f_1 & \cdots & f_{D'} \end{bmatrix}^\top$ and each $f_i$ has single output feature channel.

---

[2]In fact, we allow repeating elements in $\mathcal{S}$, therefore, $\mathcal{S}$ should be more precisely called multiset. With a slight abuse of terminology, we interchangeably use terms multiset and set throughout the whole paper.

## 2.2 DEEPSETS AND THE PROOF FOR THE ONE-DIMENSIONAL CASE ($D = 1$)

The seminal work Zaheer et al. (2017) establishes the following result which induces a neural network architecture for permutation-invariant functions.

**Theorem 2.4** (DeepSets (Zaheer et al., 2017), $D = 1$). *A continuous function $f : \mathbb{R}^N \to \mathbb{R}$ is permutation-invariant (i.e., a set function) if and only if there exists continuous functions $\phi : \mathbb{R} \to \mathbb{R}^L$ and $\rho : \mathbb{R}^L \to \mathbb{R}$ such that $f(\boldsymbol{X}) = \rho\left(\sum_{i=1}^N \phi(x^{(i)})\right)$, where $L$ can be as small as $N$. Note that, here $x^{(i)} \in \mathbb{R}$ is a scalar.*

*Remark* 2.5. The original result presented in Zaheer et al. (2017) states the latent dimension should be as large as $N + 1$. Wagstaff et al. (2019) tighten this dimension to exactly $N$.

Theorem 2.4 implies that as long as the latent space dimension $L \geq N$, any permutation-invariant functions can be implemented in a unified manner as DeepSets (Eq.1). Furthermore, DeepSets suggests a useful architecture for $\phi$ at the analysis convenience and empirical utility, which is formally defined below (in DeepSets, $\phi$ is set as $\psi_L$):

**Definition 2.6** (Power mapping). A power mapping of degree $K$ is a function $\psi_K : \mathbb{R} \to \mathbb{R}^K$ which transforms a scalar to a power series: $\psi_K(z) = \begin{bmatrix} z & z^2 & \cdots & z^K \end{bmatrix}^\top$.

However, DeepSets (Zaheer et al., 2017) focuses on the case that the feature dimension of each set element is one (i.e., $D = 1$). To demonstrate the difficulty of extending Theorem 2.4 to high-dimensional features, we reproduce the proof next, which simultaneously reveals its significance and limitation. Some intermediate results and mathematical tools will be recalled later in our proof.

We begin by defining sum-of-power mapping (of degree $K$) $\Psi_K(\boldsymbol{X}) = \sum_{i=1}^N \psi_K(x^{(i)})$, where $\psi_K$ is the power mapping following Definition 2.6. Afterward, we reveal that sum-of-power mapping $\Psi_K$ has a continuous inverse. Before stating the formal argument, we formally define the injectivity of permutation-invariant mappings:

**Definition 2.7** (Injectivity). A set function $f : \mathbb{R}^{N \times D} \to \mathbb{R}^L$ is said to be injective if and only if $\forall \boldsymbol{X}, \boldsymbol{X'} \in \mathbb{R}^{N \times D}, f(\boldsymbol{X}) = f(\boldsymbol{X'})$ implies $\boldsymbol{X} \sim \boldsymbol{X'}$.

As summarized in the following lemma shown by Zaheer et al. (2017) and improved by Wagstaff et al. (2019), $\Psi_N$ (i.e., when $K = N$) is an injective mapping. If we further constrain the image space to be the range of $\Psi_N$: $\mathcal{Z} = \{\Psi_N(\boldsymbol{X}) : \forall \boldsymbol{X} \in \mathbb{R}^N\} \subseteq \mathbb{R}^N$, then $\Psi_N$ becomes surjective and is shown to have a continuous inverse. This result comes from homeomorphism between roots and coefficients of monic polynomials (Ćurgus & Mascioni, 2006).

**Lemma 2.8** (Existence of Continuous Inverse of Sum-of-Power (Zaheer et al., 2017; Wagstaff et al., 2019)). $\Psi_N : \mathbb{R}^N \to \mathcal{Z}$ *is injective, thus there exists $\Psi_N^{-1} : \mathcal{Z} \to \mathbb{R}^N$ such that $\Psi_N^{-1} \circ \Psi_N(\boldsymbol{X}) \sim \boldsymbol{X}$. Moreover, $\Psi_N^{-1}$ is continuous.*

Now we are ready to prove necessity in Theorem 2.4 as sufficiency is easy to check. By choosing $\phi = \psi_N : \mathbb{R} \to \mathbb{R}^N$ to be the power mapping (cf. Definition 2.6), and $\rho = f \circ \Psi_N^{-1}$. For any scalar-valued set $\boldsymbol{X} = \begin{bmatrix} x^{(1)} & \cdots & x^{(N)} \end{bmatrix}^\top$, $\rho\left(\sum_{i=1}^N \phi(x^{(i)})\right) = f \circ \Psi_N^{-1} \circ \Psi_N(\boldsymbol{x}) = f(\boldsymbol{PX}) = f(\boldsymbol{X})$ for some $\boldsymbol{P} \in \Pi(N)$. The existence and continuity of $\Psi_N^{-1}$ are due to Lemma 2.8.

Theorem 2.4 gives the *exact decomposable form* (Wagstaff et al., 2019) for permutation-invariant functions, which is stricter than approximation error based expressiveness analysis. In summary, the key idea is to establish a mapping $\phi$ whose element-wise sum-pooling has a continuous inverse.

## 2.3 CURSE OF HIGH-DIMENSIONAL FEATURES ($D \geq 2$)

We argue that the proof of Theorem 2.4 is not applicable to high-dimensional set features ($D \geq 2$). The main reason is that power mapping defined in Definition 2.6 only receives scalar input. It remains elusive how to extend it to a multivariate version that admits injectivity and a continuous inverse. A plausible idea seems to be applying power mapping for each channel $\boldsymbol{x}_i$ independently, and due to the injectivity of sum-of-power mapping $\Psi_N$, each channel can be uniquely recovered individually via the inverse $\Psi_N^{-1}$. However, we point out that each recovered feature channel $\boldsymbol{x'}_i \sim \boldsymbol{x}_i, \forall i \in [D]$, does not imply $[\boldsymbol{x'}_1 \quad \cdots \quad \boldsymbol{x'}_D] \sim \boldsymbol{X}$, where the alignment of features across channels gets lost.

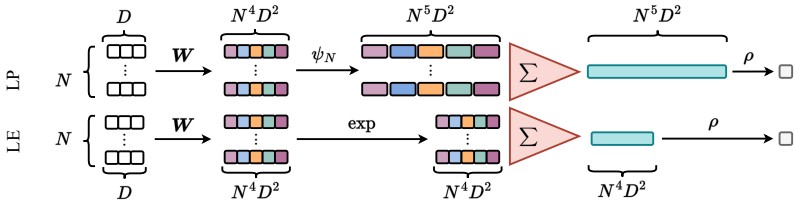

Figure 1: Illustration of the proposed linear + power mapping embedding layer (LP) and linear + exponential activation embedding layer (LE).

Hence, channel-wise power encoding no more composes an injective mapping. Zaheer et al. (2017) proposed to adopt multivariate polynomials as $\phi$ for high-dimensional case, which leverages the fact that multivariate symmetric polynomials are dense in the space of permutation invariant functions (akin to Stone-Wasserstein theorem) (Bourbaki, 2007). This idea later got formalized in the work of Segol & Lipman (2020) by setting $\phi(\boldsymbol{x}^{(i)}) = \left[ \cdots \quad \prod_{j \in [D]} (x_j^{(i)})^{\alpha_j} \quad \cdots \right]$ where $\boldsymbol{\alpha} \in \mathbb{N}^D$ traverses all $\sum_{j \in [D]} \alpha_j \leq n$ and extended to permutation equivariant functions. Nevertheless, the dimension $L = \binom{N+D}{D}$, i.e., exponential in $\min\{N, D\}$ in this case, and unlike DeepSets (Zaheer et al., 2017) which exactly recovers $f$ for $D = 1$, the architecture in Zaheer et al. (2017); Segol & Lipman (2020) can only approximate the desired function.

## 3 MAIN RESULTS

In this section, we present our main result which extends Theorem 2.4 to high-dimensional features. Our conclusion is that to universally represent a set function on sets of length $N$ and feature dimension $D$ with the DeepSets architecture (Zaheer et al., 2017) (Eq. 1), the *minimal $L$* needed for expressing the intermediate embedding space is *at most* polynomial in $N$ and $D$.

Formally, we summarize our main result in the following theorem.

**Theorem 3.1** (The main result). *Suppose $D \geq 2$. For any continuous permutation-invariant function $f : \mathbb{R}^{N \times D} \to \mathbb{R}$, there exists two continuous mappings $\phi : \mathbb{R}^D \to \mathbb{R}^L$ and $\rho : \mathcal{Z} \to \mathbb{R}$ such that for every $\boldsymbol{X} \in \mathbb{R}^{N \times D}$, $f(\boldsymbol{X}) = \rho \left( \sum_{i=1}^{N} \phi(\boldsymbol{x}^{(i)}) \right)$, where $\mathcal{Z} = \left\{ \sum_{i=1}^{N} \phi(\boldsymbol{x}^{(i)}) : \forall \boldsymbol{X} \in \mathbb{R}^{N \times D} \right\} \subset \mathbb{R}^L$ is the image of the sum-pooling and*

- *$L \in [N(D+1), N^5 D^2]$ when $\phi$ admits **linear layer + power mapping (LP)** architecture:*

$$\phi(\boldsymbol{x}) = \left[ \psi_N(\boldsymbol{w}_1^\top \boldsymbol{x})^\top \quad \cdots \quad \psi_N(\boldsymbol{w}_K^\top \boldsymbol{x})^\top \right] \tag{2}$$

*for some $\boldsymbol{w}_1, \cdots, \boldsymbol{w}_K \in \mathbb{R}^D$, and $K = L/N$.*

- *$L \in [ND, N^4 D^2]$ when $\phi$ admits **linear layer + exponential activation (LE)** architecture:*

$$\phi(\boldsymbol{x}) = \left[ \exp(\boldsymbol{v}_1^\top \boldsymbol{x}) \quad \cdots \quad \exp(\boldsymbol{v}_L^\top \boldsymbol{x}) \right] \tag{3}$$

*for some $\boldsymbol{v}_1, \cdots, \boldsymbol{v}_L \in \mathbb{R}^D$.*

The bounds of $L$ depend on the choice of the architecture of $\phi$, which are illustrated in Fig. 1. In the LP setting, we adopt a linear layer that maps each set element into $K$ dimension. Then we apply a channel-wise power mapping that separately transforms each value in the feature vector into an $N$-order power series, and concatenates all the activations together, resulting in a $KN$ dimension feature. The LP architecture is closer to DeepSets (Zaheer et al., 2017) as they share the power mapping as the main component. Theorem 3.1 guarantees the existence of $\rho$ and $\phi$ (in the form of Eq. 2) which satisfy Eq. 1 without the need to set $K$ larger than $N^4 D^2$ while $K \geq D + 1$ is necessary. Therefore, the total embedding size $L = KN$ is bounded by $N^5 D^2$ above and $N(D + 1)$ below. Note that this lower bound is not trivial as $ND$ is the degree of freedom of the input $\boldsymbol{X}$. No matter how $\boldsymbol{w}_1, ..., \boldsymbol{w}_K$ are adopted, one cannot achieve an injective mapping by just using $ND$ dimension.

In the LE architecture, we investigate the utilization of the exponential activation in set representation, which is also a valid activation function to build deep neural networks (Barron, 1993; Clevert et al., 2015). Each set entry will be linearly lifted into an $L$-dimensional space via a group of weights and then transformed by an element-wise exponential activation. The advantage of the exponential function is that the upper bound of $L$ is improved to be $N^4 D^2$. The lower bound $ND$ for the LE

architecture is a trivial bound due to the degree of freedom of the inputs. Essentially, a linear layer followed by an exponential function is equivalent to applying monomials onto exponential activations. If monomial activations are allowed as used in Segol & Lipman (2020), we can also replace the exponential function with a series of monomial mappings while yielding the same upper bound. However, in contrast to Segol & Lipman (2020), where exponentially many monomials are required, our construction of the linear weights enables a mere reliance on bivariate monomials of degree $D$, thus reducing the number of needed monomials to $O(N^2 D)$.

*Remark* 3.2. Unlike $\phi$, the form of $\rho$ cannot be explicitly specified, as it depends on the desired function $f$. The complexity of $\rho$ remains unexplored in this paper, which may be high in practice.

**Empirical Validation.** In Appendix A, we run numerical experiments to verify our argument. Fig. 2 demonstrates the polynomial dependence between the set size, feature dimension, and the minimal latent embedding dimension to achieve a small approximation error. See more details in Appendix A.

**Importance of Continuity.** We argue that the requirements of continuity on $\rho$ and $\phi$ are essential for our discussion. First, practical neural networks can only provably approximate continuous functions (Cybenko, 1989; Hornik et al., 1989). Moreover, set representation without such requirements can be straightforward (but likely meaningless in practice). It is known that there exists a discontinuous bijective mapping $r : \mathbb{R}^D \to \mathbb{R}$ if $D \geq 2$. If we let $r$ map the high-dimensional features to scalars, then its inverse exists and the same proof of Theorem 2.4 goes through, i.e. let $\phi = \psi_N \circ r$ and $\rho = f \circ r^{-1} \circ \Psi_N^{-1}$. However, we note both $\rho$ and $\phi$ lose continuity.

**Comparison with Prior Results.** Below we highlight the significance of Theorem 3.1. A quick overview has been listed in Table 1 for illustration. The lower bound in Theorem 3.1 corrects a natural misconception that the degree of freedom (i.e., $L = ND$ for multi-channel cases) is not enough for representing the embedding space (Wagstaff et al., 2022). Compared with Zweig & Bruna's finding, our result significantly improves this bound on $L$ from exponential to polynomial by allowing continuous activations that may not be complex analytic. Proof-wise, Zweig & Bruna's proof idea is hard to extend to the real domain, while ours applies to both real and complex domains and equivariant functions. Dym & Gortler (2024); Amir et al. (2024) present significant results that $L$ can be as small as $2ND + 1$. However, the continuity of decoder $\rho$ is not guaranteed when the domain of $f$ is an open set.

## 4 PROOF SKETCH

In this section, we introduce the proof techniques of Theorem 3.1, while deferring a full version and all missing proofs to the appendix. The proof is constructive and mainly consists of three steps below:

1. For the LP architecture, we construct a group of $K$ linear weights $\boldsymbol{w}_1 \cdots, \boldsymbol{w}_K \in \mathbb{R}^D$ with $K \leq N^4 D^2$, while for the LE architecture, we construct a group of $L$ linear weights $\boldsymbol{v}_1 \cdots, \boldsymbol{v}_L \in \mathbb{R}^D$ with $L \leq N^4 D^2$, such that the summation over the associated embeddings $\Phi(\boldsymbol{X}) = \sum_{i=1}^N \phi(\boldsymbol{x}^{(i)})$ is *injective*. Moreover, if $K \leq D$ for LP layer or trivially $L < ND$ for LE layer, such weights do not exist, which induces the lower bounds.

2. Given the injectivity of both LP and LE layers, we constrain the image spaces to be their ranges $\{\Phi(\boldsymbol{X}) : \boldsymbol{X} \in \mathbb{R}^{N \times D}\}$, respectively, and thus, the inverse of the sum-pooling $\Phi^{-1}$ exists. Furthermore, we show that $\Phi^{-1}$ is *continuous*.

3. Then the proof of upper bounds can be concluded for both settings by letting $\rho = f \circ \Phi^{-1}$ since
$$\rho\left(\sum_{i=1}^N \phi(\boldsymbol{x}^{(i)})\right) = f \circ \Phi^{-1} \circ \Phi(\boldsymbol{X}) = f(\boldsymbol{P}\boldsymbol{X}) = f(\boldsymbol{X}) \text{ for some } \boldsymbol{P} \in \Pi(N).$$

Next, we elaborate on the construction idea which yields injectivity for both embedding layers in Sec. 4.1 with the notion of anchor. In Sec. 4.2, we prove the continuity of the inverse map for LP and LE via arguments similar to Ćurgus & Mascioni (2006).

### 4.1 INJECTIVITY

The high-level ideas of construction and proofs are illustrated in Fig. 3, in which we first construct an *anchor*, a mathematical device introduced in Sec. 4.1.1 to induce injectivity, and then mix each

feature channel with the anchor through coupling schemes specified by LP (Sec. 4.1.2) and LE (Sec. 4.1.3) layers, respectively.

### 4.1.1 ANCHOR

Constructing an anchor stands at the core of our proof. Formally, we define *anchor* as below:

**Definition 4.1** (Anchor). Consider the data matrix $\boldsymbol{X} \in \mathbb{R}^{N \times D}$, then $\boldsymbol{a} \in \mathbb{R}^N$ is called an anchor of $\boldsymbol{X}$ if $\boldsymbol{a}_i \neq \boldsymbol{a}_j$ for any $i, j \in [N]$ such that $\boldsymbol{x}^{(i)} \neq \boldsymbol{x}^{(j)}$.

In plain language, by Definition 4.1, two entries in the anchor must be distinctive if the set elements at the corresponding indices are not equal. As a result, the union alignment property can be derived:

**Lemma 4.2** (Union Alignment). *Consider two data matrices $\boldsymbol{X}, \boldsymbol{X}' \in \mathbb{R}^{N \times D}$, $\boldsymbol{a} \in \mathbb{R}^N$ is an anchor of $\boldsymbol{X}$ and $\boldsymbol{a}' \in \mathbb{R}^N$ is an arbitrary vector. If $[\boldsymbol{a} \quad \boldsymbol{x}_i] \sim [\boldsymbol{a}' \quad \boldsymbol{x}'_i]$ for every $i \in [D]$, then $\boldsymbol{X} \sim \boldsymbol{X}'$.*

The same anchor $\boldsymbol{a}$ will be concatenated with all channels forming a series of two-column matrices. Once the permutation orbits of each coupled pair intersect, the permutation orbits of two data matrices also intersect. Our strategy to generate an anchor is through a point-wise linear combination:

**Lemma 4.3** (Anchor Construction). *There exists a set of weights $\boldsymbol{\alpha}_1, \cdots, \boldsymbol{\alpha}_{K_1}$ in general positions of $\mathbb{R}^D$ where $K_1 = N(N-1)(D-1)/2 + 1$ such that for every data matrix $\boldsymbol{X} \in \mathbb{R}^{N \times D}$, there exists $j \in [K_1]$, $\boldsymbol{X}\boldsymbol{\alpha}_j$ is an anchor of $\boldsymbol{X}$.*

From a geometric perspective, if there are enough weights $\{\boldsymbol{\alpha}_j : j \in [K_1]\}$ in general positions, at least one of them will not be orthogonal to the difference between any two columns.

### 4.1.2 INJECTIVITY OF LP

In this section, we specify $\phi$ following the definition in Eq. 2. Suppose sum-of-power mapping $\Psi_N(\boldsymbol{X}\boldsymbol{w}_i) = \Psi_N(\boldsymbol{X}'\boldsymbol{w}_i)$ for all $i \in [K]$, Lemma 2.8 guarantees $\boldsymbol{X}\boldsymbol{w}_i \sim \boldsymbol{X}'\boldsymbol{w}_i$ for all $i \in [K]$. The main technical challenge is to ensure the alignment among all feature columns. This step combines the construction of anchors and the following linear coupling scheme that ensures alignments between all pairwise stackings of feature channels and anchors.

**Lemma 4.4** (Linear Coupling). *There exists a group of coefficients $\gamma_1, \cdots, \gamma_{K_2}$ where $K_2 = N(N-1)+1$ such that the following statement holds: Given any $\boldsymbol{x}, \boldsymbol{x}', \boldsymbol{y}, \boldsymbol{y}' \in \mathbb{R}^N$ such that $\boldsymbol{x} \sim \boldsymbol{x}'$ and $\boldsymbol{y} \sim \boldsymbol{y}'$, if $(\boldsymbol{x} - \gamma_k \boldsymbol{y}) \sim (\boldsymbol{x}' - \gamma_k \boldsymbol{y}')$ for every $k \in [K_2]$, then $[\boldsymbol{x} \quad \boldsymbol{y}] \sim [\boldsymbol{x}' \quad \boldsymbol{y}']$.*

**Construction.** Our construction divides the weights $\{\boldsymbol{w}_i, i \in [K]\}$ into three groups: $\{\boldsymbol{e}_i : i \in [D]\}$, $\{\boldsymbol{\alpha}_j : j \in [K_1]\}$, and $\{\boldsymbol{\Gamma}_{i,j,k} : i \in [D], j \in [K_1], k \in [K_2]\}$. Each block is illustrated in Fig. 3b and outlined as below:

1. Let the first group of weights $\boldsymbol{e}_1, \cdots, \boldsymbol{e}_D \in \mathbb{R}^D$ buffer the original features, where $\boldsymbol{e}_i$ is the $i$-th canonical basis.

2. Design the second group of linear weights, $\boldsymbol{\alpha}_1, \cdots, \boldsymbol{\alpha}_{K_1} \in \mathbb{R}^D$ for $K_1$ as large as $N(N-1)(D-1)/2 + 1$, following the specifications in Lemma 4.3. Then, we know at least one of $\boldsymbol{X}\boldsymbol{\alpha}_j, j \in [K_1]$ forms an anchor of $\boldsymbol{X}$.

3. Design a group of weights $\boldsymbol{\Gamma}_{i,j,k}$ for $i \in [D], j \in [K_1], k \in [K_2]$ with $K_2 = N(N-1)+1$ that mixes each original channel $\boldsymbol{x}_i$ with each $\boldsymbol{X}\boldsymbol{\alpha}_j, j \in [K_1]$ by $\boldsymbol{\Gamma}_{i,j,k} = \boldsymbol{e}_i - \gamma_k \boldsymbol{\alpha}_j$, where $\gamma_k, \forall k \in [K_2]$ is the coefficient defined in Lemma 4.4.

**Injectivity.** With such configuration, injectivity can be shown by the following steps: First recalling the injectivity of power mapping (cf. Lemma 2.8), we have:

$$\sum_{n=1}^N \phi(\boldsymbol{x}^{(n)}) = \sum_{n=1}^N \phi(\boldsymbol{x}'^{(n)}) \Rightarrow \boldsymbol{X}\boldsymbol{w}_i \sim \boldsymbol{X}'\boldsymbol{w}_i, \forall i \in [K]. \tag{4}$$

It is equivalent to expand the RHS of Eq. 4 as: $\boldsymbol{x}_i \sim \boldsymbol{x}'_i$, $\boldsymbol{X}\boldsymbol{\alpha}_j \sim \boldsymbol{X}'\boldsymbol{\alpha}_j$, and $\boldsymbol{X}\boldsymbol{\Gamma}_{i,j,k} = (\boldsymbol{x}_i - \gamma_k \boldsymbol{X}\boldsymbol{\alpha}_j) \sim \boldsymbol{X}'\boldsymbol{\Gamma}_{i,j^*,k} = (\boldsymbol{x}'_i - \gamma_k \boldsymbol{X}'\boldsymbol{\alpha}_j)$ for every $i \in [D], j \in [K_1], k \in [K_2]$. By Lemma 4.4, we can further induce:

$$\boldsymbol{X}\boldsymbol{w}_i \sim \boldsymbol{X}'\boldsymbol{w}_i, \forall i \in [K] \Rightarrow [\boldsymbol{X}\boldsymbol{\alpha}_j \quad \boldsymbol{x}_i] \sim [\boldsymbol{X}'\boldsymbol{\alpha}_j \quad \boldsymbol{x}'_i], \forall i \in [D], j \in [K_1] \tag{5}$$

According to Lemma 4.3, there must be $j^* \in [K_1]$ such that $\boldsymbol{X}\boldsymbol{\alpha}_{j^*}$ is an anchor of $\boldsymbol{X}$. Then by Lemma 4.2, Eq. 5 implies:

$$[\boldsymbol{X}\boldsymbol{\alpha}_{j^*} \quad \boldsymbol{x}_i] \sim [\boldsymbol{X}'\boldsymbol{\alpha}_{j^*} \quad \boldsymbol{x}'_i], \forall i \in [D] \Rightarrow \boldsymbol{X} \sim \boldsymbol{X}'. \tag{6}$$

The total required number of weights $K = D + K_1 + DK_1K_2 \leq N^4D^2$, and the embedding length $L = NK \leq N^5D^2$ as desired.

For completeness, we add the following lemma which implies LP-induced sum-pooling is injective only if $K \geq D + 1$, when $D \geq 2$.

**Theorem 4.5** (Lower Bound). *Consider data matrices $\boldsymbol{X} \in \mathbb{R}^{N \times D}$ where $D \geq 2$. If $K \leq D$, then for every $\boldsymbol{w}_1, \cdots, \boldsymbol{w}_K$, there exists $\boldsymbol{X}' \in \mathbb{R}^{N \times D}$ such that $\boldsymbol{X} \not\sim \boldsymbol{X}'$ but $\boldsymbol{X}\boldsymbol{w}_i \sim \boldsymbol{X}'\boldsymbol{w}_i, \forall i \in [K]$.*

*Remark* 4.6. Theorem 4.5 is significant in that with high-dimensional features, the injectivity is provably not satisfied when the embedding space has a dimension equal to the degree of freedom.

### 4.1.3 INJECTIVITY OF LE

In this section, we consider $\phi$ follows the definition in Eq. 3. Our first observation is that instead of applying univariate monomials to each linearly mixed channel individually, we can directly employ bivariate monomials to pair channels with anchors and yield the same alignment results as in LP.

**Lemma 4.7** (Monomial Coupling). *For any pair of vectors $\boldsymbol{x}, \boldsymbol{y}, \boldsymbol{x}', \boldsymbol{y}' \in \mathbb{R}^N$, if $\sum_{n \in [N]} \boldsymbol{x}_n^{l-k}\boldsymbol{y}_n^k = \sum_{n \in [N]} \boldsymbol{x}'_n^{l-k}\boldsymbol{y}'_n^k$ for every $l \in [N]$, $0 \leq k \leq l$, then $[\boldsymbol{x} \quad \boldsymbol{y}] \sim [\boldsymbol{x}' \quad \boldsymbol{y}']$.*

The second observation is that each term in the RHS of Eq. 3 can be rewritten as a monomial of an exponential function:

$$\exp(\boldsymbol{v}^\top \boldsymbol{x}) = \exp(\boldsymbol{u}^\top \log(\exp(\boldsymbol{\Omega}^\top \boldsymbol{x}))) = \prod_{k=1}^{K_1+D} \exp(\boldsymbol{\Omega}\boldsymbol{x})_k^{\boldsymbol{u}_k}, \tag{7}$$

where the exponential and the logarithm are taken element-wisely, $\boldsymbol{v} = \boldsymbol{\Omega}\boldsymbol{u}$ for some $\boldsymbol{\Omega} \in \mathbb{R}^{D \times (K_1+D)}$, and $\boldsymbol{u} \in \mathbb{R}^{K_1+D}$. Recall that $K_1$ is the needed dimension to construct an anchor as shown in Lemma 4.3. Then, the assignment of $\boldsymbol{v}_1, \cdots, \boldsymbol{v}_L$ amounts to specifying the exponents for $D$ power functions within the product. Specifically, we choose $\boldsymbol{v}, \boldsymbol{\Omega}, \boldsymbol{u}$ as follows.

**Construction.** We first reindex and rewrite $\{\boldsymbol{v}_i : i \in [L]\}$ as $\{\boldsymbol{v}_{i,j,p,q} = \boldsymbol{\Omega}\boldsymbol{u}_{i,j,p,q} : i \in [D], j \in [K_1], p \in [N], q \in [p+1]\}$, where $\boldsymbol{\Omega} = [\boldsymbol{e}_1 \quad \cdots \quad \boldsymbol{e}_D \quad \boldsymbol{\alpha}_1 \quad \cdots \quad \boldsymbol{\alpha}_{K_1}] \in \mathbb{R}^{D \times (D+K_1)}$ and $\boldsymbol{u}_{i,j,p,q} \in \mathbb{R}^{D+K_1}$ are specified as below. In Fig. 3c, we depict the forward pass of an LE layer.

1. The choice of weights $\boldsymbol{\Omega}$ follows from the construction of the LP layer, i.e., $\boldsymbol{e}_i \in \mathbb{R}^D, \forall i \in [D]$ are canonical basis and $\{\boldsymbol{\alpha}_j : \forall j \in [K_1]\}$ with $K_1 = N(N-1)(D-1)/2 + 1$ are drawn according to Lemma 4.3 so that an anchor is guaranteed to be produced.

2. Design a group of weights $\boldsymbol{U} = [\cdots \quad \boldsymbol{u}_{i,j,p,q} \quad \cdots] \in \mathbb{R}^{(D+K_1) \times DK_1N(N+3)/2}$ for $i \in [D], j \in [K_1], p \in [N], q \in [p+1]$ such that $\boldsymbol{u}_{i,j,p,q} = (q-1)\boldsymbol{e}_i + (p-q+1)\boldsymbol{e}_{D+j}$.

**Injectivity.** Plugging $\boldsymbol{\Omega}$ and $\boldsymbol{U}$ into Eq. 7, we can examine each output dimension of the embedding layer: $\left[\sum_{n=1}^N \phi(\boldsymbol{x}^{(n)})\right]_{i,j,p,q} = \sum_{n=1}^N \exp(\boldsymbol{x}_i)_n^{q-1} \exp(\boldsymbol{X}\boldsymbol{\alpha}_j)_n^{p-q+1}$. Then by Lemma 4.7:

$$\sum_{n=1}^N \phi(\boldsymbol{x}^{(n)}) = \sum_{n=1}^N \phi(\boldsymbol{x}'^{(n)}) \Rightarrow [\exp(\boldsymbol{x}_i) \quad \exp(\boldsymbol{X}\boldsymbol{\alpha}_j)] \sim [\exp(\boldsymbol{x}'_i) \quad \exp(\boldsymbol{X}'\boldsymbol{\alpha}_j)], \tag{8}$$

$\forall i \in [D], j \in [K_2]$. With above implication, the proof can be concluded via the following steps: Lemma 4.3 guarantees the existence of $j^* \in [K_2]$ such that $\boldsymbol{X}\boldsymbol{w}_{j^*}$ is an anchor of $\boldsymbol{X}$, and so is $\exp(\boldsymbol{X}\boldsymbol{w}_{j^*})$ due to the strict monotonicity of $\exp(\cdot)$; Lemma 4.2 and Eq. 8 together imply:

$$[\exp(\boldsymbol{x}_i) \quad \exp(\boldsymbol{X}\boldsymbol{\alpha}_{j^*})] \sim [\exp(\boldsymbol{x}'_i) \quad \exp(\boldsymbol{X}'\boldsymbol{\alpha}_{j^*})] \, \forall i \in [D] \Rightarrow \exp(\boldsymbol{X}) \sim \exp(\boldsymbol{X}'). \tag{9}$$

And finally, notice that an element-wise function does not affect equivalence under permutation. The total number of required linear weights is $L = DK_1(N+3)N/2 \leq N^4D^2$, as desired.

## 4.2 CONTINUITY

In this section, we show that the LP and LE induced sum-pooling are both homeomorphic. We note that it is intractable to obtain the closed form of their inverse maps. Notably, the following remarkable result can get rid of inversing a function explicitly by merely examining the topological relationship between the domain and image space.

**Lemma 4.8.** *(Theorem 1.2 (Ćurgus & Mascioni, 2006)) Let $(\mathcal{X}, d_{\mathcal{X}})$ and $(\mathcal{Z}, d_{\mathcal{Z}})$ be two metric spaces and $f : \mathcal{X} \to \mathcal{Z}$ is a bijection such that **(a)** each bounded and closed subset of $\mathcal{X}$ is compact, **(b)** $f$ is continuous, **(c)** $f^{-1}$ maps each bounded set in $\mathcal{Z}$ into a bounded set in $\mathcal{X}$. Then $f^{-1}$ is continuous.*

Subsequently, we show the continuity in an informal but more intuitive way while deferring a rigorous version to the supplementary materials. Denote $\Phi(\boldsymbol{X}) = \sum_{i \in [N]} \phi(\boldsymbol{x}^{(i)})$. To begin with, we set $\mathcal{X} = \mathbb{R}^{N \times D} / \sim$ with metric $d_{\mathcal{X}}(\boldsymbol{X}, \boldsymbol{X}') = \min_{\boldsymbol{P} \in \Pi(N)} \|\boldsymbol{X} - \boldsymbol{P}\boldsymbol{X}'\|_{\infty,\infty}$ and $\mathcal{Z} = \{\Phi(\boldsymbol{X}) | \boldsymbol{X} \in \mathcal{X}\} \subseteq \mathbb{R}^L$ with metric $d_{\mathcal{Z}}(\boldsymbol{z}, \boldsymbol{z}') = \|\boldsymbol{z} - \boldsymbol{z}'\|_{\infty}$. It is easy to show that $\mathcal{X}$ satisfies the conditions **(a)** and $\Phi(\boldsymbol{X})$ satisfies **(b)** for both LP and LE embedding layers. Then it remains to conclude the proof by verifying the condition **(c)** for the mapping $\mathcal{Z} \to \mathcal{X}$, i.e., the inverse of $\Phi(\boldsymbol{X})$. We visualize this mapping following the chain of implication to show injectivity:

$$
\begin{aligned}
(LP) \quad & \Phi(\boldsymbol{X}) \xrightarrow{\text{Eq. 4}} \left[\cdots \quad \boldsymbol{P}_i \boldsymbol{X} \boldsymbol{w}_i \quad \cdots\right], i \in [K] \xrightarrow{\text{Eqs. } 5+6} \boldsymbol{P}\boldsymbol{X} \\
(LE) \quad & \underbrace{\Phi(\boldsymbol{X})}_{\mathcal{Z}} \xrightarrow{\text{Eq. 8}} \underbrace{\exp\left[\boldsymbol{Q}_{i,j}\boldsymbol{x}_i \quad \boldsymbol{Q}_{i,j}\boldsymbol{a}_j\right], i \in [D], j \in [K_1]}_{\mathcal{R}} \xrightarrow{\text{Eq. 9}} \underbrace{\boldsymbol{Q}\boldsymbol{X}}_{\mathcal{X}} ,
\end{aligned}
$$

for some $\boldsymbol{X}$ dependent $\boldsymbol{P}, \boldsymbol{Q} \in \Pi(N)$. Here, $\boldsymbol{P}_i \in \Pi(N), i \in [K]$ and $\boldsymbol{Q}_{i,j} \in \Pi(N)$, $\boldsymbol{a}_j = \boldsymbol{X}\boldsymbol{\alpha}_j, i \in [D], j \in [K_1]$. All the weights have been specified as in Sec. 4.1. According to homeomorphism between polynomial coefficients and roots (Corollary 3.2 in Ćurgus & Mascioni (2006)), any bounded set in $\mathcal{Z}$ will be mapped into a bounded set in $\mathcal{R}$. Moreover, since elements in $\mathcal{R}$ contain all the columns of $\mathcal{X}$ (up to some changes of the entry orders), a bounded set in $\mathcal{R}$ also corresponds to a bounded set in $\mathcal{X}$. Through this line of arguments, we conclude the proof.

## 5 EXTENSIONS

**Permutation Equivariance.** Permutation-equivariant functions (cf. Definition 2.3) are considered as a more general family of set functions. Our main result does not lose generality to this class of functions. By Lemma 2 of Wang et al. (2023), Theorem 3.1 can be directly extended to permutation-equivariant functions with *the same lower and upper bounds*, stated as follows:

**Theorem 5.1** (Extension to Equivariance). *For any permutation-equivariant function $f : \mathbb{R}^{N \times D} \to \mathbb{R}^N$, there exists continuous functions $\phi : \mathbb{R}^D \to \mathbb{R}^L$ and $\rho : \mathbb{R}^D \times \mathbb{R}^L \to \mathbb{R}$ such that $f(\boldsymbol{X})_j = \rho\left(\boldsymbol{x}^{(j)}, \sum_{i \in [N]} \phi(\boldsymbol{x}^{(i)})\right)$ for every $j \in [N]$, where $L \in [N(D+1), N^5 D^2]$ when $\phi$ admits LP architecture, and $L \in [ND, N^4 D^2]$ when $\phi$ admits LE architecture.*

**Complex Domain.** The upper bounds in Theorem 3.1 is also true to complex features up to a constant scale. When features are defined over $\mathbb{C}^{N \times D}$, our primary idea is to divide each channel into two real feature vectors, and recall Theorem 3.1 to conclude the arguments on an $\mathbb{R}^{N \times 2D}$ input. All of our proof strategies are still applied. This result directly contrasts to Zweig & Bruna's work whose main arguments were established on complex numbers. We show that even moving to the complex domain, polynomial length of $L$ is still sufficient for the DeepSets architecture (Zaheer et al., 2017). We state a formal version of the theorem in Appendix I.

## 6 CONCLUSION

This work investigates how many neurons are needed to model the embedding space for set representation learning with the DeepSets architecture (Zaheer et al., 2017). Our paper provides an affirmative answer that polynomial many neurons in the set size and feature dimension are sufficient. Compared with prior arts, our theory takes high-dimensional features into consideration while significantly advancing the state-of-the-art results from exponential to polynomial.

**Limitations.** The tightness of our bounds is not examined in this paper, and the complexity of $\rho$ is uninvestigated and left for future exploration.

ACKNOWLEDGMENTS

We would like to thank Dr. Yusu Wang and Dr. Puoya Tabaghi for a meaningful discussion. We also express our gratitude to Dr. Manolis C. Tsakiris for pointing out useful results in the topics of unlabeled sensing. P. Li is supported by NSF awards PHY-2117997, IIS-2239565. Z. Wang is in part supported by US Army Research Office Young Investigator Award W911NF2010240 and the NSF AI Institute for Foundations of Machine Learning (IFML).

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

## A  NUMERICAL EXPERIMENTS

To verify our theoretical claim, we conducted proof-of-concept experiments. Similar to Wagstaff et al. (2019), we train a DeepSets with $\phi$ and $\rho$ parameterized by neural networks to fit a function that takes the *median* over a vector-valued set according to the lexicographical order. Specifically, the input features are sampled from a uniform distribution, $\phi$ is chosen as one linear layer followed by a SiLU activation function (Elfwing et al., 2018), and $\rho$ is a two-layer fully-connected network with ReLU activation. During the experiment, we vary the input size, dimension, and hidden dimension of $\phi$, and record the final training error (RMSE) after the network converges. The critical width $L^*$ is taken at the point where RMSE first reaches below 10% above the minimum value for this set size. The relationship between $L^*$ and $N, D$ is plotted in Fig. 2. We observe $\log(L^*)$ grows linearly with $\log(N)$ and $\log(D)$ instead of exponentially, which validates our theoretical claim.

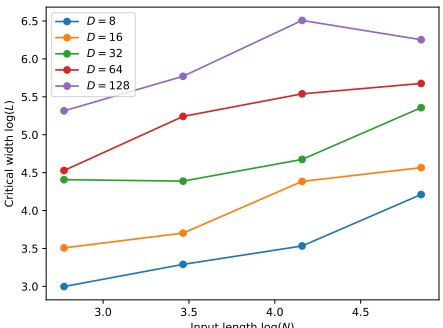
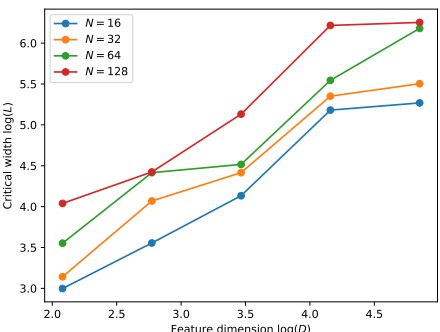

Figure 2: The relationship among the critical width $L$, set size $N$, and feature dimension $D$. The phenomenon that $\log(L)$ scales linearly with $\log(N)$ and $\log(D)$ validates our theory.

## B  OTHER RELATED WORK

Most works on neural networks to represent set functions have been discussed extensively in Sec. 1 and 3. Here, we highlight a few concurrent works. One breakthrough by Dym & Gortler (2024); Amir et al. (2024) provides both upper bound $2ND + 1$ and lower bound $N(D + 1)$ on the embedding dimension for set representation. In their construction, $\phi$ is chosen as a linear transformation followed by an arbitrary elementwise analytic function. Their proof of moment injectivity is through extending the finite-witness theorem to $\sigma$-subanalytic functions defined over $\sigma$-subanalytic sets. Despite generalization to a broader scope of invariant structures, their approach fails to show the continuity of $\rho$ when the targeted function is defined over an open set. Tabaghi & Wang (2023) has recently improved the upper bound of $L$ from $2ND + 1$ to $2ND$ considering the input feature space is compact. Whereas, our main claim (Theorem 3.1) guarantees the exact representation over the entire ambient space. The difficulty of mirroring our proof via Lemma 4.8 perhaps arises from the lack of explicit form of $\phi$ - verifying the boundedness of $\Phi^{-1}$ becomes less tractable. Complementary to this line of work on symmetric functions, Chen & Lu (2023) demonstrates polynomial reliance on $N$ to exactly represent anti-symmetric functions.

We also review other related works on the expressive power analysis of neural networks. Early works studied the expressive power of feed-forward neural networks with different activations (Hornik et al., 1989; Cybenko, 1989). Recent works focused on characterizing the benefits of the expressive power of deep architectures to explain their empirical success (Yarotsky, 2017; Liang & Srikant, 2017; Kileel et al., 2019; Cohen et al., 2016; Raghu et al., 2017). Modern neural networks often enforce some invariance properties into their architectures such as CNNs that capture spatial translation invariance. The expressive power of invariant neural networks has been analyzed recently in Yarotsky (2022); Maron et al. (2019); Zhou (2020).

The architectures studied in the above works allow universal approximation of continuous functions defined on their inputs. However, the family of practically useful architectures that enforce permutation invariance often fail in achieving universal approximation. Graph Neural Networks (GNNs) enforce permutation invariance and can be viewed as an extension of set neural networks to encode a set of pair-wise relations instead of a set of individual elements (Scarselli et al., 2008; Gilmer et al., 2017; Kipf & Welling, 2017; Hamilton et al., 2017). GNNs suffer from limited expressive power (Xu et al., 2019; Morris et al., 2019; Maron et al., 2018) unless they adopt exponential-order tensors (Keriven & Peyré, 2019). Hence, previous studies often characterized GNNs' expressive power based on their capability of distinguishing non-isomorphic graphs. Only a few works have ever discussed the function approximation property of GNNs (Chen et al., 2019; 2020; Azizian & Lelarge, 2021) while these works still miss characterizing such dependence on the depth and width of the architectures (Loukas, 2020). As practical GNNs commonly adopt the architectures that combine feed-forward neural networks with set operations (neighborhood aggregation), we believe the characterization of the needed size for set function approximation studied in Zweig & Bruna (2022) and this work may provide useful tools to study finer-grained characterizations of the expressive power of GNNs.

## C  SOME PRELIMINARY DEFINITIONS AND STATEMENTS

In this section, we begin by stating basic metric spaces, which will be used in the proofs later.

**Definition C.1** (Standard Metric). Define $(\mathcal{K}^D, d_\infty)$ as the standard metric space, where $d_\infty : \mathcal{K}^D \times \mathcal{K}^D \to \mathbb{R}_{\geq 0}$ is the $\ell_\infty$-norm induced distance metric over $\mathcal{K}^D$:

$$d_\infty(\boldsymbol{z}, \boldsymbol{z}') = \max_{i \in [D]} |\boldsymbol{z}_i - \boldsymbol{z}'_i|. \tag{10}$$

**Definition C.2** (Product Metric). Consider a metric space $(\mathcal{X}, d_\mathcal{X})$. We denote the induced product metric over the product space $(\mathcal{X}^K, d_\mathcal{X}^K)$ as $d_\mathcal{X}^K : \mathcal{X}^K \times \mathcal{X}^K \to \mathbb{R}_{\geq 0}$:

$$d_\mathcal{X}^K(\boldsymbol{Z}, \boldsymbol{Z}') = \max_{i \in [K]} d_\mathcal{X}(\boldsymbol{z}_i, \boldsymbol{z}'_i), \tag{11}$$

where $\boldsymbol{Z} = [\boldsymbol{z}_1 \quad \cdots \quad \boldsymbol{z}_K] \in \mathcal{X}^K, \boldsymbol{z}_i \in \mathcal{X}, \forall i \in [K]$.

We provide rigorous definitions to specify the topology of the input space of permutation-invariant functions.

**Definition C.3** (Set Metric). Equipped $\mathcal{K}^{N \times D}$ with the equivalence relation $\sim$ (cf. Definition 2.1), we define metric space $(\mathcal{K}^{N \times D}/ \sim, d_\Pi)$, where $d_\Pi : (\mathcal{K}^{N \times D}/ \sim) \times (\mathcal{K}^{N \times D}/ \sim) \to \mathbb{R}_{\geq 0}$ is the optimal transport distance:

$$d_\Pi(\boldsymbol{X}, \boldsymbol{X}') = \min_{\boldsymbol{P} \in \Pi(N)} \|\boldsymbol{P}\boldsymbol{X} - \boldsymbol{X}'\|_{\infty,\infty}, \tag{12}$$

and $\mathcal{K}$ can be either $\mathbb{R}$ or $\mathbb{C}$.

*Remark* C.4. The $\|\cdot\|_{\infty,\infty}$ norm takes the absolute value of the maximal entry: $\max_{i \in [N], j \in [D]} |\boldsymbol{X}_{i,j}|$. Other topologically equivalent matrix norms also apply.

**Lemma C.5.** *The function* $d_\Pi : (\mathcal{K}^{N \times D}/ \sim) \times (\mathcal{K}^{N \times D}/ \sim) \to \mathbb{R}_{\geq 0}$ *is a distance metric on* $\mathcal{K}^{N \times D}/ \sim$.

*Proof.* Identity, positivity, and symmetry trivially hold for $d_\Pi$. It remains to show the triangle inequality as below: for arbitrary $\boldsymbol{X}, \boldsymbol{X}', \boldsymbol{X}'' \in (\mathcal{K}^{N \times D}/ \sim, d_\Pi)$,

$$\begin{aligned}
d_\Pi(\boldsymbol{X}, \boldsymbol{X}'') &= \min_{\boldsymbol{P} \in \Pi(N)} \|\boldsymbol{P}\boldsymbol{X} - \boldsymbol{X}''\|_{\infty,\infty} \leq \min_{\boldsymbol{P} \in \Pi(N)} \left( \|\boldsymbol{P}\boldsymbol{X} - \boldsymbol{Q}^*\boldsymbol{X}'\|_{\infty,\infty} + \|\boldsymbol{Q}^*\boldsymbol{X}' - \boldsymbol{X}''\|_{\infty,\infty} \right) \\
&= \min_{\boldsymbol{P} \in \Pi(N)} \|\boldsymbol{P}\boldsymbol{X} - \boldsymbol{Q}^*\boldsymbol{X}'\|_{\infty,\infty} + \|\boldsymbol{Q}^*\boldsymbol{X}' - \boldsymbol{X}''\|_{\infty,\infty} \\
&= d_\Pi(\boldsymbol{X}, \boldsymbol{X}') + d_\Pi(\boldsymbol{X}, \boldsymbol{X}''),
\end{aligned}$$

where $\boldsymbol{Q}^* \in \arg\min_{\boldsymbol{Q} \in \Pi(N)} \|\boldsymbol{Q}\boldsymbol{X}' - \boldsymbol{X}''\|_{\infty,\infty}$. $\qquad \square$

Also we reveal a topological property for $(\mathcal{K}^{N \times D}/ \sim, d_\Pi)$ which is essential to show continuity later.

**Lemma C.6.** *Each bounded and closed subset of* $(\mathcal{K}^{N \times D}/ \sim, d_\Pi)$ *is compact.*

*Proof.* Without loss of generality, the proof is done by extending Theorem 2.4 in Ćurgus & Mascioni (2006) to high-dimensional set elements. Let $\varrho : (\mathcal{K}^{N \times D}, d_\infty) \to (\mathcal{K}^{N \times D}/ \sim, d_\Pi)$ maps a matrix $\boldsymbol{X}$ to a set whose elements are the rows of $\boldsymbol{X}$. We notice that $\varrho$ is a continuous mapping due to its contraction nature: $d_\Pi(\varrho(\boldsymbol{X}), \varrho(\boldsymbol{X}')) \leq \|\boldsymbol{X} - \boldsymbol{X}'\|_{\infty,\infty} = d_\infty(\boldsymbol{X}, \boldsymbol{X}')$. Let $\varrho^{-1}(\boldsymbol{Z}) = \{\boldsymbol{X} \in (\mathcal{K}^{N \times D}, d_\infty) : \varrho(\boldsymbol{X}) \sim \boldsymbol{Z}\}$. Define $\mathcal{T} \subset (\mathcal{K}^{N \times D}, d_\infty)$ such that $\varrho^{-1}(\boldsymbol{Z}) \cap \mathcal{T}$ has only one element for every $\boldsymbol{Z} \in (\mathcal{K}^{N \times D}/ \sim, d_\Pi)$. One example of picking elements for $\mathcal{T}$ is to sort every $\boldsymbol{Z} \in (\mathcal{K}^{N \times D}/ \sim, d_\Pi)$ in the lexicographical order. Now constraining the domain of $\varrho$ to be $\mathcal{T}$ yields a bijective mapping $\varrho|_\mathcal{T}$, and its inverse $\varrho|_\mathcal{T}^{-1}$. We notice that $\varrho \circ \varrho|_\mathcal{T}^{-1}$ induces an identity mapping over $(\mathcal{K}^{N \times D}/ \sim, d_\Pi)$.

Now consider an arbitrary closed and bounded subset $\mathcal{S} \subset (\mathcal{K}^{N \times D}/ \sim, d_\Pi)$. To show $\mathcal{S}$ is compact, we demonstrate every sequence $\{\boldsymbol{X}_k\}$ in $\mathcal{S}$ has a convergent subsequence. First observe that $\{\varrho|_\mathcal{T}^{-1}(\boldsymbol{X}_k)\}$ is also bounded in $(\mathcal{K}^{N \times D}, d_\infty)$. This is because $\|\varrho|_\mathcal{T}^{-1}(\boldsymbol{X})\|_{\infty,\infty} = d_\Pi(\boldsymbol{X}, \boldsymbol{0})$. Hence, by Bolzano-Weierstrass Theorem, there exists a subsequence $\{\boldsymbol{X}_{j_k}\} \subset \{\boldsymbol{X}_k\}$ such that $\{\varrho|_\mathcal{T}^{-1}(\boldsymbol{X}_{j_k})\}$ converges to some $\boldsymbol{\chi} \in (\mathcal{K}^{N \times D}, d_\infty)$. As aforementioned, since $\varrho$ is a continuous mapping, $\{\varrho \circ \varrho|_\mathcal{T}^{-1}(\boldsymbol{X})\}$ converges to $\varrho(\boldsymbol{\chi})$ in $(\mathcal{K}^{N \times D}/ \sim, d_\Pi)$. Since $\mathcal{S}$ is closed, $\varrho(\boldsymbol{\chi}) \in \mathcal{S}$. This is $\lim_{k \to \infty} \boldsymbol{X}_{j_k} = \varrho(\boldsymbol{\chi})$, which concludes the proof. $\qquad \square$

Then we can rephrase the definition of a permutation-invariant function as a proper function mapping between the two metric spaces: $f : (\mathcal{K}^{N \times D}/ \sim, d_\Pi) \to (\mathcal{K}^{D'}, d_\infty)$.

We also recall the definition of injectivity for permutation-invariant functions:

**Definition C.7** (Injectivity). A permutation-invariant function $f : (\mathcal{K}^{N \times D}/ \sim, d_\Pi) \to (\mathcal{K}^{D'}, d_\infty)$ is injective if for every $\boldsymbol{X}, \boldsymbol{X}' \in \mathcal{K}^{N \times D}$ such that $f(\boldsymbol{X}) = f(\boldsymbol{X}')$, then $\boldsymbol{X} \sim \boldsymbol{X}'$.

**Definition C.8** (Bijectivity/Invertibility). A permutation-invariant function $f : (\mathcal{K}^{N \times D}/ \sim, d_\Pi) \to (\mathcal{K}^{D'}, d_\infty)$ is bijective or invertible if there exists a function $g : (\mathcal{K}^{D'}, d_\infty) \to (\mathcal{K}^{N \times D}/ \sim, d_\Pi)$ such that for every $\boldsymbol{X} \in \mathcal{K}^{N \times D}, g \circ f(\boldsymbol{X}) \sim \boldsymbol{X}$.

A well-known and useful result in set theory connects injectivity and bijectivity:

**Lemma C.9.** *A function is bijective if and only if it is simultaneously injective and surjective.*

We give an intuitive definition of continuity for permutation-invariant functions via the epsilon-delta statement:

**Definition C.10** (Continuity). A permutation-invariant function $f : (\mathcal{K}^{N \times D}/\sim, d_\Pi) \to (\mathcal{K}, d_\infty)$ is continuous if for arbitrary $\boldsymbol{X} \in \mathcal{K}^{N \times D}$ and $\epsilon > 0$, there exists $\delta > 0$ such that for every $\boldsymbol{X}' \in \mathcal{K}^{N \times D}$, $d_\Pi(\boldsymbol{X}, \boldsymbol{X}') < \delta$ then $d_\infty(f(\boldsymbol{X}), f(\boldsymbol{X}')) < \epsilon$.

*Remark* C.11. Since $d_\Pi$ is a distance metric, other equivalent definitions of continuity still applies.

# D  PROPERTIES OF SUM-OF-POWER MAPPING FOR REAL AND COMPLEX DOMAINS

In this section, we extend the sum-of-power mapping to both real and complex domains, and explore their desirable properties that serve as prerequisites for our later proof. The proof techniques are borrowed from Ćurgus & Mascioni (2006). Below, $\mathcal{K}$ can be either $\mathbb{R}$ or $\mathbb{C}$.

**Definition D.1.** Define power mapping: $\psi_N : \mathcal{K} \to \mathcal{K}^N$, $\psi_N(z) = \begin{bmatrix} z & z^2 & \cdots & z^N \end{bmatrix}^\top$ and (complex) sum-of-power mapping $\Psi_N : (\mathcal{K}^N/\sim, d_\Pi) \to (\mathcal{K}^N, d_\infty)$, $\Psi_N(\boldsymbol{z}) = \sum_{i=1}^N \psi_N(z_i)$.

**Lemma D.2** (Existence of Continuous Inverse of Complex Sum-of-Power (Ćurgus & Mascioni, 2006)). *$\Psi_N$ is injective, thus the inverse $\Psi_N^{-1} : (\mathcal{K}^N, d_\infty) \to (\mathcal{K}^N/\sim, d_\Pi)$ exists. Moreover, $\Psi_N^{-1}$ is continuous.*

**Lemma D.3** (Corollary 3.2 (Ćurgus & Mascioni, 2006)). *Consider a function $\zeta : (\mathcal{K}^N, d_\infty) \to (\mathcal{K}^N/\sim, d_\Pi)$ that maps the coefficients of a polynomial to its root multi-set. Then for any bounded subset $\mathcal{U} \subset (\mathcal{K}^N, d_\infty)$, the image $\zeta(\mathcal{U}) = \{\zeta(\boldsymbol{z}) : \boldsymbol{z} \in \mathcal{U}\}$ is also bounded.*

*Remark* D.4. The original proof of Lemma D.3 is done for the complex domain. However, it is naturally true for real numbers because we can constrain the domain of $\zeta$ to be real coefficients such that the corresponding polynomials can fully split over the real domain, and the image to be all the real roots. Then both domain and image turn out to be a subset of the complex-valued version.

**Lemma D.5.** *Consider the $N$-degree sum-of-power mapping: $\Psi_N : (\mathcal{K}^N/\sim, d_\Pi) \to (\mathcal{K}^N, d_\infty)$, where $\Psi_N(\boldsymbol{x}) = \sum_{i=1}^N \psi_N(x_i)$. Denote the range of $\Psi_N$ as $\mathcal{Z}_{\Psi_N} \subseteq \mathcal{K}^N$ and its inverse mapping $\Psi_N^{-1} : (\mathcal{Z}_{\Psi_N}, d_\infty) \to (\mathcal{K}^N/\sim, d_\Pi)$ (existence guaranteed by Lemma D.2). Then for every bounded set $\mathcal{U} \subset (\mathcal{Z}_{\Psi_N}, d_\infty)$, the image $\Psi_N^{-1}(\mathcal{U}) = \{\Psi_N^{-1}(\boldsymbol{z}) : \boldsymbol{z} \in \mathcal{U}\}$ is also bounded.*

*Proof.* We first show this result when $\mathcal{K} = \mathbb{C}$, and naturally extend it to $\mathcal{K} = \mathbb{R}$. We borrow the proof technique from Zaheer et al. (2017) to reveal a polynomial mapping between $(\mathcal{Z}_{\Psi_N}, d_\infty)$ and coefficient space of complex polynomials $(\mathbb{C}^N, d_\infty)$. For every $\boldsymbol{\xi} \in (\mathbb{C}^N/\sim, d_\Pi)$, let $\boldsymbol{z} = \Psi_N(\boldsymbol{\xi})$ and construct a polynomial:

$$P_{\boldsymbol{\xi}}(x) = \prod_{i=1}^N (x - \xi_i) = x^N - a_1 x^{N-1} + \cdots + (-1)^{N-1} a_{N-1} x + (-1)^N a_N, \qquad (13)$$

where $\boldsymbol{\xi}$ are the roots of $P_{\boldsymbol{\xi}}(x)$ and the coefficients can be written as elementary symmetric polynomials, i.e.,

$$a_n = \sum_{1 \le j_1 \le j_2 \le \cdots \le j_n \le N} \xi_{j_1} \xi_{j_2} \cdots \xi_{j_n}, \forall n \in [N]. \qquad (14)$$

On the other hand, the elementary symmetric polynomials can be uniquely expressed as a function of $\boldsymbol{z}$ by Newton-Girard formula:

$$a_n = \frac{1}{n} \det \begin{bmatrix} z_1 & 1 & 0 & 0 & \cdots & 0 \\ z_2 & z_1 & 1 & 0 & \cdots & 0 \\ \vdots & \vdots & \vdots & \vdots & \ddots & \vdots \\ z_{n-1} & z_{n-2} & z_{n-3} & z_{n-4} & \cdots & 1 \\ z_n & z_{n-1} & z_{n-2} & z_{n-3} & \cdots & 1 \end{bmatrix} := Q(\boldsymbol{z}), \forall n \in [N] \qquad (15)$$

where the determinant $Q(z)$ is also a polynomial in $z$. Now we establish the mapping between $(\mathcal{Z}_{\Psi_N}, d_\infty)$ and $(\mathbb{C}^N/\sim, d_\Pi)$:

$$(\mathcal{Z}_{\Psi_N}, d_\infty) \xrightarrow{Q(z)} \underbrace{(\mathbb{C}^N, d_\infty)}_{\text{Coefficients}} \xrightarrow{\text{Lemma D.3}} \underbrace{(\mathbb{C}^N/\sim, d_\Pi)}_{\text{Roots}}. \tag{16}$$

Then the proof proceeds by observing that for any bounded subset $\mathcal{U} \subseteq (\mathcal{Z}_{\Psi_N}, d_\infty)$, the resulting $\mathcal{A} = Q(\mathcal{U})$ is also bounded in $(\mathbb{C}^N, d_\infty)$. Therefore, by Lemma D.3, any bounded coefficient set $\mathcal{A}$ will produce a bounded root multi-set in $(\mathbb{C}^N/\sim, d_\Pi)$.

Now we show Lemma D.3 is also true for real numbers. By Remark D.4, we can constrain the ambient space of $\mathcal{A}$ to be real coefficients whose corresponding polynomials can split over real numbers, and then the same proof proceed. $\square$

**Corollary D.6.** *Consider channel-wise high-dimensional sum-of-power* $\widehat{\Psi_N}(\boldsymbol{X}) : (\mathcal{K}^{N \times K}/\sim, d_\Pi) \to (\mathcal{Z}_{\Psi_N}^K, d_\infty)$ *defined as below:*

$$\widehat{\Psi_N}(\boldsymbol{X}) = \begin{bmatrix} \Psi_N(\boldsymbol{x}_1)^\top & \cdots & \Psi_N(\boldsymbol{x}_K)^\top \end{bmatrix}^\top \in (\mathcal{Z}_{\Psi_N}^K, d_\infty), \tag{17}$$

*where* $\mathcal{Z}_{\Psi_N} = \{\Psi_N(\boldsymbol{x}) : \boldsymbol{x} \in \mathcal{K}^N\} \subseteq \mathcal{K}^N$ *is the range of the sum-of-power mapping. Define an associated mapping* $\widehat{\Psi_N}^\dagger : (\mathcal{Z}_{\Psi_N}^K, d_\infty) \to (\mathcal{K}^N/\sim, d_\Pi)^K$:

$$\widehat{\Psi_N}^\dagger(\boldsymbol{Z}) = \begin{bmatrix} \Psi_N^{-1}(\boldsymbol{z}_1) & \cdots & \Psi_N^{-1}(\boldsymbol{z}_K) \end{bmatrix}, \tag{18}$$

*where* $\boldsymbol{Z} = \begin{bmatrix} \boldsymbol{z}_1^\top & \cdots & \boldsymbol{z}_K^\top \end{bmatrix}^\top$, $\boldsymbol{z}_i \in \mathcal{Z}_{\Psi_N}, \forall i \in [K]$. *Then the mapping* $\widehat{\Psi_N}^\dagger$ *maps any bounded set in* $(\mathcal{Z}_{\Psi_N}^K, d_\infty)$ *to a bounded set in* $(\mathcal{K}^N/\sim, d_\Pi)^K$.

*Proof.* Proved by noting that if $d_\infty(\boldsymbol{z}_i, \boldsymbol{z}'_i) \le C_1$ for some $\boldsymbol{z}_i, \boldsymbol{z}'_i \in (\mathcal{Z}_{\Psi_N}, d_\infty), \forall i \in [K]$ and a constant $C_1 \ge 0$, then $d_\Pi(\Psi_N^{-1}(\boldsymbol{z}_i), \Psi_N^{-1}(\boldsymbol{z}'_i)) \le C_2, \forall i \in [K]$ for some constant $C_2 \ge 0$ by Lemma D.5. Finally, we have:

$$d_\Pi^K\left(\widehat{\Psi_N}^\dagger(\boldsymbol{Z}), \widehat{\Psi_N}^\dagger(\boldsymbol{Z}')\right) = \max_{i \in [K]} d_\Pi(\Psi_N^{-1}(\boldsymbol{z}_i), \Psi_N^{-1}(\boldsymbol{z}'_i)) \le C_2,$$

which is also bounded above. $\square$

## E  PROOFS FOR THE PROPERTIES OF ANCHOR

The main ingredient of our construction is anchor defined in Definition 4.1. Two key properties of anchors are restated in Lemma E.1 and E.2 and proved below:

**Lemma E.1.** *Consider the data matrix* $\boldsymbol{X} \in \mathbb{R}^{N \times D}$ *and* $\boldsymbol{a} \in \mathbb{R}^N$ *an anchor of* $\boldsymbol{X}$. *Then if there exists* $\boldsymbol{P} \in \Pi(N)$ *such that* $\boldsymbol{P}\boldsymbol{a} = \boldsymbol{a}$ *then* $\boldsymbol{P}\boldsymbol{x}_i = \boldsymbol{x}_i$ *for every* $i \in [D]$.

*Proof.* Prove by contradiction. Suppose $\boldsymbol{P}\boldsymbol{x}_i \ne \boldsymbol{x}_i$ for some $i \in [D]$, then there exist some $p, q \in [N]$ such that $\boldsymbol{x}_i^{(p)} \ne \boldsymbol{x}_i^{(q)}$ while $a_p = a_q$. However, this contradicts the definition of an anchor (cf. Definition 4.1). $\square$

**Lemma E.2** (Union Alignment, Lemma 4.2). *Consider the data matrix* $\boldsymbol{X}, \boldsymbol{X}' \in \mathbb{R}^{N \times D}$, $\boldsymbol{a} \in \mathbb{R}^N$ *is an anchor of* $\boldsymbol{X}$ *and* $\boldsymbol{a}' \in \mathbb{R}^N$ *is an arbitrary vector. If* $\begin{bmatrix} \boldsymbol{a} & \boldsymbol{x}_i \end{bmatrix} \sim \begin{bmatrix} \boldsymbol{a}' & \boldsymbol{x}'_i \end{bmatrix}$ *for every* $i \in [D]$, *then* $\boldsymbol{X} \sim \boldsymbol{X}'$.

*Proof.* According to definition of equivalence, there exists $\boldsymbol{Q}_i \in \Pi(N)$ for every $i \in [D]$ such that $\begin{bmatrix} \boldsymbol{a} & \boldsymbol{x}_i \end{bmatrix} = \begin{bmatrix} \boldsymbol{Q}_i\boldsymbol{a}' & \boldsymbol{Q}_i\boldsymbol{x}'_i \end{bmatrix}$. Moreover, since $\begin{bmatrix} \boldsymbol{a} & \boldsymbol{x}_i \end{bmatrix} \sim \begin{bmatrix} \boldsymbol{a}' & \boldsymbol{x}'_i \end{bmatrix}$, it must hold that $\boldsymbol{a} \sim \boldsymbol{a}'$, i.e., there exists $\boldsymbol{P} \in \Pi(N)$ such that $\boldsymbol{P}\boldsymbol{a} = \boldsymbol{a}'$. Combined together, we have that $\boldsymbol{Q}_i\boldsymbol{P}\boldsymbol{a} = \boldsymbol{a}$.

Next, we choose $\boldsymbol{Q}'_i = \boldsymbol{P}^\top \boldsymbol{Q}_i^\top$ so $\boldsymbol{Q}'_i\boldsymbol{a} = \boldsymbol{Q}'_i\boldsymbol{Q}_i\boldsymbol{P}\boldsymbol{a} = \boldsymbol{a}$. Due to the property of anchors (Lemma E.1), we have $\boldsymbol{Q}'_i\boldsymbol{x}_i = \boldsymbol{x}_i$. Notice that $\boldsymbol{x}_i = \boldsymbol{Q}'_i\boldsymbol{x}_i = \boldsymbol{P}^\top \boldsymbol{Q}_i^\top \boldsymbol{Q}_i\boldsymbol{x}'_i = \boldsymbol{P}\boldsymbol{x}'_i$. Therefore, we can conclude the proof as we have found a permutation matrix $\boldsymbol{P}$ that simultaneously aligns $\boldsymbol{x}_i$ and $\boldsymbol{x}'_i$ for every $i \in [D]$, i.e., $\boldsymbol{X} = \begin{bmatrix} \boldsymbol{x}_1 & \cdots & \boldsymbol{x}_D \end{bmatrix} = \begin{bmatrix} \boldsymbol{P}\boldsymbol{x}_1 & \cdots & \boldsymbol{P}\boldsymbol{x}_D \end{bmatrix} = \boldsymbol{P}\boldsymbol{X}'$. $\square$

Next, we need to examine how many weights are needed to construct an anchor via linear combining all the existing channels. We restate Lemma 4.3 in Lemma E.4 with more specifications as well as a simple result from linear algebra (Lemma E.3) to prove it, as below:

**Lemma E.3.** *Consider $D$ linearly independent weight vectors $\{\boldsymbol{\alpha}_1, \cdots, \boldsymbol{\alpha}_D \in \mathbb{R}^D\}$. Then for every $p, q \in [N]$ such that $\boldsymbol{x}^{(p)} \neq \boldsymbol{x}^{(q)}$, there exists $\boldsymbol{\alpha}_j, j \in [D]$, such that $\boldsymbol{x}^{(p)\top} \boldsymbol{\alpha}_j \neq \boldsymbol{x}^{(q)\top} \boldsymbol{\alpha}_j$.*

*Proof.* This is the basic fact of full-rank linear systems. Prove by contradiction. Suppose for $\forall j \in [D]$ we have $\boldsymbol{x}^{(p)\top} \boldsymbol{\alpha}_j = \boldsymbol{x}^{(q)\top} \boldsymbol{\alpha}_j$. Then we form a linear system: $\boldsymbol{x}^{(p)\top} [\boldsymbol{\alpha}_1 \quad \cdots \quad \boldsymbol{\alpha}_D] = \boldsymbol{x}^{(q)\top} [\boldsymbol{\alpha}_1 \quad \cdots \quad \boldsymbol{\alpha}_D]$. Since $\boldsymbol{\alpha}_1, \cdots, \boldsymbol{\alpha}_D$ are linearly independent, it yields $\boldsymbol{x}^{(p)} = \boldsymbol{x}^{(q)}$, which reaches the contradiction. $\square$

**Lemma E.4** (Anchor Construction). *Consider a set of weight vectors $\{\boldsymbol{\alpha}_1, \cdots, \boldsymbol{\alpha}_{K_1} \in \mathbb{R}^D\}$ with $K_1 = N(N-1)(D-1)/2 + 1$, of which every $D$-length subset, i.e., $\{\boldsymbol{\alpha}_j : \forall j \in \mathcal{J}\}, \forall \mathcal{J} \subseteq [K_1], |\mathcal{J}| = D$, is linearly independent, then for every data matrix $\boldsymbol{X} \in \mathbb{R}^{N \times D}$, there exists $j^* \in [K_1]$, $\boldsymbol{X}\boldsymbol{\alpha}_{j^*}$ is an anchor of $\boldsymbol{X}$.*

*Proof.* Define a set of pairs which an anchor needs to distinguish: $\mathcal{I} = \{(p, q) : \boldsymbol{x}^{(p)} \neq \boldsymbol{x}^{(q)}\} \subseteq [N]^2$ Consider a $D$-length subset $\mathcal{J} \subseteq [K]$ with $|\mathcal{J}| = D$. Since $\{\boldsymbol{\alpha}_j : \forall j \in \mathcal{J}\}$ are linear independent, we assert by Lemma E.3 that for every pair $(p, q) \in \mathcal{I}$, there exists $j \in \mathcal{J}$, $\boldsymbol{x}^{(p)\top} \boldsymbol{\alpha}_j \neq \boldsymbol{x}^{(q)\top} \boldsymbol{\alpha}_j$. It is equivalent to claim: for every pair $(p, q) \in \mathcal{I}$, at most $D-1$ many $\boldsymbol{\alpha}_j, j \in [K_1]$ satisfy $\boldsymbol{x}^{(p)\top} \boldsymbol{\alpha}_j = \boldsymbol{x}^{(q)\top} \boldsymbol{\alpha}_j$. Based on pigeon-hole principle, as long as $K_1 \geq N(N-1)(D-1)/2 + 1 = (D-1)\binom{N}{2} + 1 \geq (D-1)|\mathcal{I}| + 1$, there must exist $j^* \in [K_1]$ such that $\boldsymbol{x}^{(p)\top} \boldsymbol{\alpha}_{j^*} \neq \boldsymbol{x}^{(q)\top} \boldsymbol{\alpha}_{j^*}$ for $\forall (p, q) \in \mathcal{I}$. By Definition 4.1, $\boldsymbol{X}\boldsymbol{\alpha}_{j^*}$ generates an anchor. $\square$

**Proposition E.5.** *The linear independence condition in Lemma E.4 can be satisfied with probability one by drawing i.i.d. Gaussian random vectors $\boldsymbol{\alpha}_1, \cdots, \boldsymbol{\alpha}_{K_1} \overset{i.i.d.}{\sim} \mathcal{N}(\boldsymbol{0}, \boldsymbol{I})$.*

*Proof.* We first note that generating a $D \times K_1$ Gaussian random matrix ($D \leq K_1$) is equivalent to drawing a matrix with respect to a probability measure defined over $\mathcal{M} = \{\boldsymbol{X} \in \mathbb{R}^{D \times K} : \text{rank}(\boldsymbol{X}) \leq D\}$. Since rank-$D$ matrices are dense in $\mathcal{M}$ (Tsakiris, 2023; Yao et al., 2021), we can conclude that for $\forall \mathcal{J} \subseteq [K_1], |\mathcal{J}| = D, \mathbb{P}(\{\boldsymbol{\alpha}_j : j \in \mathcal{J}\}$ are linearly independent) $= 1$. By union bound, $\mathbb{P}(\{\boldsymbol{\alpha}_j : j \in \mathcal{J}\}$ for all $\mathcal{J} \in [K], |\mathcal{J}| = D$ are linearly independent) $= 1$. $\square$

# F PROOFS FOR THE LP EMBEDDING LAYER

In this section, we complete the proofs for the LP embedding layer (Eq. 2). First we constructively show an upper bound that sufficiently achieves injectivity following our discussion in Sec. 4.1.2, and then prove Theorem 4.5 to reveal a lower bound that is necessary for injectivity. Finally, we show prove the continuity of the inverse of our constructed LP embedding layer with the techniques introduced in Sec. 4.2.

## F.1 UPPER BOUND FOR INJECTIVITY

To prove the upper bound, we construct an LP embedding layer with $L \leq N^5 D^2$ output neurons such that its induced summation is injective.

We also restate Lemma 4.4 to demonstrate the weight construction for linear coupling:

**Lemma F.1** (Linear Coupling, Lemma 4.4). *Consider a group of coefficients $\Gamma = \{\gamma_1, \cdots, \gamma_{K_2} \in \mathbb{R}\}$ with $\gamma_i \neq 0, \forall i \in [K_2], \gamma_i \neq \gamma_j, \forall i, j \in [K_2]$, and $K_2 = N(N-1) + 1$ such that for all 4-tuples $(\gamma_i, \gamma_j, \gamma_k, \gamma_l) \subset \Gamma$, if $\gamma_i \neq \gamma_j, \gamma_i \neq \gamma_k$ then $\gamma_i/\gamma_j \neq \gamma_k/\gamma_l$. It must hold that: Given any $\boldsymbol{x}, \boldsymbol{x}', \boldsymbol{y}, \boldsymbol{y}' \in \mathbb{R}^N$ such that $\boldsymbol{x} \sim \boldsymbol{x}'$ and $\boldsymbol{y} \sim \boldsymbol{y}'$, if $(\boldsymbol{x} - \gamma_k \boldsymbol{y}) \sim (\boldsymbol{x}' - \gamma_k \boldsymbol{y}')$ for every $k \in [K_2]$, then $[\boldsymbol{x} \quad \boldsymbol{y}] \sim [\boldsymbol{x}' \quad \boldsymbol{y}']$.*

*Remark* F.2. A handy choice of $\Gamma$ in Lemma F.1 are prime numbers, which are provably positive, infinitely many, and not divisible by each other.

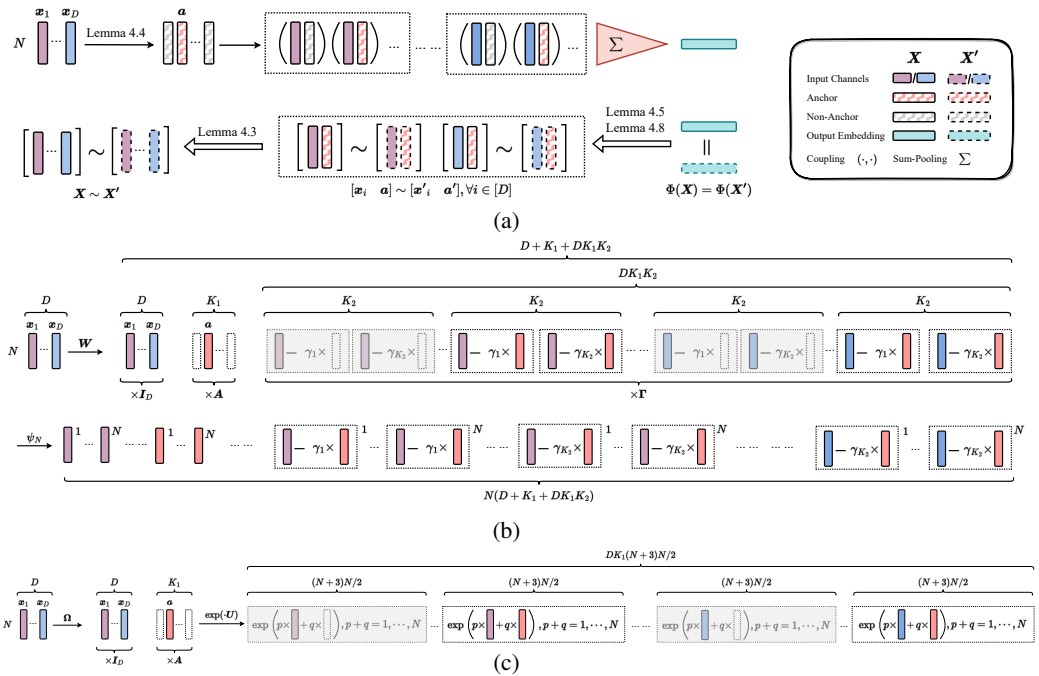

(a)

(b)

(c)

Figure 3: **(a)** illustrates the overall idea to construct LP and LE embedding layers and prove their injectivity. In the forward pass, LP and LE will 1) construct an anchor with redundant non-anchor channels through a linear layer $\boldsymbol{A} = [\boldsymbol{\alpha}_1 \quad \cdots \quad \boldsymbol{\alpha}_{K_1}]$ (Lemma 4.3), 2) and couple each feature channel with the both anchor and non-anchor channels with the their own coupling schemes, respectively. To prove injectivity, the implication follows the converse agenda of construction: 1) by the properties of coupling schemes specified by LP (Lemma 4.4) and LE (Lemma 4.7) layers, we obtain pairwise equivalence with anchors, 2) and by union alignment lemma (Lemma 4.2), we recover the global equivalence. **(b)(c)** depict the detailed construction inside the LP and LE embedding layers, respectively. LP embedding layer utilizes linear combination plus a power mapping to couple feature channels with the anchor(s) and non-anchors, while LE adopts a linear combination plus an exponential mapping, which is essentially an exponential function followed by a bivariate monomial. The constructed components marked in gray represent the redundant pairs between feature channels and non-anchor channels, which will not be used in the chain of implication to prove the injectivity.

*Proof.* We note that $\boldsymbol{x} \sim \boldsymbol{x}'$ and $\boldsymbol{y} \sim \boldsymbol{y}'$ imply that there exist $\boldsymbol{P}_x, \boldsymbol{P}_y \in \Pi(N)$ such that $\boldsymbol{P}_x \boldsymbol{x} = \boldsymbol{x}'$ and $\boldsymbol{P}_y \boldsymbol{y} = \boldsymbol{y}'$. Also $(\boldsymbol{x} - \gamma_k \boldsymbol{y}) \sim (\boldsymbol{x}' - \gamma_k \boldsymbol{y}'), \forall k \in [K_2]$ implies there exists $\boldsymbol{Q}_k \in \Pi(N), \forall k \in [K_2]$ such that $\boldsymbol{Q}_k(\boldsymbol{x} - \gamma_k \boldsymbol{y}) = \boldsymbol{x}' - \gamma_k \boldsymbol{y}'$. Substituting the former to the latter, we can obtain:

$$\left(\boldsymbol{I} - \boldsymbol{Q}_k^\top \boldsymbol{P}_x\right) \boldsymbol{x} = \gamma_k \left(\boldsymbol{I} - \boldsymbol{Q}_k^\top \boldsymbol{P}_x\right) \boldsymbol{y}, \quad \forall k \in [K_2], \tag{19}$$

where for each $k \in [K_2]$, Eq. 19 corresponds to $N$ equalities as follows. Here, we let $(\boldsymbol{Z})_i$ denote the $i$th column of the matrix $\boldsymbol{Z}$.

$$(\boldsymbol{I} - \boldsymbol{Q}_k^\top \boldsymbol{P}_x)_1^\top \boldsymbol{x} = \gamma_k (\boldsymbol{I} - \boldsymbol{Q}_k^\top \boldsymbol{P}_x)_1^\top \boldsymbol{y}$$
$$\vdots \tag{20}$$
$$(\boldsymbol{I} - \boldsymbol{Q}_k^\top \boldsymbol{P}_x)_N^\top \boldsymbol{x} = \gamma_k (\boldsymbol{I} - \boldsymbol{Q}_k^\top \boldsymbol{P}_x)_N^\top \boldsymbol{y}$$

We compare entries in $\boldsymbol{x} = [\cdots x_p \cdots]^\top$ and for each entry index $p \in [N]$, we define a set of non-zero pairwise differences between $x_p$ and other entries in $\boldsymbol{x}$: $\mathcal{D}_x^{(p)} = \{x_p - x_q : q \in [N], x_p \neq x_q\}$. Similarly, for $\boldsymbol{y}$, we define $\mathcal{D}_y^{(p)} = \{y_p - y_q : q \in [N], y_p \neq y_q\}$. We note that for every $n \in [N]$, either option (a) $(\boldsymbol{I} - \boldsymbol{Q}_k^\top \boldsymbol{P}_x)_n^\top \boldsymbol{x} = 0$ or option (b) $(\boldsymbol{I} - \boldsymbol{Q}_k^\top \boldsymbol{P}_x)_n^\top \boldsymbol{x} \in \mathcal{D}_x^{(p)}$ for some $p \in [N]$ as $(\boldsymbol{Q}_k^\top \boldsymbol{P}_x)_n^\top \boldsymbol{x}$ is one of the entries of $\boldsymbol{x}$.

Then, it is sufficient to show there must exist $k \in [K_2]$ such that all of equations in Eq. 20 are induced by the option (a) rather than the option (b), i.e.,

$$\exists k^* \in [K_2], \forall p, n \in [N] \text{ such that } (\boldsymbol{I} - \boldsymbol{Q}_{k^*}^\top \boldsymbol{P}_x)_n^\top \boldsymbol{x} \notin \mathcal{D}_x^{(p)}. \tag{21}$$

This is because Eq. 21 implies:

$$(\boldsymbol{I} - \boldsymbol{Q}_{k^*}\boldsymbol{P}_x)^\top \boldsymbol{x} = \boldsymbol{0} \quad \Rightarrow \quad \boldsymbol{x} = \boldsymbol{Q}_{k^*}^\top \boldsymbol{P}_x \boldsymbol{x} = \boldsymbol{Q}_{k^*}^\top \boldsymbol{x}',$$

$$\text{(Since } \gamma_k \neq 0, \forall k \in [K_2]) \quad (\boldsymbol{I} - \boldsymbol{Q}_{k^*}\boldsymbol{P}_y)^\top \boldsymbol{y} = \boldsymbol{0} \quad \Rightarrow \quad \boldsymbol{y} = \boldsymbol{Q}_{k^*}^\top \boldsymbol{P}_y \boldsymbol{y} = \boldsymbol{Q}_{k^*}^\top \boldsymbol{y}',$$

which is $\begin{bmatrix} \boldsymbol{x} & \boldsymbol{y} \end{bmatrix} = \boldsymbol{Q}_{k^*}^\top \begin{bmatrix} \boldsymbol{x}' & \boldsymbol{y}' \end{bmatrix}$.

To show Eq. 21, we construct $N$ bipartite graphs $\mathcal{G}^{(p)} = (\mathcal{D}_x^{(p)}, \mathcal{D}_y^{(p)}, \mathcal{E}^{(p)})$ for $p \in [N]$ where each $\alpha \in \mathcal{D}_x^{(p)}$ or each $\beta \in \mathcal{D}_y^{(p)}$ is viewed as a node and $(\alpha, \beta) \in \mathcal{E}^{(p)}$ gives an edge if $\alpha = \gamma_k \beta$ for some $k \in [K_2]$. Then we prove the existence of $k^*$ via seeing a contradiction that does counting the number of connected pairs $(\alpha, \beta)$ from two perspectives.

**Perspective of $\mathcal{D}_x^{(p)}$.** We argue that for $\forall p \in [N]$ and arbitrary $\alpha_1, \alpha_2 \in \mathcal{D}_x^{(p)}$, $\alpha_1 \neq \alpha_2$, there exists at most one $\beta \in \mathcal{D}_y^{(p)}$ such that $(\alpha_1, \beta) \in \mathcal{E}^{(p)}$ and $(\alpha_2, \beta) \in \mathcal{E}^{(p)}$. Otherwise, suppose there exists $\beta' \in \mathcal{D}_y^{(p)}$, $\beta' \neq \beta$ such that $(\alpha_1, \beta') \in \mathcal{E}^{(p)}$ and $(\alpha_2, \beta') \in \mathcal{E}^{(p)}$. Then we have $\alpha_1 = \gamma_i \beta$, $\alpha_2 = \gamma_j \beta$, $\alpha_1 = \gamma_k \beta'$, and $\alpha_2 = \gamma_l \beta'$ for some $\gamma_i, \gamma_j, \gamma_k, \gamma_l \in \Gamma$, which is $\gamma_i / \gamma_k = \gamma_k / \gamma_l$. As $\alpha_1 \neq \alpha_2$, it is obvious that $\gamma_i \neq \gamma_j$. Similarly, we have $\gamma_i \neq \gamma_k$. Altogether, it contradicts our assumption on $\Gamma$. Therefore, $|\mathcal{E}^{(p)}| \leq 2\max\{|\mathcal{D}_x^{(p)}|, |\mathcal{D}_y^{(p)}|\} \leq 2(N-1)$. And the total edge number of all bipartite graphs should be less than $2N(N-1)$.

**Perspective of $\Gamma$.** We note that if for some $k \in [K_2]$ that makes $(\boldsymbol{I} - \boldsymbol{Q}_k^\top \boldsymbol{P}_x)_n^\top \boldsymbol{x} \in \mathcal{D}_x^{(p)}$ for some $p, n \in [N]$, i.e., $(\boldsymbol{I} - \boldsymbol{Q}_k^\top \boldsymbol{P}_x)_n^\top \boldsymbol{x} = \gamma_k (\boldsymbol{I} - \boldsymbol{Q}_k^\top \boldsymbol{P}_y)_n^\top \boldsymbol{y} \neq 0$, this $\gamma_k$ contributes at least two edges in the entire bipartite graph, i.e., there being another $n' \in [N]$, $(\boldsymbol{I} - \boldsymbol{Q}_k^\top \boldsymbol{P}_x)_{n'}^\top \boldsymbol{x} = \gamma_k (\boldsymbol{I} - \boldsymbol{Q}_k^\top \boldsymbol{P}_y)_{n'}^\top \boldsymbol{y} \neq 0$. Otherwise, there exists a unique $n^* \in [N]$ such that $(\boldsymbol{I} - \boldsymbol{Q}_k^\top \boldsymbol{P}_x)_{n^*}^\top \boldsymbol{x} \in \mathcal{D}_x^{(p)} (\neq 0)$ and $(\boldsymbol{I} - \boldsymbol{Q}_k^\top \boldsymbol{P}_x)_n^\top \boldsymbol{x} = 0$ for all $n \neq n^*$. This cannot be true because $\boldsymbol{1}^\top (\boldsymbol{I} - \boldsymbol{Q}_k^\top \boldsymbol{P}_x)\boldsymbol{x} = 0$. By which, if $\forall k \in [K_2]$, $\exists p, n \in [N]$ such that $(\boldsymbol{I} - \boldsymbol{Q}_k^\top \boldsymbol{P}_x)_n^\top \boldsymbol{x} \in \mathcal{D}_x^{(p)}$ (i.e., Eq. 21 is always false), then the total number of edges is at least $2K_2 = 2N(N-1) + 2$.

Hereby, we conclude the proof by the contradiction, in which the minimal count of edges $2K_2$ by **Perspective of $\Gamma$** already surpasses the maximal number $2N(N-1)$ by **Perspective of $\mathcal{D}_x^{(p)}$**. $\square$

We wrap off this section by formally stating and proving the injectivity statement of the LP layer.

**Theorem F.3.** *Suppose $\phi : \mathbb{R}^D \to \mathbb{R}^L$ admits the form of Eq. 2,*

$$\phi(\boldsymbol{x}) = \begin{bmatrix} \psi_N(\boldsymbol{w}_1^\top \boldsymbol{x})^\top & \cdots & \psi_N(\boldsymbol{w}_K^\top \boldsymbol{x})^\top \end{bmatrix}, \tag{22}$$

*where $\psi_N$ is the power mapping of degree $N$, $L = KN \leq N^5 D^2$, $K = D + K_1 + DK_1 K_2$ and $\boldsymbol{W} = \begin{bmatrix} \boldsymbol{e}_1 & \cdots & \boldsymbol{e}_D & \boldsymbol{\alpha}_1 & \cdots & \boldsymbol{\alpha}_{K_1} & \cdots & \boldsymbol{\Gamma}_{i,j,k} & \cdots \end{bmatrix}$ is constructed as follows:*

1. *Let the first group of weights $\boldsymbol{e}_1, \cdots, \boldsymbol{e}_D \in \mathbb{R}^D$ buffer the original features, where $\boldsymbol{e}_i$ represents the $i$-th canonical basis.*

2. *Choose the second group of linear weights, $\boldsymbol{\alpha}_1, \cdots, \boldsymbol{\alpha}_{K_1} \in \mathbb{R}^D$ for $K_1$ as large as $N(N-1)(D-1)/2 + 1$, such that the conditions in Lemma E.4 are satisfied.*

3. *Design the third group of weights $\boldsymbol{\Gamma}_{i,j,k}$ for $i \in [D], j \in [K_1], k \in [K_2]$ where $\boldsymbol{\Gamma}_{i,j,k} = \boldsymbol{e}_i - \gamma_k \boldsymbol{\alpha}_j$, $K_2 = N(N-1) + 1$, and $\gamma_k, k \in [K_2]$ are chosen such that conditions in Lemma F.1 are satisfied.*

*Then $\sum_{i=1}^N \phi(\boldsymbol{x}^{(i)})$ is injective (cf. Definition C.7).*

*Proof.* Suppose $\sum_{n=1}^N \phi(\boldsymbol{x}^{(n)}) = \sum_{n=1}^N \phi(\boldsymbol{x}'^{(n)})$ for some $\boldsymbol{X}, \boldsymbol{X}' \in \mathbb{R}^{N \times D}$. Due to the injectivity property of sum-of-power mapping (cf. Lemma D.2):

$$\sum_{n=1}^N \phi\left(\boldsymbol{x}^{(n)}\right) = \sum_{n=1}^N \phi\left(\boldsymbol{x}'^{(n)}\right) \Rightarrow \boldsymbol{X}\boldsymbol{e}_i \sim \boldsymbol{X}'\boldsymbol{e}_i, \forall i \in [D], \boldsymbol{X}\boldsymbol{\alpha}_i \sim \boldsymbol{X}'\boldsymbol{\alpha}_i, \forall i \in [K_1], \tag{23}$$

$$\boldsymbol{X}\boldsymbol{\Gamma}_{i,j,k} \sim \boldsymbol{X}'\boldsymbol{\Gamma}_{i,j,k}, \forall i \in [D], j \in [K_1], k \in [K_2].$$

By Lemma E.4, it is guaranteed that there exists $j^* \in [K_1]$ such that $\boldsymbol{X}\boldsymbol{\alpha}_{j^*}$ is an anchor, and according to Eq. 23, we have $\boldsymbol{X}\boldsymbol{\alpha}_{j^*} \sim \boldsymbol{X}'\boldsymbol{\alpha}_{j^*}$. By Lemma F.1, we induce:

$$\boldsymbol{X}\boldsymbol{e}_i \sim \boldsymbol{X}'\boldsymbol{e}_i, \forall i \in [D], \boldsymbol{X}\boldsymbol{\alpha}_{j^*} \sim \boldsymbol{X}'\boldsymbol{\alpha}_{j^*}, \boldsymbol{X}\boldsymbol{\Gamma}_{i,j,k} \sim \boldsymbol{X}'\mathbf{T}_{i,j,k}, \forall i \in [D], j \in [K_1], k \in [K_2]$$

$$\Rightarrow [\boldsymbol{X}\boldsymbol{\alpha}_{j^*} \quad \boldsymbol{x}_i] \sim [\boldsymbol{X}'\boldsymbol{\alpha}_{j^*} \quad \boldsymbol{x}'_i], \forall i \in [D]. \tag{24}$$

Since $\boldsymbol{X}\boldsymbol{\alpha}_{j^*}$ is an anchor, by union alignment (Lemma E.2), we have:

$$[\boldsymbol{X}\boldsymbol{\alpha}_{j^*} \quad \boldsymbol{x}_i] \sim [\boldsymbol{X}'\boldsymbol{\alpha}_{j^*} \quad \boldsymbol{x}'_i], \forall i \in [D] \Rightarrow \boldsymbol{X} \sim \boldsymbol{X}'. \tag{25}$$

Here $K = D + K_1 + DK_1K_2 \leq N^4 D^2$, thus $L = KN \leq N^5 D^2$, which concludes the proof. $\square$

## F.2 CONTINUITY

Next, we show that under the construction of Theorem F.3, the inverse of $\sum_{i=1}^{N} \phi(\boldsymbol{x}^{(i)})$ is continuous. The main idea is to check the three conditions provided in Lemma 4.8.

**Theorem F.4.** *Suppose $\phi$ admits the form of Eq. 22 and follows the construction in Theorem F.3, then the inverse of LP-induced sum-pooling $\sum_{n=1}^{N} \phi(\boldsymbol{x}^{(n)})$ is continuous.*

*Proof.* The proof is done by invoking Lemma 4.8. First of all, the inverse of $\Phi(\boldsymbol{X}) = \sum_{i=1}^{N} \phi(\boldsymbol{x}^{(i)})$, denoted as $\Phi^{-1}$, exists due to Theorem F.3. By Lemma C.6, any closed and bounded subset of $(\mathbb{R}^{N \times D}/\sim, d_\Pi)$ is compact. Trivially, $\Phi(\boldsymbol{X})$ is continuous. Then it remains to show the condition **(c)** in Lemma 4.8. Same with Corollary D.6 while focusing on the real domain, let $\mathcal{Z}_{\Psi_N} = \{\Psi_N(\boldsymbol{x}) : \boldsymbol{x} \in \mathbb{R}^N\} \subseteq \mathbb{R}^N$ be the range of sum-of-power mapping, and define $\mathcal{Z}_\Phi = \{\Phi(\boldsymbol{X}) : \boldsymbol{X} \in \mathbb{R}^{N \times D}\} \subseteq \mathcal{Z}_{\Psi_N}^K$ be the range of $\Phi$, which is also a subset of $\mathcal{Z}_{\Psi_N}^K$. We decompose $\Phi^{-1} = \pi \circ \widehat{\Psi_N}^\dagger$ into two mappings following the similar idea of proving its existence:

$$(\mathcal{Z}_\Phi, d_\infty) \xrightarrow{\widehat{\Psi_N}^\dagger} \mathcal{R} \subseteq (\mathbb{R}^N/\sim, d_\Pi)^K \xrightarrow{\pi} (\mathbb{R}^{N \times D}/\sim, d_\Pi) ,$$

where $\widehat{\Phi_N}^\dagger$ is defined in Eq. 18 (Corollary D.6), $\mathcal{R}$ is the image of $\mathcal{Z}$ under $\widehat{\Phi_N}^\dagger$, and $\pi$ exists due to union alignment (i.e., Eqs. 24 and 25 in Theorem F.3). Also according to our construction in Theorem F.3, for any $\boldsymbol{Z} \in \mathcal{R}$, consider its first $D$ columns produced by $\{\boldsymbol{e}_i = \boldsymbol{e}_i, \forall i \in D\}$, we know that $\boldsymbol{z}_i \sim \pi(\boldsymbol{Z})_i$ for every $i \in [D]$. Therefore, $\forall \boldsymbol{Z}, \boldsymbol{Z}' \in \mathcal{R}$ such that $d_\Pi^K(\boldsymbol{Z}, \boldsymbol{Z}') \leq C$ for some constant $C > 0$ in terms of the product metric $d_\Pi^K$ (cf. Definition C.2), the inequality holds:

$$d_\Pi(\pi(\boldsymbol{Z}), \pi(\boldsymbol{Z}')) \leq \max_{i \in [D]} d_\Pi(\boldsymbol{z}_i, \boldsymbol{z}'_i) \leq d_\Pi^K(\boldsymbol{Z}, \boldsymbol{Z}') \leq C, \tag{26}$$

which implies $\pi$ maps every bounded set in $\mathcal{R}$ to a bounded set in $(\mathbb{R}^{N \times D}/\sim, d_\Pi)$. Now we conclude the proof by the following chain of argument:

$$\mathcal{S} \subseteq (\mathcal{Z}_\Phi, d_\infty) \text{ is bounded} \xRightarrow{\text{Corollary D.6}} \widehat{\Psi_N}^\dagger(\mathcal{S}) \text{ is bounded} \xRightarrow{\text{Eq. 26}} \pi \circ \widehat{\Psi_N}^\dagger(\mathcal{S}) \text{ is bounded.}$$

$\square$

## F.3 LOWER BOUND FOR INJECTIVITY

In this section, we prove Theorem 4.5 which shows that $K \geq D + 1$ is necessary for injectivity of LP-induced sum-pooling when $D \geq 2$. Our argument mainly generalizes Lemma 2 of Tsakiris & Peng (2019) to our equivalence class. To proceed our argument, we define the linear subspace $\mathcal{V}$ by vectorizing $[\boldsymbol{X}\boldsymbol{w}_1 \quad \cdots \quad \boldsymbol{X}\boldsymbol{w}_K]$ as below:

$$\mathcal{V} := \left\{ \begin{bmatrix} \boldsymbol{X}\boldsymbol{w}_1 \\ \vdots \\ \boldsymbol{X}\boldsymbol{w}_K \end{bmatrix} : \boldsymbol{X} \in \mathbb{R}^{N \times D} \right\} = \mathcal{R}\left( \begin{bmatrix} (\boldsymbol{w}_1^\top \otimes \boldsymbol{I}_N) \\ \vdots \\ (\boldsymbol{w}_K^\top \otimes \boldsymbol{I}_N) \end{bmatrix} \right), \tag{27}$$

where $\mathcal{R}(\boldsymbol{Z})$ denotes the column space of $\boldsymbol{Z}$ and $\otimes$ is the Kronecker product. $\mathcal{V}$ is a linear subspace of $\mathbb{R}^{NK}$ with dimension at most $\mathbb{R}^{ND}$, characterized by $\boldsymbol{w}_1, \cdots, \boldsymbol{w}_K \in \mathbb{R}^D$. For the sake of notation simplicity, we denote $\Pi(N)^{\otimes K} = \{\text{diag}(\boldsymbol{Q}_1, \cdots, \boldsymbol{Q}_K) : \forall \boldsymbol{Q}_1, \cdots, \boldsymbol{Q}_K \in \Pi(N)\}$, and $\boldsymbol{I}_K \otimes \Pi(N) = \{\boldsymbol{I}_K \otimes \boldsymbol{Q} : \forall \boldsymbol{Q} \in \Pi(N)\}$. Next, we define the notion of unique recoverability (Tsakiris & Peng, 2019) as below:

**Definition F.5** (Unique Recoverability). The subspace $\mathcal{V}$ is called uniquely recoverable under $\boldsymbol{Q} \in \Pi(N)^{\otimes K}$ if whenever $\boldsymbol{x}, \boldsymbol{x}' \in \mathcal{V}$ satisfy $\boldsymbol{Q}\boldsymbol{x} = \boldsymbol{x}'$, there exists $\boldsymbol{P} \in \boldsymbol{I}_K \otimes \Pi(N)$, $\boldsymbol{P}\boldsymbol{x} = \boldsymbol{x}'$.

Subsequently, we derive a necessary condition for the unique recoverability:

**Lemma F.6.** *A linear subspace $\mathcal{V} \subseteq \mathbb{R}^{NK}$ is uniquely recoverable under $\boldsymbol{Q} \in \Pi(N)^{\otimes K}$ only if there exists $\boldsymbol{P} \in \boldsymbol{I}_K \otimes \Pi(N)$, $\boldsymbol{Q}(\mathcal{V}) \cap \mathcal{V} \subseteq \mathcal{E}_{\boldsymbol{Q}\boldsymbol{P}^\top, \lambda=1}$, where $\mathcal{E}_{\boldsymbol{Q}\boldsymbol{P}^\top, \lambda}$ denotes the eigenspace corresponding to the eigenvalue $\lambda$.*

*Proof.* It is sufficient to prove that $\boldsymbol{Q}(\mathcal{V}) \cap \mathcal{V} \subseteq \bigcup_{\boldsymbol{P} \in \boldsymbol{I}_K \otimes \Pi(K)} \mathcal{E}_{\boldsymbol{Q}\boldsymbol{P}^\top, \lambda=1}$. This is because the LHS is a subspace and the RHS is a union of subspaces. When a subspace is a subset of a union of subspaces, such a subspace must be a subset of one of the subspaces, i.e., $\boldsymbol{Q}(\mathcal{V}) \cap \mathcal{V} \subseteq \bigcup_{\boldsymbol{P} \in \boldsymbol{I}_K \otimes \Pi(K)} \mathcal{E}_{\boldsymbol{Q}\boldsymbol{P}^\top, \lambda=1}$ implies there exists $\boldsymbol{P} \in \boldsymbol{I}_K \otimes \Pi(K)$ such that $\boldsymbol{Q}(\mathcal{V}) \cap \mathcal{V} \subseteq \mathcal{E}_{\boldsymbol{Q}\boldsymbol{P}^\top, \lambda=1}$.

Next, we prove $\boldsymbol{Q}(\mathcal{V}) \cap \mathcal{V} \subseteq \bigcup_{\boldsymbol{P} \in \boldsymbol{I}_K \otimes \Pi(K)} \mathcal{E}_{\boldsymbol{Q}\boldsymbol{P}^\top, \lambda=1}$ by contradiction. Suppose there exists $\boldsymbol{x} \in \boldsymbol{Q}(\mathcal{V}) \cap \mathcal{V}$ but $\boldsymbol{x} \notin \bigcup_{\boldsymbol{P} \in \boldsymbol{I}_K \otimes \Pi(K)} \mathcal{E}_{\boldsymbol{Q}\boldsymbol{P}^\top, \lambda=1}$. Or equivalently, there exists $\boldsymbol{x}' \in \mathcal{V}$ and $\boldsymbol{x} = \boldsymbol{Q}\boldsymbol{x}'$, while for $\forall \boldsymbol{P} \in \boldsymbol{I}_K \otimes \Pi(N)$, $\boldsymbol{x} \neq \boldsymbol{Q}\boldsymbol{P}^\top \boldsymbol{x}$. This implies $\boldsymbol{Q}^\top \boldsymbol{x} = \boldsymbol{x}' \neq \boldsymbol{P}\boldsymbol{x}$ for $\forall \boldsymbol{P} \in \boldsymbol{I}_K \otimes \Pi(N)$. However, this contradicts the fact that $\mathcal{V} \subseteq \mathbb{R}^{NK}$ is uniquely recoverable (cf. Definition F.5). $\square$

We also introduce a useful Lemma F.7 that gets rid of the discussion on $\boldsymbol{Q}$ in the inclusion:

**Lemma F.7.** *Suppose $\mathcal{V} \subseteq \mathbb{R}^N$ is a linear subspace, and $\boldsymbol{T}$ is a linear mapping. $\boldsymbol{T}(\mathcal{V}) \cap \mathcal{V} \cap \mathcal{E}_{\boldsymbol{T}, \lambda} = \boldsymbol{0}$ if and only if $\mathcal{V} \cap \mathcal{E}_{\boldsymbol{T}, \lambda} = \boldsymbol{0}$.*

*Proof.* The sufficiency is straightforward. The necessity is shown by contradiction: Suppose $\mathcal{V} \cap \mathcal{E}_{\boldsymbol{T}, \lambda} \neq \boldsymbol{0}$, then there exists $\boldsymbol{x} \in \mathcal{V} \cap \mathcal{E}_{\boldsymbol{T}, \lambda}$ such that $\boldsymbol{x} \neq \boldsymbol{0}$. Then $\boldsymbol{T}\boldsymbol{x} = \lambda \boldsymbol{x}$ implies $\boldsymbol{x} \in \boldsymbol{T}(\mathcal{V})$. Hence, $\boldsymbol{x} \in \boldsymbol{T}(\mathcal{V}) \cap \mathcal{V} \cap \mathcal{E}_{\boldsymbol{T}, \lambda}$ which reaches the contradiction. $\square$

Now we are ready to present the proof of Theorem 4.5, restated below:

**Theorem F.8** (Lower Bound, Theorem 4.5). *Consider data matrices $\boldsymbol{X} \in \mathbb{R}^{N \times D}$ where $D \geq 2$. If $K \leq D$, then for every $\boldsymbol{w}_1, \cdots, \boldsymbol{w}_K$, there exists $\boldsymbol{X}' \in \mathbb{R}^{N \times D}$ such that $\boldsymbol{X} \not\sim \boldsymbol{X}'$ but $\boldsymbol{X}\boldsymbol{w}_i \sim \boldsymbol{X}'\boldsymbol{w}_i$ for every $i \in [K]$.*

*Proof.* Proved by contrapositive. First notice that, $\forall \boldsymbol{X}, \boldsymbol{X}' \in \mathbb{R}^{N \times D}, \boldsymbol{X}\boldsymbol{w}_i \sim \boldsymbol{X}'\boldsymbol{w}_i, \forall i \in [K] \Rightarrow \boldsymbol{X} \sim \boldsymbol{X}'$ holds if and only if $\dim \mathcal{V} = ND$ and $\mathcal{V}$ is uniquely recoverable under all possible $\boldsymbol{Q} \in \Pi(N)^{\otimes K}$. By Lemma F.6, for every $\boldsymbol{Q} \in \Pi(N)^{\otimes K}$, there exists $\boldsymbol{P} \in \boldsymbol{I}_K \otimes \Pi(N)$ such that $\boldsymbol{Q}(\mathcal{V}) \cap \mathcal{V} \subset \mathcal{E}_{\boldsymbol{Q}\boldsymbol{P}^\top, \lambda=1}$. This is $\boldsymbol{Q}(\mathcal{V}) \cap \mathcal{V} \cap \mathcal{E}_{\boldsymbol{Q}\boldsymbol{P}^\top, \lambda} = 0$ for all $\lambda \neq 1$. By Lemma F.7, we have $\mathcal{V} \cap \mathcal{E}_{\boldsymbol{Q}\boldsymbol{P}^\top, \lambda} = 0$ for all $\lambda \neq 1$. Then proof is concluded by discussing the dimension of ambient space $\mathbb{R}^{NK}$ such that an $ND$-dimensional subspace $\mathcal{V}$ can reside. To ensure $\mathcal{V} \cap \mathcal{E}_{\boldsymbol{Q}\boldsymbol{P}^\top, \lambda} = 0$ for all $\lambda \neq 1$, it is necessary that $\dim \mathcal{V} \leq \min_{\lambda \neq 1} \operatorname{codim} \mathcal{E}_{\boldsymbol{Q}\boldsymbol{P}^\top, \lambda}$ for every $\boldsymbol{Q} \in \Pi(N)^{\otimes K}$ and its associated $\boldsymbol{P} \in \boldsymbol{I}_K \otimes \Pi(N)$. Relaxing the dependence between $\boldsymbol{Q}$ and $\boldsymbol{P}$, we derive the inequality:

$$ND = \dim \mathcal{V} \leq \min_{\boldsymbol{Q} \in \Pi(N)^{\otimes K}} \max_{\boldsymbol{P} \in \boldsymbol{I}_K \otimes \Pi(N)} \min_{\lambda \neq 1} \operatorname{codim} \mathcal{E}_{\boldsymbol{Q}\boldsymbol{P}^\top, \lambda} \leq NK - 1, \tag{28}$$

where the last inequality considers the scenario where every non-one eigenspace is one-dimensional, which is achievable when $K \geq 2$. Hence, we can bound $K \geq (ND + 1)/N$, i.e., $K \geq D + 1$. $\square$

# G PROOFS FOR LE EMBEDDING LAYER

In this section, we present the complete proof for the LE embedding layer following Sec. 4.1.3.

## G.1 UPPER BOUND FOR INJECTIVITY

To prove the upper bound, we construct an LE embedding layer with $L \leq N^4 D^2$ output neurons such that its induced sum-pooling is injective. The main construction idea is to couple every channel and anchor with the real and imaginary components of complex numbers and invoke the injectiviy of sum-of-power mapping over the complex domain to show the invertibility.

With Lemma D.2, we can prove Lemma 4.7 restated and proved as below:

**Lemma G.1.** *For any pair of vectors $\boldsymbol{x}, \boldsymbol{y} \in \mathbb{R}^N, \boldsymbol{x}', \boldsymbol{y}' \in \mathbb{R}^N$, if $\sum_{i\in[N]} \boldsymbol{x}_i^{l-k}\boldsymbol{y}_i^k = \sum_{i\in[N]} \boldsymbol{x}'^{l-k}_i\boldsymbol{y}'^k_i$ for every $l, k \in [N]$ such that $0 \leq k \leq l$, then $[\boldsymbol{x} \quad \boldsymbol{y}] \sim [\boldsymbol{x}' \quad \boldsymbol{y}']$.*

*Proof.* If for any pair of vectors $\boldsymbol{x}, \boldsymbol{y} \in \mathbb{R}^N, \boldsymbol{x}', \boldsymbol{y}' \in \mathbb{R}^N$ such that $\sum_{i\in[N]} \boldsymbol{x}_i^{l-k}\boldsymbol{y}_i^k = \sum_{i\in[N]} \boldsymbol{x}'^{l-k}_i\boldsymbol{y}'^k_i$ for every $l, k \in [N]$, $0 \leq k \leq l$, then for $\forall l \in [N]$,

$$\sum_{i=1}^{N} \psi_N(\boldsymbol{x}_i + \boldsymbol{y}_i\sqrt{-1})_l = \sum_{i=1}^{N}(\boldsymbol{x}_i + \boldsymbol{y}_i\sqrt{-1})^l \tag{29}$$

$$= \sum_{i=1}^{N}\sum_{k=0}^{l}(\sqrt{-1})^k\boldsymbol{x}_i^{l-k}\boldsymbol{y}_i^k = \sum_{k=0}^{l}(\sqrt{-1})^k\left(\sum_{i=1}^{N}\boldsymbol{x}_i^{l-k}\boldsymbol{y}_i^k\right) \tag{30}$$

$$= \sum_{k=0}^{l}(\sqrt{-1})^k\left(\sum_{i=1}^{N}\boldsymbol{x}'^{l-k}_i\boldsymbol{y}'^k_i\right) = \sum_{i=1}^{N}\sum_{k=0}^{l}(\sqrt{-1})^k\boldsymbol{x}'^{l-k}_i\boldsymbol{y}'^k_i \tag{31}$$

$$= \sum_{i=1}^{N}(\boldsymbol{x}'_i + \boldsymbol{y}'_i\sqrt{-1})^l = \sum_{i=1}^{N}\psi_N(\boldsymbol{x}'_i + \boldsymbol{y}'_i\sqrt{-1})_l, \tag{32}$$

in which we reorganize terms in the summation and apply the given condition to establish equality between Eq. 30 and 31. $\psi_N$ denotes the complex power mapping of degree $N$ (cf. Definition D.1). Consider Eq. 32 for every $l \in [N]$, we can yield:

$$\Psi_N\left(\boldsymbol{x} + \boldsymbol{y}\sqrt{-1}\right) = \Psi_N\left(\boldsymbol{x}' + \boldsymbol{y}'\sqrt{-1}\right),$$

where $\Psi_N$ is the sum-of-power mapping (cf. Definition D.1). Then by Lemma D.2, we have $(\boldsymbol{x} + \boldsymbol{y}\sqrt{-1}) \sim (\boldsymbol{x}' + \boldsymbol{y}'\sqrt{-1})$, which is equivalent to the statement $[\boldsymbol{x} \quad \boldsymbol{y}] \sim [\boldsymbol{x}' \quad \boldsymbol{y}']$. $\qquad\square$

**Lemma G.2.** *Suppose $f : \mathbb{R} \to \mathbb{R}$ is an injective function. We denote $f(\boldsymbol{X})$ as applying $f$ element-wisely to entries in $\boldsymbol{X}$, i.e., $f(\boldsymbol{X})_{i,j} = f(\boldsymbol{X}_{i,j}), \forall i \in [N], j \in [D]$. Then for any $\boldsymbol{X}, \boldsymbol{X}' \in \mathbb{R}^{N\times D}$, $f(\boldsymbol{X}) \sim f(\boldsymbol{X}')$ implies $\boldsymbol{X} \sim \boldsymbol{X}'$.*

*Proof.* Since $f(\boldsymbol{X}) \sim f(\boldsymbol{X}')$, there exists $\boldsymbol{P} \in \Pi(N)$ such that $f(\boldsymbol{X}) = \boldsymbol{P}f(\boldsymbol{X}')$. Notice that element-wise functions are permutation-equivariant, then $f(\boldsymbol{X}) = f(\boldsymbol{P}\boldsymbol{X}')$. Since $f$ is injective, we conclude the proof by applying its inverse $f^{-1}$ to both sides. $\qquad\square$

**Lemma G.3.** *Suppose $f : \mathbb{R} \to \mathbb{R}$ is an injective function. We denote $f(\boldsymbol{X})$ as applying $f$ element-wisely to entries in $\boldsymbol{X}$, i.e., $f(\boldsymbol{X})_{i,j} = f(\boldsymbol{X}_{i,j}), \forall i \in [N], j \in [D]$. For any $\boldsymbol{X} \in \mathbb{R}^{N\times D}$, if $\boldsymbol{a}$ is an anchor of $\boldsymbol{X}$ (cf. Definition 4.1), then $f(\boldsymbol{a})$ is also an anchor of $f(\boldsymbol{X})$.*

*Proof.* Proved by contradiction. Suppose $f(\boldsymbol{a})$ is not an anchor of $f(\boldsymbol{X})$. Then there exists $i, j \in [N]$, $f(\boldsymbol{x}^{(i)}) \neq f(\boldsymbol{x}^{(j)})$ while $f(\boldsymbol{a}_i) = f(\boldsymbol{a}_j)$. Since $f$ is injective, then $f(\boldsymbol{x}^{(i)}) \neq f(\boldsymbol{x}^{(j)})$ implies $\boldsymbol{x}^{(i)} \neq \boldsymbol{x}^{(j)}$, whereas $f(\boldsymbol{a}_i) = f(\boldsymbol{a}_j)$ induces $\boldsymbol{a}_i = \boldsymbol{a}_j$. This leads to a contradiction. $\qquad\square$

Now we are ready to prove the injectiviy of the LE layer.

**Theorem G.4.** *Suppose $\phi : \mathbb{R}^D \to \mathbb{R}^L$ admits the form of Eq. 3:*

$$\phi(\boldsymbol{x}) = \begin{bmatrix} \exp(\boldsymbol{v}_1^\top\boldsymbol{x}) & \cdots & \exp(\boldsymbol{v}_L^\top\boldsymbol{x}) \end{bmatrix}, \tag{33}$$

*where $L = DK_1N(N+3)/2 \leq N^4D^2$, $\boldsymbol{V} = [\cdots \quad \boldsymbol{v}_{i,j,p,q} \quad \cdots] \in \mathbb{R}^{D\times L}$, $i \in [D], j \in [K_1], p \in [N], q \in [p+1]$, are constructed as follows:*

1. *Define a group of weights $\boldsymbol{e}_1, \cdots, \boldsymbol{e}_D \in \mathbb{R}^D ss$, where $\boldsymbol{e}_i$ is the $i$-th canonical basis.*

2. *Choose another group of linear weights, $\boldsymbol{\alpha}_1, \cdots, \boldsymbol{\alpha}_{K_1} \in \mathbb{R}^D$ for $K_1$ as large as $N(N-1)(D-1)/2 + 1$, such that the conditions in Lemma E.4 are satisfied.*

3. *Design the weight matrix as $\boldsymbol{v}_{i,j,p,q} \in \mathbb{R}^D$ for $i \in [D], j \in [K_1], p \in [N], q \in [p+1]$ such that $\boldsymbol{v}_{i,j,p,q} = (q-1)\boldsymbol{e}_i + (p-q+1)\boldsymbol{\alpha}_j$.*

*Then $\sum_{i=1}^N \phi(\boldsymbol{x}^{(i)})$ is injective (cf. Definition C.7).*

*Proof.* First of all, we count the number of weight vectors $\{\boldsymbol{v}_{i,j,p,q}\}$ where $i \in [D], j \in [K_1], p \in [N], q \in [p+1]$: $L = DK_1 \sum_{p=1}^N (p+1) = DK_1(N+3)N/2 \leq N^4 D^2$, as desired.

Let $\boldsymbol{\Omega} = [\boldsymbol{e}_1 \quad \cdots \quad \boldsymbol{e}_D \quad \boldsymbol{\alpha}_1 \quad \cdots \quad \boldsymbol{\alpha}_{K_1}] \in \mathbb{R}^{D \times (D+K_1)}$, and $\boldsymbol{u}_{i,j,p,q} = (q-1)\boldsymbol{e}_i + (p-q+1)\boldsymbol{e}_{j+D} \in \mathbb{R}^{D+K_1}$, then we can rewrite $\boldsymbol{v}_{i,j,p,q} = \boldsymbol{\Omega}\boldsymbol{u}_{i,j,p,q}$ for every $i \in [D], j \in [K_1], p \in [N], q \in [p+1]$. Then, for $\boldsymbol{x} \in \mathbb{R}^D$, we can cast Eq. 33 into:

$$\phi(\boldsymbol{x}) = \begin{bmatrix} \cdots & \exp(\boldsymbol{u}_{i,j,p,q}^\top \boldsymbol{\Omega}^\top \boldsymbol{x}) & \cdots \end{bmatrix} = \begin{bmatrix} \cdots & \exp(\boldsymbol{u}_{i,j,p,q}^\top \log(\exp(\boldsymbol{\Omega}^\top \boldsymbol{x}))) & \cdots \end{bmatrix}, \quad (34)$$

where $\log(\cdot)$ and $\exp(\cdot)$ operate on vectors element-wisely. By the arithmetic rule of exponential and logarithm, we can rewrite for $\forall i \in [D], j \in [K_1], p \in [N], q \in [p+1]$

$$\phi(\boldsymbol{x})_{i,j,p,q} = \exp(\boldsymbol{u}_{i,j,p,q}^\top \log(\exp(\boldsymbol{\Omega}^\top \boldsymbol{x}))) = \prod_{k=1}^{D+K_1} \left[\exp\left(\boldsymbol{\Omega}^\top \boldsymbol{x}\right)_k\right]^{(\boldsymbol{u}_{i,j,p,q})_k} \quad (35)$$

$$= \exp(\boldsymbol{e}_i^\top \boldsymbol{x})^{q-1} \exp(\boldsymbol{\alpha}_j^\top \boldsymbol{x})^{p-q+1} = \exp(\boldsymbol{x}_i)^{q-1} \exp(\boldsymbol{\alpha}_j^\top \boldsymbol{x})^{p-q+1}. \quad (36)$$

Then for $\boldsymbol{X}, \boldsymbol{X}' \in \mathbb{R}^{N \times D}$, we have:

$$\sum_{n \in [N]} \phi\left(\boldsymbol{x}^{(n)}\right) = \sum_{n \in [N]} \phi\left(\boldsymbol{x}'^{(n)}\right) \quad (37)$$

$$\Updownarrow \quad (38)$$

$$\sum_{n \in [N]} \exp(\boldsymbol{x}_i)_n^{q-1} \exp\left(\boldsymbol{X}\boldsymbol{\alpha}_j\right)_n^{p-q+1} = \sum_{n \in [N]} \exp(\boldsymbol{x}_i')_n^{q-1} \exp\left(\boldsymbol{X}'\boldsymbol{\alpha}_j\right)_n^{p-q+1}, \quad (39)$$

$$\forall i \in [D], j \in [K_1], p \in [N], q \in [p+1].$$

By Lemma G.1, we obtain that $[\exp(\boldsymbol{X}\boldsymbol{\alpha}_j) \quad \exp(\boldsymbol{x}_i)] \sim [\exp(\boldsymbol{X}'\boldsymbol{\alpha}_j) \quad \exp(\boldsymbol{x}'_i)]$ for $\forall i \in [D], j \in [K_1]$. By Lemma E.4, there exists $j^* \in [K_1]$ such that $\boldsymbol{X}\boldsymbol{\alpha}_{j^*}$ is an anchor of $\boldsymbol{X}$. By Lemma G.3, $\exp(\boldsymbol{X}\boldsymbol{\alpha}_{j^*})$ is also an anchor of $\exp(\boldsymbol{X})$. By union alignment (Lemma E.2), we have:

$$[\exp\left(\boldsymbol{X}\boldsymbol{\alpha}_{j^*}\right) \quad \exp(\boldsymbol{x}_i)] \sim [\exp\left(\boldsymbol{X}'\boldsymbol{\alpha}_{j^*}\right) \quad \exp(\boldsymbol{x}'_i)], \forall i \in [D] \Rightarrow \exp(\boldsymbol{X}) \sim \exp(\boldsymbol{X}'). \quad (40)$$

Finally, we conclude the proof by Lemma G.2:

$$\exp(\boldsymbol{X}) \sim \exp(\boldsymbol{X}') \Rightarrow \boldsymbol{X} \sim \boldsymbol{X}'. \quad (41)$$

$\square$

## G.2 CONTINUITY

The proof idea of continuity for LE layer shares the similar spirit with the LP layer, but involves additional steps. This is because we cannot directly achieve the end-to-end boundedness for condition **(c)** in Lemma 4.8 if decomposing the inverse map of the sum-pooling, following the proof idea of injectivity (cf. Theorem G.4), since the last step (Eq. 41) requires taking logarithm over $\exp(\boldsymbol{X})$ while logarithm does not preserve boundedness.

**Theorem G.5.** *Suppose $\phi$ admits the form of Eq. 33 and follows the construction in Theorem G.4, then the inverse of LE-induced sum-pooling $\sum_{n=1}^N \phi(\boldsymbol{x}^{(n)})$ is continuous.*

*Proof.* Our proof requires the following mathematical tool to help rewrite $\phi$. First, following the construction in Theorem G.4 and by Eq. 34, we can rewrite $\phi(\boldsymbol{x})$ as:

$$\phi(\boldsymbol{x}) = \begin{bmatrix} \cdots & \exp(\boldsymbol{u}_{i,j,p,q}^\top \log(\exp(\boldsymbol{\Omega}^\top \boldsymbol{x}))) & \cdots \end{bmatrix}, \quad (42)$$

where $\boldsymbol{\Omega} = [\boldsymbol{I}_D \quad \boldsymbol{A}]$, and $\boldsymbol{A} = [\boldsymbol{\alpha}_1 \quad \cdots \quad \boldsymbol{\alpha}_{K_1}]$. Define $\widehat{\phi} : \mathbb{R}_{>0}^D \to \mathbb{R}^L$ as below:

$$\widehat{\phi}(\boldsymbol{x}) = \left[ \cdots \quad \exp\left( \boldsymbol{u}_{i,j,p,q}^\top \log\left( \begin{bmatrix} \boldsymbol{x} \\ \exp(\boldsymbol{A}^\top \log(\boldsymbol{x})) \end{bmatrix} \right) \right) \quad \cdots \right]. \tag{43}$$

Notice that $\phi(\boldsymbol{x}) = \widehat{\phi} \circ \exp(\boldsymbol{x})$. Recall $\Phi(\boldsymbol{X}) = \sum_{i=1}^N \phi(\boldsymbol{x}^{(i)})$ and define $\widehat{\Phi} : \mathbb{R}_{>0}^D \to \mathbb{R}^L$ as $\widehat{\Phi}(\boldsymbol{X}) = \sum_{i=1}^N \widehat{\phi}(\boldsymbol{x}^{(i)})$, and then $\Phi(\boldsymbol{X}) = \widehat{\Phi}(\exp(\boldsymbol{X}))$. The proof can be concluded by two steps: 1) notice that $\widehat{\Phi}$ has a continuous inverse $\widehat{\Phi}^{-1}$ by Lemma G.6 and G.7, and then 2) show that the continuous inverse of $\Phi$ exists by letting $\Phi^{-1}(\boldsymbol{X}) = \log \circ \widehat{\Phi}^{-1}(\boldsymbol{X})$. $\qquad\square$

**Lemma G.6.** *Consider* $\widehat{\phi} : \mathbb{R}_{>0}^D \to \mathbb{R}^L$ *as defined in Eq. 43, Theorem G.5, then* $\widehat{\Phi}(\boldsymbol{X}) = \sum_{i=1}^N \widehat{\phi}(\boldsymbol{x}^{(i)})$ *is injective.*

*Proof.* We use the fact that $\widehat{\phi}(\boldsymbol{x}) = \phi \circ \log(\boldsymbol{x})$, and borrow the same proof from Theorem G.4:

$$\sum_{n \in [N]} \widehat{\phi}\left( \boldsymbol{x}^{(n)} \right) = \sum_{n \in [N]} \widehat{\phi}\left( \boldsymbol{x}'^{(n)} \right) \tag{44}$$

$$\Updownarrow$$

$$\sum_{n \in [N]} (\boldsymbol{x}_i)_n^{q-1} \exp\left( \log(\boldsymbol{X})\boldsymbol{\alpha}_j \right)_n^{p-q+1} = \sum_{n \in [N]} (\boldsymbol{x}'_i)_n^{q-1} \exp\left( \log(\boldsymbol{X}')\boldsymbol{\alpha}_j \right)_n^{p-q+1}, \tag{45}$$

$$\forall i \in [D], j \in [K_1], p \in [N], q \in [p+1].$$

By Lemma G.1, we obtain that $[\exp(\log(\boldsymbol{X})\boldsymbol{\alpha}_j) \quad \boldsymbol{x}_i] \sim [\exp(\log(\boldsymbol{X}')\boldsymbol{\alpha}_j) \quad \boldsymbol{x}'_i]$ for $\forall i \in [D], j \in [K_1]$. By Lemma E.4, there exists $j^* \in [K_1]$ such that $\log(\boldsymbol{X})\boldsymbol{\alpha}_{j^*}$ is an anchor of $\log(\boldsymbol{X})$. By Lemma G.3, $\exp(\log(\boldsymbol{X})\boldsymbol{\alpha}_{j^*})$ is also an anchor of $\boldsymbol{X}$. By union alignment (Lemma E.2):

$$[\exp(\log(\boldsymbol{X})\boldsymbol{\alpha}_{j^*}) \quad \boldsymbol{x}_i] \sim [\exp(\log(\boldsymbol{X}')\boldsymbol{\alpha}_{j^*}) \quad \boldsymbol{x}'_i], \forall i \in [D] \Rightarrow \boldsymbol{X} \sim \boldsymbol{X}'. \tag{46}$$

$$\square$$

**Lemma G.7.** *Consider* $\widehat{\phi} : \mathbb{R}_{>0}^D \to \mathbb{R}^L$ *and* $\widehat{\Phi}(\boldsymbol{X}) = \sum_{i=1}^N \widehat{\phi}(\boldsymbol{x}^{(i)})$ *as defined in Eq. 43, Theorem G.5. Let* $\mathcal{Z}_{\widehat{\Phi}} = \{\widehat{\Phi}(\boldsymbol{X}) : \boldsymbol{X} \in \mathbb{R}_{>0}^{N \times D}\} \subseteq \mathbb{R}^L$. *Note that* $\mathcal{Z}_{\widehat{\Phi}} = \mathcal{Z}_\Phi (\triangleq \{\Phi(\boldsymbol{X}) : \boldsymbol{X} \in \mathbb{R}^{N \times D}\})$. *Then* $\widehat{\Phi} : \mathbb{R}_{>0}^{N \times D} \to \mathcal{Z}_{\widehat{\Phi}}$ *has inverse* $\widehat{\Phi}^{-1}$, *which is continuous.*

*Proof.* Since we constrain the image of $\widehat{\Phi}$ to be the range, $\widehat{\Phi}$ becomes surjective. Then invertibility is simply induced by injectivity (Lemma G.6).

Now it remains to show $\widehat{\Phi}^{-1}$ is continuous, which is done by verifying three conditions in Lemma 4.8. By Lemma C.6, any closed and bounded subset of $(\mathbb{R}_{>0}^{N \times D} / \sim, d_\Pi)$ is compact. Obviously, $\widehat{\Phi}(\boldsymbol{X})$ is continuous. And for condition **(c)** in Lemma 4.8, we will decompose $\widehat{\Phi}^{-1}$ into a series bounded mappings following the clue of proving its existence, similar to Theorem F.4. Recall each element in $\widehat{\Phi}(\boldsymbol{X})$ has the form $\sum_{n \in [N]} (\boldsymbol{x}_i)_n^{q-1} \exp(\log(\boldsymbol{X})\boldsymbol{\alpha}_j)_n^{p-q+1}$ for some $i \in [D], j \in [K_1], p \in [N], q \in [p+1]$ (cf. Eq. 45 in Lemma G.6). Hence, by Eq. 30 shown in Lemma G.1, $\widehat{\Phi}$ can be transformed into a sum-of-power mapping (cf. Definition D.1) via a (complex-valued) linear mapping: $\Psi_N(\boldsymbol{x}_i + \exp(\log(\boldsymbol{X})\boldsymbol{\alpha}_j)\sqrt{-1}) = \boldsymbol{\Upsilon}_{i,j}\widehat{\Phi}(\boldsymbol{X})$, where $\boldsymbol{\Upsilon}_{i,j} \in \mathbb{C}^{N \times L}$ for every $i \in [D], j \in [K_1]$. Let $K = DK_1$ and concatenate $\boldsymbol{\Upsilon}_{i,j}$: $\boldsymbol{\Upsilon} = [\cdots \quad \boldsymbol{\Upsilon}_{i,j} \quad \cdots] \in \mathbb{C}^{NK \times L}$. Recall $\mathcal{Z}_{\Psi_N} = \{\Psi_N(\boldsymbol{x}) : \boldsymbol{x} \in \mathbb{C}^N\} \subseteq \mathbb{C}^N$ denotes the range of the sum-of-power mapping.

Then we leverage the following decomposition to demonstrate the end-to-end boundedness:

$$(\mathcal{Z}_{\widehat{\Phi}}, d_\infty) \xrightarrow{\boldsymbol{\Upsilon}} \mathcal{O} \subseteq (\mathcal{Z}_{\Psi_N}^K, d_\infty) \xrightarrow{\widehat{\Psi_N}^\dagger} \mathcal{R} \subseteq (\mathbb{C}^N / \sim, d_\Pi)^K \xrightarrow{\pi} (\mathbb{R}_{>0}^{N \times D} / \sim, d_\Pi),$$

where $\widehat{\Psi_N}^\dagger$ is defined in Eq. 18 (Corollary D.6), $\mathcal{O}, \mathcal{R}$ are ranges of $\boldsymbol{\Upsilon}$ and $\widehat{\Psi_N}^\dagger$, respectively, and $\pi$ exists due to union alignment (cf. Eq. 46 and Lemma E.2). Therefore, for any $\boldsymbol{Z} \in \mathcal{R}$, there exists $\boldsymbol{X} \in (\mathbb{R}_{>0}^{N \times D} / \sim, d_\Pi)$ such that $\pi(\boldsymbol{Z}) \sim \boldsymbol{X}$. We denote $\boldsymbol{Z} = [\cdots \quad \boldsymbol{z}_{i,j} \quad \cdots], \forall i \in [D], j \in [K_1]$.

In the meanwhile, according to our construction of $\widehat{\Psi_N}^\dagger, \mathbf{\Upsilon}, \widehat{\Phi}$, we demonstrate the relationship between $\mathbf{Z}$ and $\pi(\mathbf{Z}) \sim \mathbf{X}$:

$$\mathbf{z}_{i,j} \sim \widehat{\Psi_N}^\dagger \left[ \mathbf{\Upsilon} \widehat{\Phi}(\mathbf{X}) \right]_{i,j} \sim \Psi_N^{-1} \left[ \mathbf{\Upsilon}_{i,j} \widehat{\Phi}(\mathbf{X}) \right] \sim \left( \mathbf{x}_i + \exp(\log(\mathbf{X})\boldsymbol{\alpha}_j)\sqrt{-1} \right) \qquad (47)$$
$$\forall i \in [D], j \in [K_1],$$

With this relationship, now we consider $\forall \mathbf{Z}, \mathbf{Z}' \in \mathcal{R}$ such that $d_\Pi^K(\mathbf{Z}, \mathbf{Z}') \leq C$ for some constant $C > 0$ and the product metric $d_\Pi^K$ (cf. Definition C.2), we have inequality:

$$d_\Pi(\pi(\mathbf{Z}), \pi(\mathbf{Z}')) = d_\Pi(\pi(\mathbf{Z})_{i^*}, \pi(\mathbf{Z}')_{i^*}) \leq \max_{j \in [K_1]} d_\Pi(\mathbf{z}_{i^*,j}, \mathbf{z}'_{i^*,j}) \leq d_\Pi^K(\mathbf{Z}, \mathbf{Z}') \leq C, \quad (48)$$

where $i^* \in [D]$ is the column which $\ell_{\infty,\infty}$-norm takes value at (cf. Definition C.3). Eq. 48 implies $\pi$ maps every bounded set in $\mathcal{R}$ to a bounded set in $(\mathbb{R}^{N \times D}/\sim, d_\Pi)$. Now we conclude the proof by the following chain of argument:

$$\mathcal{S} \subseteq (\mathcal{Z}_{\widehat{\Phi}}, d_\infty) \text{ is bounded} \xRightarrow{(*)} \mathbf{\Upsilon}(\mathcal{S}) \text{ is bounded} \xRightarrow{\text{Corollary D.6}}$$
$$\widehat{\Psi_N}^\dagger \circ \mathbf{\Upsilon}(\mathcal{S}) \text{ is bounded} \xRightarrow{\text{Eq. 48}} \pi \circ \widehat{\Psi_N}^\dagger \circ \mathbf{\Upsilon}(\mathcal{S}) \text{ is bounded},$$

where $(*)$ holds due to that $\mathbf{\Upsilon}$ is a finite-dimensional linear mapping. $\qquad \square$

## H  EXTENSION TO PERMUTATION EQUIVARIANCE

In this section, we prove Theorem 5.1, the extension of Theorem 3.1 to equivariant functions, following the similar arguments with Wang et al. (2023):

**Lemma H.1** (Wang et al. (2023); Sannai et al. (2019)). $f : \mathbb{R}^{N \times D} \to \mathbb{R}^N$ *is a permutation-equivariant function if and only if there is a function* $\tau : \mathbb{R}^{N \times D} \to \mathbb{R}$ *that is permutation invariant to the last $N - 1$ entries, such that* $f(\mathbf{Z})_i = \tau(\mathbf{z}^{(i)}, \underbrace{\mathbf{z}^{(i+1)}, \cdots, \mathbf{z}^{(N)}, \cdots, \mathbf{z}^{(i-1)}}_{N-1})$ *for any $i \in [N]$.*

*Proof.* (Sufficiency) Define $\pi : [N] \to [N]$ be an index mapping associated with the permutation matrix $\mathbf{P} \in \Pi(N)$ such that $\mathbf{P}\mathbf{Z} = \left[ \mathbf{z}^{(\pi(1))}, \cdots, \mathbf{z}^{(\pi(N))} \right]^\top$. Then we have:

$$f\left( \mathbf{z}^{(\pi(1))}, \cdots, \mathbf{z}^{(\pi(N))} \right)_i = \tau\left( \mathbf{z}^{(\pi(i))}, \mathbf{z}^{(\pi(i+1))}, \cdots, \mathbf{z}^{(\pi(N))}, \cdots, \mathbf{z}^{(\pi(i-1))} \right).$$

Since $\tau(\cdot)$ is invariant to the last $N - 1$ entries, it can shown that:

$$f(\mathbf{P}\mathbf{Z})_i = \tau\left( \mathbf{z}^{(\pi(i))}, \mathbf{z}^{(\pi(i+1))}, \cdots, \mathbf{z}^{(\pi(N))}, \cdots, \mathbf{z}^{(\pi(i-1))} \right) = f(\mathbf{Z})_{\pi(i)}.$$

(Necessity) Given a permutation-equivariant function $f : \mathbb{R}^{N \times D} \to \mathbb{R}^N$, we first expand it to the following form: $f(\mathbf{Z})_i = \tau_i(\mathbf{z}^{(1)}, \cdots, \mathbf{z}^{(N)})$. Permutation-equivariance means $\tau_{\pi(i)}(\mathbf{z}^{(1)}, \cdots, \mathbf{z}^{(N)}) = \tau_i(\mathbf{z}^{\pi(1)}, \cdots, \mathbf{z}^{\pi(N)})$ for any permutation mapping $\pi$. Suppose given an index $i \in [N]$, consider any permutation $\pi : [N] \to [N]$ such that $\pi(i) = i$. Then, we have:

$$\tau_i\left( \mathbf{z}^{(1)}, \cdots, \mathbf{z}^{(i)}, \cdots, \mathbf{z}^{(N)} \right) = \tau_{\pi(i)}\left( \mathbf{z}^{(1)}, \cdots, \mathbf{z}^{(i)}, \cdots, \mathbf{z}^{(N)} \right) = \tau_i\left( \mathbf{z}^{(\pi(1))}, \cdots, \mathbf{z}_i, \cdots, \mathbf{z}^{(\pi(N))} \right),$$

which implies $\tau_i : \mathbb{R}^{N \times D} \to \mathbb{R}$ must be invariant to the $N - 1$ elements other than the $i$-th element. Now, consider a permutation $\pi$ where $\pi(1) = i$. Then we have:

$$\tau_i\left( \mathbf{z}^{(1)}, \mathbf{z}^{(2)}, \cdots, \mathbf{z}^{(N)} \right) = \tau_{\pi(1)}\left( \mathbf{z}^{(1)}, \mathbf{z}^{(2)}, \cdots, \mathbf{z}^{(N)} \right) = \tau_1\left( \mathbf{z}^{(\pi(1))}, \mathbf{z}^{(\pi(2))}, \cdots, \mathbf{z}^{(\pi(N))} \right)$$
$$= \tau_1\left( \mathbf{z}^{(i)}, \mathbf{z}^{(i+1)}, \cdots, \mathbf{z}^{(N)}, \cdots, \mathbf{z}^{(i-1)} \right),$$

where the last equality is due to the invariance to $N - 1$ elements, stated beforehand. This implies two results. First, for all $i$, $\tau_i(\mathbf{z}^{(1)}, \mathbf{z}^{(2)}, \cdots, \mathbf{z}^{(i)}, \cdots, \mathbf{z}^{(N)}), \forall i \in [N]$ should be written in terms of $\tau_1(\mathbf{z}^{(i)}, \mathbf{z}^{(i+1)}, \cdots, \mathbf{z}^{(N)}, \cdots, \mathbf{z}^{(i-1)})$. Moreover, $\tau_1$ is permutation invariant to its last $N - 1$ entries. Therefore, we just need to set $\tau = \tau_1$ and broadcast it accordingly to all entries. We conclude the proof. $\qquad \square$

**Lemma H.2.** *Consider a continuous permutation-equivariant $f : (\mathbb{R}^{N \times D}, d_\Pi) \to (\mathbb{R}^N, d_\infty)$ and an associated $\tau : (\mathbb{R}^D, d_\infty) \times (\mathbb{R}^{(N-1) \times D}, d_\Pi) \to \mathbb{R}$ as specified in Lemma H.1, i.e., $f(\mathbf{Z})_i = \tau(\mathbf{z}^{(i)}, \mathbf{z}^{(i+1)}, \cdots, \mathbf{z}^{(N)}, \cdots, \mathbf{z}^{(i-1)})$ for any $\mathbf{Z} \in (\mathbb{R}^{N \times D}, d_\Pi)$ and $i \in [N]$. Then $\tau$ is continuous.*

*Proof.* Define $d_\tau(\mathbf{Z}, \mathbf{Z}') = \max\{d_\infty(\mathbf{z}^{(1)}, \mathbf{z}'^{(1)}), d_\Pi([\mathbf{z}^{(2)} \cdots \mathbf{z}^{(N)}]^\top, [\mathbf{z}'^{(2)} \cdots \mathbf{z}'^{(N)}]^\top)\}$ as the corresponding product metric of $(\mathbb{R}^D, d_\infty) \times (\mathbb{R}^{(N-1) \times D}, d_\Pi)$. Fix arbitrary $\mathbf{Z} \in \mathbb{R}^{N \times D}$ and $\epsilon > 0$. Since $f$ is continuous, by Definiton C.10, there exists $\delta > 0$ such that for every $\mathbf{Z}' \in \mathbb{R}^{N \times D}$ satisfying $d_\Pi(\mathbf{Z}, \mathbf{Z}') < \delta$, we have $d_\infty(f(\mathbf{Z}), f(\mathbf{Z}')) < \epsilon$. Then for the same $\delta$, consider every $\mathbf{Z}' \in \mathbb{R}^{N \times D}$, but under the $d_\tau$ metric, such that $d_\tau(\mathbf{Z}, \mathbf{Z}') < \delta$. We note that:

$$d_\Pi(\mathbf{Z}, \mathbf{Z}') = \min_{\mathbf{Q} \in \Pi(N)} \|\mathbf{Q}\mathbf{Z} - \mathbf{Z}'\|_{\infty,\infty} \leq \min_{\mathbf{Q} \in \Pi(N-1)} \left\| \begin{bmatrix} 1 & \\ & \mathbf{Q} \end{bmatrix} \mathbf{Z} - \mathbf{Z}' \right\|_{\infty,\infty} = d_\tau(\mathbf{Z}, \mathbf{Z}') < \delta.$$

Therefore, using the fact $d_\infty(\tau(\mathbf{Z}), \tau(\mathbf{Z}')) \leq d_\infty(f(\mathbf{Z}), f(\mathbf{Z}')) < \epsilon$, we conclude the proof. $\square$

The following result, restated from Theorem 5.1, can be derived from Theorem 3.1, equipped with Lemma H.1 and H.2.

**Theorem H.3** (Extension to Equivariance, Theorem 5.1). *For any permutation-equivariant function $f : \mathbb{R}^{N \times D} \to \mathbb{R}^N$, there exists continuous functions $\phi : \mathbb{R}^D \to \mathbb{R}^L$ and $\rho : \mathbb{R}^D \times \mathbb{R}^L \to \mathbb{R}$ such that $f(\mathbf{X})_j = \rho\left(\mathbf{x}^{(j)}, \sum_{i \in [N]} \phi(\mathbf{x}^{(i)})\right)$ for every $j \in [N]$, where $L \in [N(D+1), N^5 D^2]$ when $\phi$ admits LP architecture, and $L \in [ND, N^4 D^2]$ when $\phi$ admits LE architecture.*

*Proof.* Sufficiency can be shown by verifying the equivariance. We conclude the proof by showing the necessity with Lemma H.1. First we rewrite any permutation equivariant function $f(\mathbf{x}^{(1)}, \cdots, \mathbf{x}^{(N)}) : \mathbb{R}^{N \times D} \to \mathbb{R}^N$ as:

$$f\left(\mathbf{x}^{(1)}, \cdots, \mathbf{x}^{(N)}\right)_i = \tau\left(\mathbf{x}^{(i)}, \mathbf{x}^{(i+1)}, \cdots, \mathbf{x}^{(N)}, \cdots, \mathbf{x}^{(i-1)}\right), \tag{49}$$

where $\tau : \mathbb{R}^{N \times D} \to \mathbb{R}$ is invariant to the last $N - 1$ elements, according to Lemma H.1. By Lemma H.2, the continuity of $f$ suggests $\tau$ is also continuous. Given $\phi$ with either LP or LE architectures, $\Phi(\mathbf{X}) = \sum_{i=1}^N \phi(\mathbf{x}^{(i)}) \in \mathbb{R}^L$ is injective and has continuous inverse if:

- for some $L \in [N(D+1), N^5 D^2]$ when $\phi$ admits LP architecture (by Theorem F.3 and F.4).

- for some $L \in [ND, N^4 D^2]$ when $\phi$ admits LE architecture (by Theorem G.4 and G.5).

Let $\mathcal{Z}_\Phi = \{\sum_i \phi(\mathbf{x}^{(i)}) : \mathbf{X} \in \mathbb{R}^{N \times D}\} \subseteq \mathbb{R}^L$ be the range of the sum-pooling $\Phi$, and define mapping $\rho : \mathbb{R}^D \times \mathcal{Z}_\Phi \to \mathbb{R}$ taking the form of $\rho(\mathbf{x}, \mathbf{z}) = \tau(\mathbf{x}, \Phi^{-1}(\mathbf{z} - \phi(\mathbf{x})))$. It is straightforward to see that $\rho$ as a composition of continuous mappings, is also continuous. Finally, we show that function $\tau$ can be written in terms of $\rho$ by its invariance to last $N - 1$ elements, which concludes the proof:

$$\tau\left(\mathbf{x}^{(i)}, \mathbf{x}^{(i+1)}, \cdots, \mathbf{x}^{(N)}, \cdots, \mathbf{x}^{(i-1)}\right) = \tau\left(\mathbf{x}^{(i)}, \Phi^{-1} \circ \Phi(\mathbf{x}^{(i+1)}, \cdots, \mathbf{x}^{(N)}, \cdots, \mathbf{x}^{(i-1)})\right)$$

$$= \tau\left(\mathbf{x}^{(i)}, \Phi^{-1}\left(\Phi\left(\mathbf{x}^{(1)}, \cdots, \mathbf{x}^{(N)}\right) - \phi(\mathbf{x}^{(i)})\right)\right)$$

$$= \rho\left(\mathbf{x}^{(i)}, \sum_{i=1}^N \phi(\mathbf{x}^{(i)})\right)$$

$\square$

# I EXTENSION TO COMPLEX NUMBERS

In this section, we formally introduce the nature extension of our Theorem 3.1 to the complex numbers:

**Corollary I.1** (Extension to Complex Domain). *For any permutation-invariant function $f$ : $\mathbb{C}^{N \times D} \to \mathbb{C}$, there exists continuous functions $\phi : \mathbb{C}^D \to \mathbb{R}^L$ and $\rho : \mathbb{R}^L \to \mathbb{C}$ such that $f(\boldsymbol{X}) = \rho\left(\sum_{i \in [N]} \phi(\boldsymbol{x}^{(i)})\right)$ for every $j \in [N]$, where $L \in [2N(D+1), 4N^5 D^2]$ when $\phi$ admits LP architecture, and $L \in [2ND, 4N^4 D^2]$ when $\phi$ admits LE architecture.*

*Proof.* We let $\phi$ first map complex features $\boldsymbol{x}^{(i)} \in \mathbb{C}^D, \forall i \in [N]$ to real features $\widetilde{\boldsymbol{x}}^{(i)} = \left[\Re(\boldsymbol{x}^{(i)})^\top \quad \Im(\boldsymbol{x}^{(i)})^\top\right] \in \mathbb{R}^{2D}, \forall i \in [N]$ by divide the real and imaginary parts into separate channels, then utilize either LP or LE embedding layer to map $\widetilde{\boldsymbol{x}}^{(i)}$ to the latent space. The upper bounds of desired latent space dimension are scaled by $4$ for both architectures due to the quadratic dependence on $D$. Then the same proofs of Theorems F.3, F.4, G.4, and G.5 applies. □

It is also straightforward to extend this result to the permutation-equivariant case:

**Corollary I.2.** *For any permutation-equivariant function $f : \mathbb{C}^{N \times D} \to \mathbb{C}^N$, there exists continuous functions $\phi : \mathbb{C}^D \to \mathbb{R}^L$ and $\rho : \mathbb{R}^D \times \mathbb{R}^L \to \mathbb{C}$ such that $f(\boldsymbol{X})_j = \rho\left(\boldsymbol{x}^{(j)}, \sum_{i \in [N]} \phi(\boldsymbol{x}^{(i)})\right)$ for every $j \in [N]$ for every $j \in [N]$, where $L \in [2N(D+1), 4N^5 D^2]$ when $\phi$ admits LP architecture, and $L \in [2ND, 4N^4 D^2]$ when $\phi$ admits LE architecture.*

*Proof.* Combine the proof of Corollary I.1 with Theorem H.3. □

## J    THEORETICAL CONNECTION TO UNLABELED SENSING

Unlabeled sensing (Unnikrishnan et al., 2018), also known as linear regression without correspondence (Hsu et al., 2017; Tsakiris & Peng, 2019; Tsakiris et al., 2020; Peng & Tsakiris, 2020), solves $\boldsymbol{x} \in \mathbb{R}^N$ in the following linear system:

$$\boldsymbol{y} = \boldsymbol{P}\boldsymbol{A}\boldsymbol{x} \quad \text{or} \quad \min_{\boldsymbol{x}, \boldsymbol{P}} \|\boldsymbol{y} - \boldsymbol{P}\boldsymbol{A}\boldsymbol{x}\|_2^2, \tag{50}$$

where $\boldsymbol{A} \in \mathbb{R}^{M \times N}$ is a given measurement matrix, $\boldsymbol{P} \in \Pi(M)$ is an unknown permutation, and $\boldsymbol{y} \in \mathbb{R}^M$ is the measured data. The results in Unnikrishnan et al. (2018); Tsakiris & Peng (2019) show that as long as $\boldsymbol{A}$ is over-determinant ($M \geq 2N$), such problem is well-posed (i.e., has a unique solution) for almost all cases. Unlabeled sensing shares the similar structure with our LP embedding layers in which a linear layer lifts the feature space to a higher-dimensional ambient space, ensuring the solvability of alignment across each channel. Specifically, as revealed in Theorem F.3, showing the injectivity of the LP layer is to establish the argument:

$$\boldsymbol{X}\boldsymbol{w}_i \sim \boldsymbol{X}'\boldsymbol{w}_i, \forall i \in [K] \Rightarrow \boldsymbol{X} \sim \boldsymbol{X}', \tag{51}$$

for arbitrary $\boldsymbol{X}, \boldsymbol{X}' \in \mathbb{R}^{N \times D}$, constructed weights $\boldsymbol{w}_i, i \in [K]$, and large enough $K$. Whereas, to show the well-posedness of unlabeled sensing, it is to show the following statement (Tsakiris & Peng, 2019):

$$\boldsymbol{A}\boldsymbol{x} \sim \boldsymbol{A}\boldsymbol{x}' \Rightarrow \boldsymbol{x} = \boldsymbol{x}', \tag{52}$$

for sufficiently many measurements $M$. We note that our bijectivity is defined between the set and embedding spaces, which allows a change of order in the results and differs from exact recovery of unknown variables expected in unlabeled sensing.

In fact, the well-posedness of unlabeled PCA (Yao et al., 2021), studying low-rank matrix completion with shuffle perturbations, shares the identical definition with our set function injectivity. We rephrase it as below:

$$\boldsymbol{x}_i \sim \boldsymbol{x}'_i, \forall i \in [N], \boldsymbol{X}, \boldsymbol{X}' \in \mathcal{M} \Rightarrow \boldsymbol{X} \sim \boldsymbol{X}', \tag{53}$$

where $\mathcal{M} = \{\boldsymbol{X} \in \mathbb{R}^{M \times N} : \text{rank}(\boldsymbol{X}) < r\}$ is a set of low-rank matrices, and $\boldsymbol{x}_i$ denotes the $i$-th column of $\boldsymbol{X}$. Based on Theorem 1 in Yao et al. (2021), we can obtain the following results:

**Lemma J.1.** *Suppose $\mathcal{M} = \{\boldsymbol{X} \in \mathbb{R}^{N \times K} : \text{rank}(\boldsymbol{X}) \leq r\}$ with $r < \min\{N, K\}$. Then there exists an open dense set $\mathcal{U} \subset \mathcal{M}$ such that for every $\boldsymbol{X}, \boldsymbol{X}' \in \mathcal{U}$ such that $\boldsymbol{x}_i \sim \boldsymbol{x}'_i, \forall i \in [K]$, then $\boldsymbol{X} \sim \boldsymbol{X}'$.*

*Proof.* According to Theorem 1 in Yao et al. (2021), there exists a Zariski-open dense set $\mathcal{U} \subset \mathcal{M}$ such that: for every $\boldsymbol{X} \in \mathcal{U}$ and $\boldsymbol{P}_i \in \Pi(N), \forall i \in [K]$, $\text{rank}([\boldsymbol{P}_1 \boldsymbol{x}_1 \quad \cdots \quad \boldsymbol{P}_K \boldsymbol{x}_K]) \geq r$. And moreover, $\text{rank}([\boldsymbol{P}_1 \boldsymbol{x}_1 \quad \cdots \quad \boldsymbol{P}_K \boldsymbol{x}_K]) = r$ if and only if $\boldsymbol{P}_1 = \cdots = \boldsymbol{P}_K = \boldsymbol{P} \in \Pi(N)$.

For every $\boldsymbol{X}, \boldsymbol{X}' \in \mathcal{U}$ such that $\boldsymbol{x}_i \sim \boldsymbol{x}'_i, \forall i \in [K]$, if $\boldsymbol{X} \not\sim \boldsymbol{X}'$, then either $\text{rank}(\boldsymbol{X}) > r$ or $\text{rank}(\boldsymbol{X}') > r$. This contradicts the fact that $\boldsymbol{X}, \boldsymbol{X}' \in \mathcal{U} \subset \mathcal{M}$. $\qquad\square$

As a result, we can establish the injectivity of an LP layer restricted to a dense set.

**Theorem J.2.** *Assume $D < N$. Suppose $\phi : \mathbb{R}^D \to \mathbb{R}^L$ takes the form of an LP embedding layer (Eq. 3):*

$$\phi(\boldsymbol{x}) = \begin{bmatrix} \psi_N(\boldsymbol{w}_1^\top \boldsymbol{x})^\top & \cdots & \psi_N(\boldsymbol{w}_K^\top \boldsymbol{x})^\top \end{bmatrix}, \tag{54}$$

*where $K = D+1$, $L = NK = N(D+1)$, and $\boldsymbol{W} = [\boldsymbol{e}_1 \quad \cdots \quad \boldsymbol{e}_D \quad \boldsymbol{w}] \in \mathbb{R}^{D \times K}$. There exists a open dense subset $\mathcal{V} \subseteq \mathbb{R}^{N \times D}$ such that for any $\boldsymbol{X}, \boldsymbol{X}' \in \mathcal{V}$, $\sum_{n \in [N]} \phi(\boldsymbol{x}^{(n)}) = \sum_{n \in [N]} \phi(\boldsymbol{x}'^{(n)})$ implies $\boldsymbol{X} \sim \boldsymbol{X}'$.*

*Proof.* Define $\mathcal{M} = \{\boldsymbol{X} \in \mathbb{R}^{N \times K} : \text{rank}(\boldsymbol{X}) \leq D\}$. Since $\boldsymbol{W}$ has full row rank, then $\tau(\boldsymbol{X}) = \boldsymbol{X}\boldsymbol{W} : \mathbb{R}^{N \times D} \to \mathcal{M}$ is surjective. Let $\mathcal{V} = \{\boldsymbol{X} \in \mathbb{R}^{N \times D} : \tau(\boldsymbol{X}) \in \mathcal{U}\}$ be the preimage of $\mathcal{U}$ under $\tau$. Since $\mathcal{U}$ is open dense in $\mathcal{M}$, then $\mathcal{V}$ is open dense in $\mathbb{R}^{N \times D}$. So far we have found an open dense set $\mathcal{V}$ such that for all $\boldsymbol{X}, \boldsymbol{X}' \in \mathcal{V}$, $\boldsymbol{X}\boldsymbol{W}, \boldsymbol{X}'\boldsymbol{W} \in \mathcal{U}$. By Lemma D.2, $\sum_{n \in [N]} \phi(\boldsymbol{x}^{(n)}) = \sum_{n \in [N]} \phi(\boldsymbol{x}'^{(n)})$ implies $\boldsymbol{X}\boldsymbol{w}_i \sim \boldsymbol{X}'\boldsymbol{w}_i$ for ever $i \in [K]$. By Lemma J.1, it can induce $\boldsymbol{X}\boldsymbol{W} \sim \boldsymbol{X}'\boldsymbol{W}$, and namely $\boldsymbol{X} \sim \boldsymbol{X}'$. $\qquad\square$

Theorem J.2 gives a much tighter upper bound on the dimension of the embedding space, which is bilinear in $N$ and $D$. However, it is noteworthy that this result is subject to the scenario where the input feature dimension is smaller than the set length, and the feature space is restricted to *a dense subset of the ambient space*. Moreover, it is intractable to establish the continuity over such dense set. Our Theorem F.3 dismisses this denseness condition, serving as a stronger results in considering *all possible inputs*. This indicates Theorem F.3 could potentially bring new insights into the field of unlabeled sensing, which may be of an independent interest.

## K  REMARK ON AN ERROR IN PROPOSITION 3.10 IN FEREYDOUNIAN ET AL. (2022)

Fereydounian et al. examine the expressiveness of GNNs with a mathematical tool summarized in Proposition 3.10, which in turn seems to indicate a much tighter upper bound $ND^2$ for the size of the embedding space for set representation. However, as we will show later, their proof might be deficient, or at least incomplete in the assumptions.

We rephrase their Proposition 3.10 in our language as below:

**Claim K.1** (An incorrect claim). *Suppose $\phi : \mathbb{R}^D \to \mathbb{R}^L$ where $L = N^2 D$ takes the following form:*

$$\phi(\boldsymbol{x})_{i,j,l} = \begin{cases} \Re\left((\boldsymbol{x}_i + \boldsymbol{x}_j \sqrt{-1})^l\right), & i > j \\ \Im\left((\boldsymbol{x}_i + \boldsymbol{x}_j \sqrt{-1})^l\right), & i \leq j \end{cases}, \tag{55}$$

*for every $i, j \in [D], l \in [N]$. Then $\sum_{n \in [N]} \phi(\boldsymbol{x}^{(n)})$ is injective.*

The authors' proof technique can be illustrated via the following chain of arguments: for every $\boldsymbol{X}, \boldsymbol{X}' \in \mathbb{R}^{N \times D}$,

$$\sum_{n \in [N]} \phi(\boldsymbol{x}^{(n)}) = \sum_{n \in [N]} \phi(\boldsymbol{x}'^{(n)}) \xRightarrow{\text{Lemma D.2}} (\boldsymbol{x}_i + \boldsymbol{x}_j \sqrt{-1}) \sim (\boldsymbol{x}'_i + \boldsymbol{x}'_j \sqrt{-1}), \forall i, j \in [D] \tag{56}$$

$$\Rightarrow [\boldsymbol{x}_i \quad \boldsymbol{x}_j] \sim [\boldsymbol{x}'_i \quad \boldsymbol{x}'_j], \forall i, j \in [D] \xRightarrow{(*)} \boldsymbol{X} \sim \boldsymbol{X}'. \tag{57}$$

While the first two steps is correct, the last implication $(*)$ is not true in general unless one of $\boldsymbol{x}_i, i \in [D]$ happens to be an anchor of $\boldsymbol{X}$. We formally disprove this argument below.

Consider $\boldsymbol{X}, \boldsymbol{X}' \in \mathbb{R}^{N \times D}$ and let $\mathcal{P}_i = \{\boldsymbol{P} \in \Pi(N) : \boldsymbol{P} \boldsymbol{x}'_i = \boldsymbol{x}_i\}$. Then $[\boldsymbol{x}_i \quad \boldsymbol{x}_j] \sim [\boldsymbol{x}'_i \quad \boldsymbol{x}'_j]$ for every $i, j \in [D]$ is equivalent to saying $\mathcal{P}_i \cap \mathcal{P}_j \neq \emptyset, \forall i, j \in [D]$. While $\boldsymbol{X} \sim \boldsymbol{X}'$ is identical to $\bigcap_{i \in [D]} \mathcal{P}_i \neq \emptyset$. It is well-known that intersection between each pair of sets is non-empty cannot necessarily imply the intersection among all sets is non-empty, i.e., $\mathcal{P}_i \cap \mathcal{P}_j \neq \emptyset, \forall i, j \in [D] \not\Rightarrow \bigcap_{i \in [D]} \mathcal{P}_i \neq \emptyset$, which disproves this result.

This also reveals the significance of the our defined anchor. Suppose $\boldsymbol{x}_{i^*}$ for some $i^* \in [D]$ is an anchor of $\boldsymbol{X}$. Then by Lemma E.2, $\bigcap_{i \in [D]} \mathcal{P}_i = \mathcal{P}_{i^*}$. Thus $\mathcal{P}_{i^*} \cap \mathcal{P}_j \neq \emptyset$ for every $j \in [D]$ implies $\mathcal{P}_{i^*} \neq \emptyset$, which essentially says $\bigcap_{i \in [D]} \mathcal{P}_i \neq \emptyset$.

Specifically, we can construct a counter-example. Suppose $\boldsymbol{X} = [\boldsymbol{x}_1 \quad \boldsymbol{x}_2 \quad \boldsymbol{x}_3], \boldsymbol{X}' = [\boldsymbol{x}'_1 \quad \boldsymbol{x}'_2 \quad \boldsymbol{x}'_3]$ take values as below,

$$
\boldsymbol{x}_1 = \boldsymbol{x}'_1 = \begin{bmatrix} 1 \\ 1 \\ 2 \\ 2 \end{bmatrix}, \boldsymbol{x}_2 = \boldsymbol{x}'_2 = \begin{bmatrix} 1 \\ 2 \\ 1 \\ 2 \end{bmatrix}, \boldsymbol{x}_3 = \begin{bmatrix} 1 \\ 2 \\ 2 \\ 1 \end{bmatrix}, \boldsymbol{x}'_3 = \begin{bmatrix} 2 \\ 1 \\ 1 \\ 2 \end{bmatrix}, \tag{58}
$$

and we can see:

$$
\begin{bmatrix} 1 & 1 \\ 1 & 2 \\ 2 & 1 \\ 2 & 2 \end{bmatrix} = \begin{bmatrix} 1 & 0 & 0 & 0 \\ 0 & 1 & 0 & 0 \\ 0 & 0 & 1 & 0 \\ 0 & 0 & 0 & 1 \end{bmatrix} \begin{bmatrix} 1 & 1 \\ 1 & 2 \\ 2 & 1 \\ 2 & 2 \end{bmatrix} \Rightarrow [\boldsymbol{x}_1 \quad \boldsymbol{x}_2] \sim [\boldsymbol{x}'_1 \quad \boldsymbol{x}'_2], \tag{59}
$$

$$
\begin{bmatrix} 1 & 1 \\ 1 & 2 \\ 2 & 2 \\ 2 & 1 \end{bmatrix} = \begin{bmatrix} 0 & 1 & 0 & 0 \\ 1 & 0 & 0 & 0 \\ 0 & 0 & 0 & 1 \\ 0 & 0 & 1 & 0 \end{bmatrix} \begin{bmatrix} 1 & 2 \\ 1 & 1 \\ 2 & 1 \\ 2 & 2 \end{bmatrix} \Rightarrow [\boldsymbol{x}_1 \quad \boldsymbol{x}_3] \sim [\boldsymbol{x}'_1 \quad \boldsymbol{x}'_3], \tag{60}
$$

$$
\begin{bmatrix} 1 & 1 \\ 2 & 2 \\ 1 & 2 \\ 2 & 1 \end{bmatrix} = \begin{bmatrix} 0 & 0 & 1 & 0 \\ 0 & 0 & 0 & 1 \\ 1 & 0 & 0 & 0 \\ 0 & 1 & 0 & 0 \end{bmatrix} \begin{bmatrix} 1 & 2 \\ 2 & 1 \\ 1 & 1 \\ 2 & 2 \end{bmatrix} \Rightarrow [\boldsymbol{x}_2 \quad \boldsymbol{x}_3] \sim [\boldsymbol{x}'_2 \quad \boldsymbol{x}'_3]. \tag{61}
$$

However, notice that:

$$
\begin{bmatrix} 1 & 1 & 1 \\ 1 & 2 & 2 \\ 2 & 1 & 2 \\ 2 & 2 & 1 \end{bmatrix} \not\sim \begin{bmatrix} 1 & 1 & 2 \\ 1 & 2 & 1 \\ 2 & 1 & 1 \\ 2 & 2 & 2 \end{bmatrix} \Rightarrow \boldsymbol{X} = [\boldsymbol{x}_1 \quad \boldsymbol{x}_2 \quad \boldsymbol{x}_3] \not\sim \boldsymbol{X}' = [\boldsymbol{x}'_1 \quad \boldsymbol{x}'_2 \quad \boldsymbol{x}'_3], \tag{62}
$$

which contradicts the implication $(*)$ in Claim K.1.

