# OpenReview forum: "Polynomial Width is Sufficient for Set Representation with High-dimensional Features"
_ICLR.cc/2024/Conference — ICLR 2024 poster_

### Official Review · Reviewer_oY1k · 2023-10-31

**Soundness:** 4 excellent
**Presentation:** 3 good
**Contribution:** 3 good
**Rating:** 8
**Confidence:** 3

**Summary:**

The paper considers the representational powers of the permutation-invariant DeepSets architecture, which maps inputs $X \in \mathbb{R}^{N \times D}$ to features $\phi(X) \in \mathbb{R}^{N \times L}$, computes a sum, and then computes an output with $\rho: \mathbb{R}^L \to \mathbb{R}$. Past results have shown that the DeepSets architecture can represent any permutation-invariant continuous function when $D = 1$ as long as $L \geq D$, and that $L \geq \exp(\Omega(\min(\sqrt{N}, D)))$ is necessary under certain assumptions about the $\phi$ and $\rho$ being implementable with analytic activation functions. This work shows that without these activation restrictions, permutation-invariant continuous functions can be exactly represented using embedding dimension $L = \text{poly}(N, D)$.

They do so by introducing a pair of positive results in Theorem 3.1, one having a power mapping activation (where each input is mapped to its powers up to degree $N$) and another having an exponential activation. The constructions use a similar proof structure, both of which involve finding $\phi$ that that the function $X \mapsto \sum_i \phi(x_i)$ is continuous and invertible, which makes it possible to invert the mapping in $\rho$ and then apply the permutation-invariant function directly. (Note that this means that, while $\phi$ is a simple construction that can be made explicit, the inversion of $\rho$ is an existential result that may correspond to a highly non-smooth function.) The feature mapping $\phi$ is constructed carefully to ensure that inputs that are element-wise permutation-invariant but *not* vector-wise permutation-invariant map to different features; the authors do so by using *anchor* vectors to distinguish each vector. The remainder of the construction is dedicated to ensuring that some anchor exists for every possible input and that the desired constraints are enforced.

They contextualize their results by including several negative results to illuminate their design decisions, including Lemma 3.4 (why $\rho$ and $\phi$ must be continuous) and Theorem 4.5 (why there must be at least $D$ candidate anchor vectors for each input to have an anchor). They run brief experiments in Appendix K that train DeepSets models to compute lexicographical medians, and find that the necessary width $L$ grows polynomially in $N$ and $D$.

**Strengths:**

The contrast to the Zweig and Bruna lower bound is of theoretical interest, since it shows that requiring analytic activation functions changes the minimum width cost from polynomial in $N$ and $D$ to exponential. The work, coupled with its experimental results, suggests that an impressive amount of information can be encoded in sums of relatively low-dimensional feature embeddings, and that the functions that cannot be represented by standard DeepSets architectures are likely highly pathological or discontinuous.

The results are technically impressive, and I found no major errors in the proofs. In particular, the LP and LE constructions were clever, meticulous, and well-visualized by figures.

**Weaknesses:**

While I don't expect this paper to solve the problem, the non-explicitness of the construction means that $\rho$ is likely to be a highly non-smooth function that is difficult to compactly approximate using a neural network architecture. In future works, I'd be interested in understanding whether feature dimensions that are larger polynomials in $N$ and $D$ make it possible to yield explicit (and ideally more smooth) $\rho$ mappings. Perhaps the work could discuss possible approaches to bounding the smoothness of inversion functions? Or perhaps the authors can discuss which kinds of permutation-invariant functions $f$ are expected to have smooth features?

## Minor issues
* I was momentary confused by the conditions on $L$ in Theorem 3.1. At first, I thought $L$ *could not* be larger than $N^5D^2$ and not that this is an upper bound on the smallest $L$. Perhaps the lower bounds could be mentioned separately, for the sake of clarity?
* Page 17 says "pigeon-hold" instead of "pigeon-hole."
* Lemma E.1 is much easier to parse after noting Remark E.2. Perhaps the remark could be included before the proof, or maybe the lemma could just define the $\gamma_i$ values explicitly as prime numbers, since a generality in $\gamma_i$ is not necessary for any of the proofs?

**Questions:**

N/A

---

> ### Author Response · Authors · 2023-11-21
> **Response to Reviewer oY1k**
>
> We thank Reviewer oY1k for the detailed summary of our results and acknowledging our theoretical significance. Regarding your questions, please see our response below:
>
> **1. Brief discussion on $\rho$.**
>
> Fine-grained analysis on $\rho$ remains a widely open question in this area. We agree with Reviewer oY1k that $\rho$ is highly non-smooth and hard to learn [1]. Nevertheless, we hypothesize that modern ML algorithms can approximate non-smooth inversion of the sum-pooling well. This argument is also supported by our experiments in Appendix K. We doubt that non-smooth points in a generic set are often located at the regime of no interest, meaning worst-case may not occur in practical scenarios. We also agree a rigorous study demystifying the $\rho$ mapping is very necessary and the suggested direction of bounding the smoothness of the inversion function is of high interest to explore in the future. Our results can be easily extended to equivariant functions, as a foundation to form multi-layer set functions. Under this scenario, $\phi$ may be simplified.
>
> [1] Murphy et al., Janossy Pooling: Learning Deep Permutation-Invariant Functions for Variable-Size Inputs
>
> **2. Presentation issues.**
>
> We have revised our introduction to clarify the bound of $L$ means the upper bound of the smallest $L$, and fixed all the typos pointed out.
>
> **3. Questions on Lemma E.1.**
>
> It is a good point to mention $\gamma_i$ can be chosen as prime numbers in Lemma E.1. But we would remark that the non-divisibility is the most crucial property to let Lemma E.1 hold, which allows for any other choices of $\gamma_i$ satisfying the condition in Lemma E.1.

---

> > ### Comment · Reviewer_oY1k · 2023-11-23
> >
> > I thank the authors for their detailed responses. I maintain my score and believe that the contributions are substantive and technically interesting.

---

### Official Review · Reviewer_jrmV · 2023-11-01

**Soundness:** 3 good
**Presentation:** 2 fair
**Contribution:** 3 good
**Rating:** 8
**Confidence:** 3

**Summary:**

This work concerns the expressive power of the DeepSets architecture [Zaheer et al., 2017], which is designed to model permutation invariant functions taking as input *sets* whose elements are real vectors. The basic parameters relevant for the model are N and D; N denotes the size of the input set and D denotes the dimension of its elements. The novelty of this work is in proving a poly(N,D) upper bound on the latent dimension sufficient for DeepSets to express *any* permutation-invariant function. Previously known bounds were exponential in the relevant parameters N and D.

To elaborate, the DeepSets architecture is of the form $g(x) = \rho(\sum_{i=1}^N \phi(x^i))$, where $\phi: \mathbb{R}^D \to \mathbb{R}^L$ is a feature map for the set elements $x^i \in \mathbb{R}^D$, and $\rho: \mathbb{R}^L \to \mathbb{R}$ is an activation function. Both $\phi$ and $\rho$ are chosen by the model designer. Writing $\Phi(X) = \sum_{i=1}^N \phi(x^i)$, where $X \in \mathbb{R}^{N \times D}$ we can view $\Phi$ a “sum-pooled” feature map for the input X. A key question for the practitioner is “How large should I set the latent dimension L to model *any* permutation-invariant target function?” This work shows that L = poly(N, D) is both sufficient and necessary (the bounds are not tight, however). Previously known upper bounds on L were exponential in N and D, thus this work represents a significant improvement over the previous state-of-the-art in theory.

**Strengths:**

The paper presents novel ideas in designing the feature map $\Phi : \mathbb{R}^{N \times D} \to \mathbb{R}^L$ to overcome limitations of previous work which did not generalize beyond the scalar-elements case (i.e., D=1). The combinatorial argument presented in the proof of Lemma 4.4 is particularly nice, though it is somewhat difficult to follow in the current exposition. Overall, I find the proofs insightful and quite surprising (I will elaborate on this below). Hence, I am inclined to accepting this paper.

The proof of the main result can be understood in stages. Let $\Phi(X) = \sum_{i=1}^N \phi(x^i)$, where $x^i$ denotes D-dimensional vectors which are elements of the given “set” X. The ultimate goal is to design $\Phi : \mathbb{R}^{N \times D}$ such that $\Phi(X) = \Phi(X’)$ implies $X \sim X’$, where $A \sim B$ means that the two matrices are equivalent up to row permutations (the opposite implication is obvious). The key difficulty is in “surviving” the sum-pooling operation $\sum_{i=1}^N$.

Ideas from previous work, such as the degree-N polynomial mapping $\psi(z) = (z, z^2, z^3, \ldots, z^N)$, can be applied entrywise to ensure that $\Phi(X) = \Phi(X’)$ implies that the rows of X and X’ are equivalent *individually*, which is a weaker implication than $X \sim X’$. This only works only in the D=1 case and not for D > 1 since $X \sim X’$ requires the coordinates of the rows $x^i$ be *jointly* aligned. To overcome this limitation, the authors propose novel ideas in the design of $\Phi$ so that alignment between the coordinates of $x^i$ are ensured as well. Personally, I found this issue of coordinate-wise alignment quite challenging and was pleased to see its resolution here.

**Weaknesses:**

One shortcoming of this paper is that the exposition is quite hard to follow. It would help this paper reach a wider audience if the authors improved their exposition. I would suggest using less confusing notation and providing a more structured and detailed proof overview. Specific examples include:

- The proof of their main theorem can be presented in stages as follows “Suppose $\Phi(X) = \Phi(X’)$. The polynomial mapping ensures that the *rows* of X and X’, when viewed as multi-sets, are equal. The main technical challenge is to ensure that the coordinates of the rows are aligned as well. This step combines two ideas: the “anchors” and the set $\Gamma$, representing the additional linear mappings that ensure *pairwise* alignment …”
- The linear projections that form the coordinates of $\phi(x^i)$ are all represented as $w_j$’s. It would be more informative if different symbols were used for the different “types” of these mappings (standard basis, anchors, \gamma’s …).
- In p.7, the superscript notation for $w$ is confusing. For the samples, the superscript “(n)” was used to index elements of the set. Here it is used to denote different “types” of w. As mentioned before, simply using different symbols for different groups would avoid the use of unnecessary superscripts.

**Questions:**

- One of the key technical proof ideas is showing that there exists some set of real numbers $\Gamma$ s.t. for any $a,a’,b,b’ \in \mathbb{R}^N$, if $a \sim a’$, $b \sim b’$, and $(a-\gamma b) \sim (a’ - \gamma b’)$ for all $\gamma \in \Gamma$, then $[a, b] \sim [a’, b’]$ (here, we view [a b] as “row $a$ stacked on top of row $b$”). Is the idea of using such linear transformations (of the form $a - \gamma b$) to ensure alignment between the coordinates new? Or is this “linear coupling” a well-known fact in mathematics?

---

> ### Author Response · Authors · 2023-11-21
> **Response to Reviewer jrmV**
>
> We sincerely appreciate Reviewer jrmV's constructive advice to improve our paper's clarity. We have carefully revised the text and adopted a more informative notation system. Please check our updated version.
>
> **1. The exposition can be improved.**
>
> We have revised our high-level description of the proof sketch in Sec. 4 to enhance clarity. We also improved our notations in the following way: $w_i^{(1)}$ -> $e_i$ for canonical basis, $w_j^{(2)} \rightarrow \alpha_j$ for anchor construction, and $w_{i,j,k}^{(3)} \rightarrow \Gamma_{i,j,k}$ for linear coupling.
>
> **2. Question regarding linear coupling.**
>
> Sorry for the confusion. $a$ and $b$ here are both column vectors, and we stack them to form an $N$ by 2 matrices. The form of linear coupling may have appeared in other proof techniques, but to our best knowledge, it is the first time used to constrain the permutation orbits of matrices to ensure alignment.

---

### Official Review · Reviewer_a75R · 2023-11-02

**Soundness:** 3 good
**Presentation:** 3 good
**Contribution:** 3 good
**Rating:** 8
**Confidence:** 3

**Summary:**

The authors propose a theoretical bound for the size of the embedding dimension of permutation invariant set functions which is not restricted to a single feature dimension, as some previous works.

**Strengths:**

- The problem is very relevant, as many works which utilize permutation invariant functions do not consider the importance of the embedding dimension.
- The derivation in the text appears thorough and rigorous

**Weaknesses:**

- It took me a while to grasp the concept of anchors. In lemma 4.2 I think it needs to be stressed that the same anchor is being applied over all the channels. Although the notation states this, it would be good to state it in plain text as well.
- Under Lemma 4.2 there is a sentence which says "then once each coupled pairs are aligned, two data matrices are globally aligned." Can you elaborate on the precise meaning of this sentence? What does "aligned" and "globally aligned" signify?
- Why is Lemma 4.4 necesary? I do not see the reason for this, and I do not think it is explained well in the text either. At the beginning of page 8, it is stated that: "Eq. 5 implies Eq. 6, but none of these seem to depend on Lemma 4.4 or "Contruction point #3" so I am not sure why it is necessary. I think this needs to be explained better in the text.

**Questions:**

- Right before section 2.2, it is stated: "The obtained results for D′ = 1 can also be easily extended to D′ > 1 as
otherwise f can be written as [f1 · · · fD′ ]⊤ and each fi has single output feature channel." I assume this is referring to the previous sentence and means "the results obtained for invariance can be extended to equivariance, as.." Is this correct?

I would be curious to hear the authors opinion on the following:

According to this and prior theoretical works, a very large embedding dimension $L$ is needed to maintain the universal function approximation ability of permutation invariant functions, however, many practical works which utilize permutation invariant set functions do not use such a large embedding dimension. Therefore, what is the practical takeaway for an implementation which wishes to utilize a permutation invariant function? There seems to be quite a large disconnect between theory and practice on this topic.

---

Overall, my biggest conflict with this work is that there is no empirical experiment to corroborate the theoretical findings presented in the paper. While I cannot find issue with any of the claims made in the paper, if there is no way to empirically verify the given bounds, then it is quite difficult to understand their practical significance. Therefore, if the authors could provide an experiment or further discussion which illuminates this topic, I would be happy to revisit my current score.

---

> ### Author Response · Authors · 2023-11-21
> **Response to Reviewer a75R**
>
> We thank Reviewer a75R for acknowledging our mathematical rigor and high relevance with ML community. For your questions, please see our response below:
>
> **1. Confusion by Lemma 4.2 and its implication.**
>
> Thanks to Reviewer a75R's suggestion, we have revised the manuscript with an emphasis on "the same anchor is being applied over all the channels". The terminology "alignment" between $X$ and $X'$ refers to $X \sim X'$. Hence, our argument "once each coupled pairs are aligned, two data matrices are globally aligned" can be rephrased in a more rigorous way: "once the permutation orbits of each coupled pair intersect, the permutation orbits of the two data matrices also intersect", and can be translated into our mathematical statement: $[x_i, a] \sim [x'_i, a'] \Rightarrow X \sim X'$.
>
>
> **2. Why is Lemma 4.4 necessary?**
>
> Proof in Sec. 4.1.2 proceeds in the following way:
> 1) Sum-pooling $\sum_n \phi(x^{(n)}) = \sum_n \phi(x'^{(n)})$ implies column-wisely $Xw_i \sim X'w_i, \forall i \in [K]$  by injectivity of power mapping, which induces $(x_i - \gamma_k Xw^{(2)}_j) \sim (x'_i - \gamma_k X' w^{(2)}_j), \forall i,j,k$.
> 2) **By Lemma 4.4**, column-wisely $(x_i - \gamma_k Xw^{(2)}_j) \sim (x'_i - \gamma_k X' w^{(2)}_j), \forall i,j,k$ implies $[x_i, Xw^{(2)}_j] \sim [x'_j, X'w^{(2)}_j]$ for any $i,j$.
> 3) By union alignment (Lemma 4.2), $[x_i, Xw^{(2)}_j] \sim [x'_j, X'w^{(2)}_j]$ implies $X \sim X'$, which is essentially Eq. 5 $\Rightarrow$ Eq. 6.
>
> Lemma 4.4 plays a critical role in our proof, as the construction therein mixes anchor channels with the original feature channels. And this construction guarantees that alignment between mixed channels can induce alignment between the tuples of anchor and feature channels.
>
> **3. Question regarding "The obtained results for $D' = 1$ can also be easily extended to $D' > 1$ as otherwise f can be written as $[f_1, \cdots, f_D']^\top$ and each $f_i$ has single output feature channel".**
>
> We apologize for the confusion caused. The original definition in Def. 2.2 states a general permutation-invariant function can admit vector outputs of dimension $D'$. Note that here $D'$ denotes the number of output channels (e.g., classification logits) instead of number of output set elements. We used the aforementioned argument to clarify that our investigation starting from Sec. 2.2 will only focus on scalar outputs without loss of generality. Our permutation-equivariant results are not as trivial as extending the dimension of the outputs. It requires some additional theoretical establishment Lemma G.1 to extend the proof of Theorem 3.1.
>
> **4. Practical significance and empirical verification.**
>
> In many practical applications of DeepSets-like architectures, even though the embedding dimension is not carefully chosen, it is observed that a reasonably small latent dimension is sufficient to let those networks perform well. Such empirical observations appear to contradict existing theoretical studies, in which exponential many neurons are often needed. In this work, we provide a more efficient construction of DeepSet embedding layers, which results in polynomially many hidden neurons for the first time, successfully explaining the empirical promise of DeepSets-like architectures. Designing experiments to directly testify our theoretical bound can be inherently challenging. We provide a preliminary experiment in Appendix K, in which we show that the minimal hidden dimension required to achieve a small training error scales with sequence length and feature dimension at a polynomial rate.

---

> > ### Comment · Reviewer_a75R · 2023-11-23
> > **Thank you for the response**
> >
> > Thank you for the response. I will raise my score. I would strongly recommend to add a section about practical takeaways and to move the existing empirical experiment to the main text if possible.

---

### Official Review · Reviewer_XKbR · 2023-11-03

**Soundness:** 4 excellent
**Presentation:** 3 good
**Contribution:** 3 good
**Rating:** 5
**Confidence:** 3

**Summary:**

## High level summary
This paper delves into the question of representations of sets as continuous functions in vector spaces. The work builds up on the ideas proposed in DeepSet, which shows that proves any permutation-invariant continuous function, can be restated by mapping each element to a vector, summing those vectors, and then mapping them to a scalar again. This paper addresses the unresolved question, whether sets over high-dimensional vectors can be represented efficiently, i.e., polynomially wrt to the set and feature size.

More formally, $X\in R^{N\times D}$ represents $N$ feature vectors of dimension $D,$ and function $f:R^{N\times D} \to R$ is defined to be permutation-invariant if for any permutation matrix $P$ it satisfies $f(P X) = f(X).$  The main contributions of the paper can be enumerated as
- Thm 3.1 asserts that there $\phi:R^D\to R^L$ and $\rho:R^L\to R$ such that $f$ can be re-written as $f(X) = \rho(\sum_{i=1}^N \phi(x^{(i)}))$ where $L$ is at most polynomial wrt to $N$ and $D$, with two different constructions that authors refer to as "power mapping " and exponential activation". The theorem further asserts that $L$ is lower-bounded roughly by $ND$.
- Thm 5.1 further shows that permutation equivariant functions, i.e., $f:R^{N\times D}\to R^{N\times D'}$ such that  $f(P X) = P f(X),$ then there are $\phi:R^D\to R^L$ and $\rho:R^D\times R^L\to R$ such that $f(X)_j = \rho\left(x^{(j)}, \sum_i^N \phi(x^{(i)})\right)$ where again $L$ is polynomial in $N$ and $D$.

## Technical summary

*Set representation for one-dimensional elements $D=1$* The previous result proves that when $D=1$ there are $\phi:R\to R^L$ and $\rho:R^L\to R$ such that $f = \rho(\sum_i^N \phi(x_i)$). This result mostly hinges on a particular function, referred to as power-mapping $\Psi_N :R^N\to R^N$, defined as  $\Psi_N(X)_k = \sum_i^N (x^{(i)} )^k.$ The paper goes on to explain that $\Psi_N$ is "bijective" in a particular sense that deviates from the standard definition, in that $\Psi_N(X) = \Psi_N(X')$ implies rows of $X$ are a permutation of $X'.$ This allows us to introduce the mapping and its inverse as an identity and conclude $ f = f \circ \Psi_N^{-1} \circ \Psi_N  = \rho(\sum_i^N \phi(X))$ where $\rho:= f\circ \Psi_N^{-1}$ and $\phi:= \psi_N$.

*The channel alignment problem in $D\ge 2$* The first insight is explaining that the straightforward approach of representing each channel (dimension) of the high-dimensional features, will not work. This is because the mapping $\Psi_N:R^N\to R^N$ is permutation-invariant, meaning that while extending it to high dimension will preserve permutation invariance across each channel, these permutations are not going to be necessarily the same, referred to as alignment. This means that while we can ensure that while the naive approach ensures that each channel is preserved with permutation invariance, these permutations are not necessarily the same, and thus the vectors structure, where indices of vector do matter, may be lost.

*Linearly lifting dimension to $L$ to resolve alignment* The main idea behind the proposed theory, as far as I can understand, is how to prevent this *alignment problem* by linearly lifting the $D$-dimensional elements to a much higher dimension $L$ that has lots of redundancies. Crucially, this lifting has the property channels of two different matrices after lifting can be aligned independently, then one matrix is a row-permutation of another.

Here I will summarise the more detailed theoretical insights I can draw from various parts of the paper
- *Anchor.* The main idea for the entire theoretical construction in fact develops on top of this alignment problem, suggesting a way to "anchor" different channels. The theoretical construct. Formally, anchor $a\in R^N$ for data $X \in R^{N\times D}$, preserves the equality structure of the rows of $X$: $x^{(i)}\neq x^{(j)}\implies a_i\neq a_j$.
- *Coupling of alignment with anchor.* If $a$ is an anchor of $X$, and $a'$ is a permutation of $a,$ and same permutation $P$ can be applied on every channel of $X$ to transform to the equivalent channel of $X'$, then rows of $X$ are a permutation of rows of $X'$. Thus, we can couple the alignment of various channels to the alignment of $a$ to $a',$ we can easily conclude that $X\sim X'$
-  *Linear probes for anchors. * We can pick linear probes $w_1,\dots, w_k\in R^D,$ such that for any matrix $X,$ at least one of the $X w_1,\dots, X w_k$ must be an anchor of $X$. This is achieved by picking a large enough $K$, and $w_k$'s such that every subset of size $D$ will be linearly independent, i.e., in general position, which is achievable by simply drawing from some Gaussian multivariate distribution. The claim follows from a cute pigeonhole principle. Now, if we do set representation of all $x^{(i)}$ and these linear probes $X w_k$'s, we can ensure that each of them are injective, and at least one of the $X w_k$'s is an anchor, but we still haven't coupled the alignment of the anchor to the rest of the channels.
- *Coupling channel alignments to the anchor* Next, authors prove existence of coefficients $\gamma_1,\dots, \gamma_K$ such that if$x\sim x', y\sim y'$ and  $x - \gamma_k y$ is a permutation of $x' - \gamma_k y'$ for every $k = 1, \dots, K$, then permutation of $x\sim x'$ and $y\sim y'$ can be aligned.
- *Lifting* Here, using the linear probes $w_k$ and coefficients $\gamma_k$'s from previous step, we can define lifting operator colums  $w_{i,j,k} = e_i - \gamma_k w_j$  for a polynomial range of $i,j,k$, and put all of them as columns of the lifting linear $W \in R^{D\times L}.$ Then each channel (column) of $Y = X W$ will be of the form $x^{(i)} - \gamma_k X w_j.$ Therefore, by previous properties, if every channel (column) of $X W$ can be aligned to (is a permutation of) $X' W$, then rows of $X$ are a permutation of $X'.$ The *set representation* of each channel will follow naturally from this step.

**Strengths:**

Here are the main strengths I find in the appear:
- The theory of the paper, to be best of my understanding, is sound and accurate. The statement of the theorems, lemmas, and the proofs, as much as I delved into, are sound and clearly stated.
- The main problem that this paper focuses on, is a natural abstraction of real-world problems.
- There are many clever and intuitive proof techniques used in this work, which may be interesting for the reader

**Weaknesses:**

Main issues:
- While the paper presents a mathematically intriguing case, I am not quite sure what to draw from it from a machine learning perspective.  While theoretical contributions should certainly be welcome in the ML field, I think the theoretical works should take a few steps in demonstrating the relevance of their results for the broader community. For example, are there any concrete examples or use cases that these theoretical findings would be relevant?
- Following up on the previous point, despite the embedding-sum decomposition, the construction of the $\rho$ function $\rho:R^L \to R$ is still a complete black box. Again, these non-constructive arguments do not seem to add up to any "practical insight." While this is not a 100\% a critical point, if authors can think of ways to make it easier for the reader to imagine applications, they would broaden their audience and enhance the impact of the paper.
- While the paper is mathematically sound, it could benefit from more high-level idea developments both before and after the theorems. There are several places where a geometric interpretation is available, but not discussed in the main text. For example in the proof of Lemma 4.2 it becomes clear that a system of $w_k$'s in general position, if there are "enough" of them, at least one will be one that is not in the hyperplane orthogonal to difference between each two columns of $X$, perhaps this can even be visualised for some toy example with $D=2$ and $N=3.$ With the intuition that authors have built upon this case, if they are able to come up with more intuitive/visual depiction of these concepts, it would dramatically improve the paper's readability and accessibility.

 Minor issues:
- The notion of injectivity for functions (Def 2.7), that $f(X) = f(X')$ implies rows of $X$ are a permutation of rows of $X$', slightly deviates from the standard notion that assumes "a function f that maps distinct elements of its domain to distinct elements." It took me quite a few iterations to notice this slight difference. Perhaps author can caution the reader before or afterwards. Most readers will be inclined to assume the "default" notion which could lead to some confusions later on.
- The notation $[a\ x]\sim [a'\ x']$ which (for example used in Lemma 4.2) is somewhat ambiguous. Initially, I interpreted it as concatenating the two vectors along their main axis $[a\ x] \in R^{2N}$ which leads to lots of contradictions, while the authors implied stacking them along a new dimension $[a\ x]\in R^{2\times N}$ which makes sense. While this might be consistent with the papers notation elsewhere, it is still good and helpful to highlight it for the readers that this implies stacking them and not concatenation.
- "Comparison with Prior Arts"? I'm guessing authors mean "articles" here? Is this a common shorthand for articles? While I don't want to sound authoritative, this seems like a rather informal writing style.
- (page 17) "pigeon-hold principle" I'm guessing authors refer to "pigeonhole principle" :)

**Questions:**

- The main results of this paper are obtained for a notion of permutation-invariance of $f,$ which is somewhat strictly defined over the entire input set $X.$ In real world applications, often the feature vectors $x$ represent learned or encoded features of a particular dataset. So in these cases, the "user" will be interested in "relaxed set preserving properties", in the sense that permutation invariance is only held over a subset (possibly even countable subset) of $R^{N\times D}$. Can authors think of any interesting relaxations  on $X$ and extend their theory for those? (assuming it has certain properties or its feature vectors are chosen from a smaller/finite set)
- Upon reading the paper, the architecture that authors propose resembles a network with depth "2" (perhaps this mysteriously encoded $\rho$ , which in reality could be a highly complex $R^L\to R$ function). In certain scenarios, there could be a hierarchical set representations, e.g., we want to embed sets of sets of sets, (or even more). Can authors comment on extendability of their theory, or in general any comments they may have for such multilevel/hierarchical sets?

---

> ### Author Response · Authors · 2023-11-21
> **Response to Reviewer XKbR**
>
> We genuinely thank Reviewer XKbR for acknowledging our theoretical soundness and providing a detailed summary of our technical insights. Regarding your questions, please see our response below:
>
> **1. Practical implications and relevance to the ML community.**
>
> When the input data is a set or a graph, machine learning models often require permutation invariance or equivariance. DeepSets or sum-pooling-based architectures are ubiquitous in these models. Representative examples encompass GNNs and PointNet. GNNs learn to pass information along graph topology via a neighborhood aggregation operation at each layer of computation, essentially corresponding to a set function [1]. PointNet [2] processes point clouds by representing points as an unordered set and follows a DeepSets-like architecture. However, the analysis of expressive power for most of these usages does not take the embedding dimension into consideration, as also acknowledged by Reviewer a75R. Our work provides a rigorous justification that moderately many neurons are sufficient to represent a high-dimensional set with DeepSets. This affirms the feasibility of the DeepSets architecture given high-dimensional features and explains why DeepSet-like operations can be effectively adopted in GNNs and PointNet.
>
> [1] Xu et al., How Powerful are Graph Neural Networks?
>
> [2] Qi et al., Pointnet: Deep learning on point sets for 3d classification and segmentation
>
> **2. Construction of $\rho$ remains black-box**
>
> Among most of works analyzing expressiveness of DeepSets architectures [1,2], identifying bijectivity (also known as orbit separation) of the sum-pooling layer is of most theoretical interest. We note that the resultant $\rho$ in our construction is not a "complete" black box. Our construction guarantees continuity of $\rho$, a crucial property which allows it to be universally approximated by a commonly-used neural network. We leave more fine-grained discussion of $\rho$ to our future work and a potential direction can be analyzing the learnability of $\rho$ depending on smoothness of the embedding layer ($\phi$).
>
> [1] Zaheer et al., Deep Sets
> [2] Wagstaff et al., On the Limitations of Representing Functions on Sets
>
>
> **3. More intuitive interpretation of our proof.**
>
> We appreciate the reviewer's insightful interpretation of Lemma 4.3 and we have included it in our latest revision. We have also added some high-level proof ideas upon Reviewer jrmV's suggestion and would like to direct Reviewer XKbR to Fig. 2, an illustration of our proof, which Reviewer oY1k found helpful.  Although the majority of our proofs are algebraic, a simple geometric interpretation could be that we identify a non-trivial subspace embedded in a higher-dimension ambient space such that for any two points in this subspace, the intersection of their orbits, generated by column-wise permutations, signifies the orbit intersection after a global row permutation.
>
>
> **4. Typos and confused notation.**
>
> Thanks for pointing out typos. We have fixed all the typos and added additional remarks for confusing notations in our updated manuscript. The notion of injectivity we adopted here also follows a standard definition, as "$X \not\sim X' \Rightarrow f(X) \neq f(X')$" is equivalent to "$f(X) = f(X') \Rightarrow X
> \sim X'$" by contrapositive. The term "prior arts" is usually used interchangeably with "prior results", referring to previous works showcasing state-of-the-art results, as exemplified in [1]. To avoid further confusion, we change it to "prior results".
>
> [1] Tsakiris & Peng, Homomorphic sensing

---

> ### Author Response · Authors · 2023-11-21
> **(Cont.) Response to Reviewer XKbR**
>
> **5. Relaxation of the input space.**
>
> Most discussion on permutation-invariant models focuses on the invariance defined over the entire ambient space. Meanwhile, it is common in network expressiveness analysis to consider a general input (e.g., arbitrary open/close/compact sets). We appreciate reviewers' thoughtful suggestions on considering a constrained input space. If we know a part of elements in $X$'s order is irrelevant while other elements' order matter, one can apply the idea of DeepSets to the first subset of elements and concatenate the output with the representations of the remaining elements, and finally adopt standard fully connected networks to process the concatenation. This will still give universal approximation. Our theory is still applied to the DeepSet part. Some other relaxation can also be explored in our follow-up works. For instance, $X$ can be assumed from a low-dimensional manifold and further obey permutation invariance/equivariance. However, such discussion is beyond the scope of this work. Nevertheless, our results considering the most general inputs can already cover a lot of practical scenarios such as GNNs and PointNets.
>
> **6. Solution to hierarchical set.**
>
> Extending our results to hierarchical sets is beyond the scope of this paper. A straightforward solution based on our current results could be hierarchically apply the sum-pooling $\Phi$ and append a final output layer $\rho$. For instance, consider a set of depth 2: $X \in \mathbb{R}^{M \times N \times D}$, one can provably realize a function in terms of this special invariant structure by:
> $$
> f(X) = \rho(\sum_i^M \phi(\sum_j^N \phi(X^{(i,j)})))
> $$
> where $\rho = f \circ \Phi^{-1} \circ \Phi^{-1}$. If LP embedding layer is adopted, the minimal needed latent dimension for the innermost sum-pooling is upper bounded by $O(N^5D^2)$ and for the outer sum-pooling is upper bounded by $O(M^5N^{10}D^4)$.

---

### Meta-Review · Area_Chair_NbDh · 2023-12-12

**Metareview:**

Summary: The article investigates the impact of embedding dimension on the expressive power of DeepSets.

Strengths: Referees found the paper presented novel ideas, the contributions substantive and technically interesting, appreciated a sound and accurate theory, found the problem relevant, and proof techniques thorough and rigorous.

Weaknesses: On the critical side, a referee expressed concern about unclear implications from a machine learning perspective, lack of practical insight, although they also pointed out this was not a critical point. They also suggested providing more high level ideas. Following the discussion period some of these concerns could be clarified, prompting some referees to raise their recommendations.

In view of the positive reception, I am recommending accept. The authors are strongly encouraged to work on making the intuitions as clear as possible as well as the practical takeaways in the final version of the manuscript, as this was a general point in the reviews.

**Justification For Why Not Higher Score:**

At the end of the discussion period referees still strongly recommend to add a section about practical takeaways.

**Justification For Why Not Lower Score:**

The strengths are well above the weaknesses; the article had a generally positive evaluation from all referees.

---

### Decision · Program_Chairs · 2024-01-16

Accept (poster)